# METTL9 sustains vertebrate neural development primarily via non-catalytic functions

Azzurra Codino [1], Luca Spagnoletti[1], Claudia Olobardi[2], Alessandro Cuomo[3], Helena Santos-Rosa[4], Martina Palomba[5], Natasha Margaroli[5], Stefania Girotto [5], Rita Scarpelli [5], Shi-Lu Luan[6,11], Eleonora Crocco [7], Paolo Bianchini [8], Andrew J. Bannister [4], Stefano Gustincich[1], Tony Kouzarides[4], Riccardo Rizzo[9], Isaia Barbieri [6,10], Federico Cremisi[7], Robert Vignali [2] & Luca Pandolfini [1] ✉

METTL9 is an enzyme catalysing N1-methylation of histidine residues (1MH) within eukaryotic proteins. Given its high expression in vertebrate nervous system and its potential association with neurodevelopmental delay, we dissected *Mettl9* role during neural development. We generated three distinct mouse embryonic stem cell lines: a complete *Mettl9* knock-out (KO), an inducible METTL9 Degron and a line endogenously expressing a catalytically inactive protein, and assessed their ability to undergo neural differentiation. In parallel, we down-regulated *mettl9* in *Xenopus laevis* embryos and characterised their neural development. Our multi-omics data indicate that METTL9 exerts a conserved role in sustaining vertebrate neurogenesis. This is largely independent of its catalytic activity and occurs through modulation of the secretory pathway. METTL9 interacts with key regulators of cellular transport, endocytosis and Golgi integrity; moreover, in *Mettl9^{KO}* cells Golgi becomes fragmented. Overall, we demonstrate a developmental function of *Mettl9* and link it to a 1MH-independent pathway, namely, the maintenance of the secretory system, which is essential throughout neural development.

Seven-β-strand methyltransferase-like (METTL) proteins encompass a diverse group of enzymes sharing the evolutionary conserved *S*-adenosylmethionine (SAM) binding domain, which endows them with catalytic activity[1,2]. Upon SAM cofactor binding, METTL proteins can transfer methyl groups from SAM to different macromolecules, such as RNA or proteins.

Amplification and deletion of various METTL genes have been associated with different pathological conditions, including cancer,

[1]Istituto Italiano di Tecnologia (IIT), Center for Human Technologies, Via Enrico Melen 83, 16152 Genoa, Italy. [2]Department of Biology, University of Pisa, via Luca Ghini 13, 56126 Pisa, Italy. [3]Department of Experimental Oncology, IEO, European Institute of Oncology IRCCS, Milan 20139, Italy. [4]The Wellcome Trust/Cancer Research UK Gurdon Institute and Department of Pathology, University of Cambridge, Tennis Court Road, Cambridge CB2 1QN, United Kingdom. [5]Istituto Italiano di Tecnologia (IIT), Center for Convergent Technologies, Via Morego 30, 16163 Genoa, Italy. [6]University of Cambridge, Dep. of Pathology, 10 Tennis Court Road, Cambridge CB2 1QP, United Kingdom. [7]BIO@SNS, Scuola Normale Superiore di Pisa, Via Giuseppe Moruzzi, 56124 Pisa, Italy. [8]NIC@IIT, Italian Institute of Technology (IIT), Via Enrico Melen 83, 16152 Genoa, Italy. [9]Institute of Nanotechnology, National Research Council (CNR-NANOTEC), Campus Ecotekne, Via Monteroni, 73100 Lecce, Italy. [10]Department of molecular biotechnology and health sciences, Molecular Biotechnology Center, University of Turin, Via Nizza 52, 10126 Turin, Italy. [11]Present address: Protein & Nucleic Acid Chemistry Division, MRC Laboratory of Molecular Biology, Cambridge, United Kingdom. ✉e-mail: luca.pandolfini@iit.it

where the same METTLs can act either as tumour suppressors or as oncogenes, depending on the tissue and cellular context[3]. Besides their established role in cancer, METTLs sustain many physiological processes, including cell fate specification and differentiation[3]. METTL3 maintains hematopoietic stem cell quiescence[4] besides acting as an oncogene in acute myeloid leukemia[5]. METTL1 safeguards self-renewal potential in mouse embryonic stem cells (mESCs) and regulates ectodermal and neural differentiation of mESCs[6]. Similarly, METTL6 is important for preserving mESCs pluripotency[7] and METTL17 knock-out (KO) delays embryonic stem cell differentiation by decreasing mitochondrial respiration[8]; METTL11A sustains myoblasts differentiation[9] and prevents premature ageing through the maintenance of the quiescent neural stem cell pool[10].

Histidine (His) methylation is a post-translational modification whose biological significance has remained elusive for a long time[11]. Only 2 His-specific methyltransferases have been identified: METTL18, which methylates the nitrogen in position 3 of histidine imidazole ring (generating 3-methylhistidine, 3MH) mainly in RPL3 protein, modulating translational elongation[12,13] and METTL9, which was found to methylate the nitrogen 1 of histidines (generating 1MH) in hundreds of proteins containing the H[ANGST]H motif[14]. More recently, the structural details of METTL9 substrate recognition and catalysis have also been clarified[15,16]. METTL9 targets include mitochondrial respiration factors like NUFB3 and zinc transporters, whose zinc binding affinity is changed upon METTL9 KO[14].

Similarly to other METTL proteins, METTL9 has also been linked to cancer biology. It is upregulated in many tumours (where its expression level correlates with poor prognosis) and decreasing METTL9 expression in hepatocellular carcinoma cells slows down their proliferation and ability to generate tumours in vivo as well as their invasiveness[17]. Moreover, METTL9-dependent methylation of certain zinc transporters is associated with tumour growth[18]. Interestingly, methylation of the immunomodulatory S100A9 protein[14,19] is dynamically regulated in neutrophils upon infection[20]. Low levels of S100A9 methylation endow mice with a better anti-bacterial ability by increasing S100A9 affinity to zinc[20]. However, besides its role in cancer and in immune response, the biological function of METTL9 in physiological and developmental processes has remained unknown. Here we show that METTL9 is highly expressed in the vertebrate nervous system where it sustains early neurogenesis. By using complementary genetic systems, we also reveal that this function is mainly independent of its catalytic activity and that it is linked to the secretory pathway and Golgi integrity.

## Results

### Mettl9 is important for vertebrate neural development

To uncover the physiological relevance of Mettl9, we first sought to assess whether the human *METTL9* locus was associated with any disease. Importantly, by mining the DECIPHER database[21], we identified 6 patients harbouring heterozygous deletions of a genomic region encompassing the *METTL9* gene in chromosome 16 (Fig. 1a). Interestingly, 5 out of the 6 patients displayed at least one nervous system-related phenotype such as cognitive impairment, autistic behaviour, morphological central nervous system abnormality and intellectual disability, while the sixth patient displayed global developmental delay (Fig. 1a). Some of them also showed varying degrees of hearing impairment, possibly ascribed to the loss of the inner ear specific gene Otoancorin, *OTOA*, which is also present within the deleted region together with the *IGSF6* gene, and was previously linked to deafness[22,23] (See Fig. 1a legend).

Since the phenotypes associated with these *METTL9*-containing deletions suggested a putative role for *METTL9* in neurodevelopmental disorders, we assessed *METTL9* expression in the developing brain during prenatal life. By interrogating the BrainSpan Atlas of the Developing Human Brain[24], we found that *METTL9* is expressed at high levels during human neural development, while its expression decreases in postnatal brain (Supplementary Fig. 1a). Within the adult brain *METTL9* levels peak in the striatum (Supplementary Fig. 1a, 30-40 yrs), which was also confirmed by the three-dimensional map of adult human brain (Allen Human Brain atlas[25]) (Fig. 1b) and exploration of the GTEx datasets[26] (Supplementary Fig. 1b). Furthermore, we confirmed that *Mettl9* is highly expressed in the direct and indirect spiny projection neurons of the mouse striatum, as shown by re-analysis of available scRNA-seq data[27] (Supplementary Fig. 1c). In summary, given the high expression levels of METTL9 in mouse and human brain, as well as the neurodevelopmental phenotypes potentially related to its deletion in patients, we hypothesised that Mettl9 dosage could be important for sustaining early neural development.

Since the most crucial and earliest events of mammalian brain development are shared among vertebrates[28], the African clawed frog *Xenopus laevis* represents an exceptional model organism for phenotypic screening of highly conserved genes with a putative developmental function, mainly due to the ease of in vivo embryo manipulation[29].

The *Mettl9* gene is evolutionarily conserved across the animal kingdom[2], and the encoded protein is highly conserved between *Mammalia* and *Amphibia* (*Xenopus laevis)* both in terms of amino acid sequence (which shows an overall similarity of 82.4% and 82.7% with *Mus musculus* and *Homo sapiens*, respectively; Supplementary Fig. 2a) and structure (Supplementary Fig. 2b), particularly in the region encoding the catalytic SAM binding domain (Supplementary Fig. 2a, b). Therefore, we reasoned that its core developmental functions likely originated early in evolution and have since been conserved. Thus, we performed whole-mount RNA in situ hybridization (WISH) and analysed the spatio-temporal expression pattern of *mettl9* throughout *X. laevis* early development (Supplementary Fig. 2c). Interestingly, *mettl9* mRNA was detected very early during embryogenesis, in the NF6.5-NF7 blastula (Supplementary Fig. 2d), probably due to maternal genome contribution. It remained highly expressed at gastrula stage (NF10.5), following zygotic gene activation and germ cell layer and axis specification. Importantly, *mettl9* was highly expressed during early neurulation (NF14-18 stages) (Fig. 1c), both in the neural fold (nf) and neural plate (np), before neural tube folding and closure (NF14 and 18). After the completion of neurulation, *mettl9* expression becomes more restricted to the anterior part of the early (NF20-23) (Supplementary Fig. 2d) and late (NF26 and 31) tailbud stages (Fig. 1c), in the encephalon (en) and optical vesicles (ov; NF26) (Fig. 1c), and also in neural crest cells (ncc; NF31) (Supplementary Fig. 2e).

These data strongly indicate that *mettl9* is highly expressed in the nervous system during early amphibian development and its levels in the brain remain significant at later stages. Importantly, these data are consistent with mammalian expression patterns, suggesting that Mettl9 expression in the developing nervous system is a conserved feature of vertebrates.

We then set out to investigate the requirement of *mettl9* for neurogenesis in vivo, by analysing *mettl9* knock-down (k.d.) embryos. To this end, we designed a *mettl9*-morpholino antisense oligonucleotide (MO) which targets the exon1/intron1 junction of the *mettl9* pre-mRNA, thus impairing *mettl9* splicing (Supplementary Fig. 2f), as confirmed by qPCR (Supplementary Fig. 2g). We injected *mettl9*-MO or *ctrl*-MO into one of the two dorsal blastomeres of 4-cell stage embryos (Fig.1d), leaving the contralateral side of the embryo as an internal control. To assess whether *mettl9*-MO embryos were able to undergo early neurogenesis, we analysed the expression pattern of key neural markers by WISH: strikingly, we found that *neurog2*, a marker of neuronal precursors[30], *notch*, involved in regulation of lateral inhibition within the developing neural plate[31] and *elrC*, a marker of committed neuronal progenitors[32], were down-regulated at the neurula stage (Fig. 1e–g). In particular, *neurog2* was down-regulated in the medial,

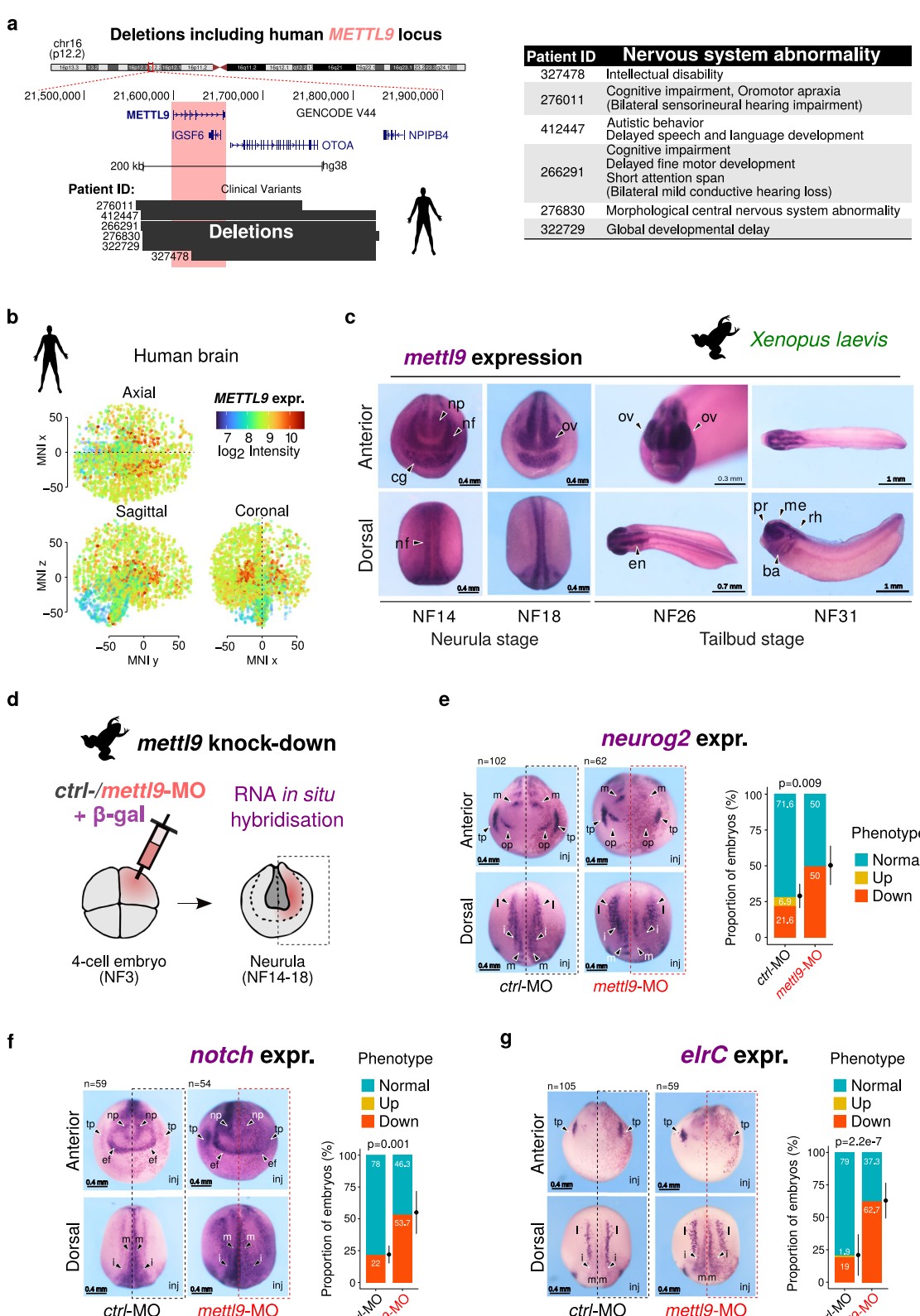

intermediate and lateral columns of prospective neuronal precursors, and also in the olfactory and trigeminal placodes (Fig. 1e, arrowheads); *notch* pattern was severely affected on the injected side both in the anterior domain around the presumptive eye field and in the trunk region (Fig. 1f, arrowheads); and *elrC* was also affected in the developing neurons (Fig. 1g), as shown by alterations in the intermediate

neuron precursors and trigeminal precursors (arrowheads). These effects were found with a significantly higher proportion in *mettl9*-MO injected embryos than in *ctrl*-MO injected embryos (Fig. 1e–g).

Overall, these data supported the idea that *mettl9* may exert a role in early neurogenesis and prompted us to further dissect its function in the context of a mammalian model.

**Fig. 1 | Mettl9 is expressed in vertebrate nervous system and is important for *X. laevis* neural development in vivo. a** Deletions encompassing human *METTL9* locus on Chromosome 16 in patients, who display nervous system-related phenotypes (table on the right), from DECIPHER. Deletions also include: the *OTOA* gene; the *IGSF6* gene encoding a predicted transmembrane receptor of the immune system and, in few cases, also the *RRN3P1* pseudogene. **b** *METTL9* normalised microarray expression level (log₂ intensity) in the three sections of a three-dimensional map of adult human brain (Allen Human Brain Atlas). Coordinates are referred to the Montreal Neurological Institute (MNI) standard spatial template. **c** Representative anterior and dorsal views images of *X. laevis* embryos showing *mettl9* mRNA expression (purple) by whole-mount RNA in situ hybridisation (WISH). At Niewkoop-Faber stage 14 (NF14), black arrowheads indicate: neural plate (np), neural fold (nf) and cement gland (cg) and at NF18 they show optical vesicles (ov). At tailbud stage (NF26) it is shown the encephalon (en) which, at NF31, is subdivided into prosencephalon (pr), mesencephalon (me) and rhombencephalon (rh). Branchial arches (ba) are also shown at NF31. N ≥ 30 embryos. **d** *mett9* knock down (k.d.) strategy at 4-cell stage embryo: microinjection of *mettl9* morpholino oligonucleotide (MO) (pink) in the dorsal left blastomere is shown. WISH is then performed at later stages (neurula) to assess potential developmental abnormalities in the injected side of embryos (right side, pink). **e–g** Representative images of *ctrl*-MO and *mett9*-MO k.d. embryos at neurula stage (NF14), showing the expression of neural markers *neurog2* (*ngn*) (**e**), *notch* (**f**) and *elrC* (**g**) by WISH. Lateral (*l*), intermediate (*i*) and medial (*m*) stria, trigeminal and olfactory placodes (tp, op), neural plate (np) and eye field (ef) are shown. Arrowheads indicate the regions affected in *mettl9*-MO. Inj.: injected side. Bar graphs on the right show the quantification of *ctrl*- or *mettl9*-MO k.d. embryos screened for altered *neurog2*, *notch* or *elrC* expression (χ2 test). The number of embryos analysed for each group is shown above the respective panels. Error bars are mean ± SD.

## Constitutive depletion of Mettl9 from mESCs impairs neural fate commitment and neural differentiation

Given that *mettl9* depletion negatively affected early neural development in *X. laevis* embryos, we speculated that loss of Mettl9 would similarly impair mammalian neural fate specification. To address this, we used mouse embryonic stem cells (mESCs) to study neural commitment and differentiation. mESCs can faithfully recapitulate the main steps of mammalian neurogenesis in a few days in vitro and allow fine genetic and molecular manipulations. Thus, we steered mESCs towards a neural fate by culturing them in the absence of serum/leukaemia inhibitory factor (LIF) and by inhibiting both BMP and Wnt signalling, as previously described[33,34] (Fig. 2a). After 5 days in culture with Wnt/BMP inhibitors (DIV5), mESCs acquire neural stem cell (NSC) identity, as shown by the expression of the neural markers *Nestin*, *Dlx2* and *Gsx2* (Supplementary Fig. 3a). Then, after supplying Sonic Hedgehog (SHH) agonist from DIV5 until DIV10, NSCs acquire the identity of ventral telencephalic neural progenitor cells (NPCs), as shown by the progressive up-regulation of the striatal genes *Drd1a*, *Drd2* and *Gad1* (Supplementary Fig. 3a). Importantly, we observed that *Mettl9* was also up-regulated both at the mRNA (Fig. 2b) and protein level (Fig. 2c) upon mESCs commitment to NSCs, and it reached even higher levels in NPCs. Overall, these data are consistent with our previous findings in the *X. laevis* and mammalian studies, reinforcing the notion that our cell model is suitable to study the physiological roles of *Mettl9* during mammalian neurogenesis.

We then constitutively depleted *Mettl9* from mESCs and assessed their ability to undergo neural priming and differentiation. To this end, we engineered a *Mettl9* knock-out (*Mettl9^{KO}*) mESC line using CRISPR/Cas9 technology, by targeting the first exon of the endogenous *Mettl9* locus with 3 sgRNAs (Fig. 2d). We obtained two *Mettl9^{KO}* clones (#88 and #90), each harbouring biallelic deletions within the first exon of *Mettl9* (Supplementary Fig. 3b, c), which resulted in the complete loss of METTL9 protein, as shown by WB (Fig. 2e). Consistently, *Mettl9* mRNA levels were almost undetectable by qPCR in *Mettl9^{KO}* cells (Supplementary Fig. 3d), suggesting that METTL9 degradation was most likely due to the non-sense mediated mRNA decay pathway[35]. Importantly, the sgRNAs used did not generate off-target mutations across the genome, as confirmed by whole-genome sequencing (Supplementary Fig. 3e).

We first assessed whether *Mettl9^{KO}* mESCs could differentiate into bona fide NSCs; strikingly, we found a significantly lower number of neural NESTIN-positive (NES⁺) cells in *Mettl9^{KO}* cultures compared to the *Mettl9^{WT}* (parental line, E14), as shown by immunofluorescent (IF) staining of NSCs and cell counting, at DIV6 (Fig. 2f).

To gain a comprehensive understanding of the molecular processes affected by *Mettl9* loss, we profiled gene-expression of *Mettl9^{KO}* NSCs by RNA-seq (Fig. 2g). Importantly, at DIV5 we found severe transcriptomic alterations, as we identified 5732 mis-regulated genes in both the #88 and #90 clonal *Mettl9^{KO}* NSCs compared to the controls clonal and parental (E14) *Mettl9^{WT}* cell lines, with 2862 down-regulated and 2870 up-regulated genes (Supplementary Fig. 4a; Supplementary Data 1). Notably, both *Mettl9^{KO}* clones exhibited a very consistent gene regulation, distinct from both parental (E14) and clonal (i.e. non-edited) wild-type (*Mettl9^{WT}*) lines (Supplementary Fig. 4b). As anticipated by the decrease in NES⁺ neural cells shown by IF, among the most down-regulated Gene Ontology (GO) Molecular Function terms we found many neural-related entries such as "neurogenesis" and "neuron differentiation" (Fig. 2h) (refer to Supplementary Data 2 for full GO list). These were exemplified by down-regulation of the NSC markers *Sox1*, *Nestin*, *Foxg1* and *Shh* (Supplementary Fig. 4c) as well as of the early basal telencephalic gene markers *Dlx2*, *Lhx1*, *Ascl1* and *Foxp2* (Fig. 2i) and *Dlx1*, *Lhx5* (Supplementary Fig. 4c), which was accompanied by the up-regulation of stemness (mESC) markers such as *Pou5f1*, *Nanog*, *Nodal* (Supplementary Fig. 4c). Interestingly, "cell projection organization" was also found among the most down-regulated GO terms. Consistently, some of the most down-regulated GO terms concerning Cellular Component were: "synapses", "neuron projection", "microtubule cytoskeleton", "Golgi apparatus" and "ER" (Fig. 2j; Supplementary Data 2). Moreover, among the up-regulated Molecular Functions we found many metabolic and biosynthetic processes (Supplementary Fig. 4d; Supplementary Data 2). Overall, these transcriptomic data indicate that upon *Mettl9* loss, differentiating mESCs cannot completely shut down their stemness program and cannot undergo a neurogenetic route, thus precluding their acquisition of a typical NSCs identity.

After discovering the impairment in NSC commitment at DIV5, we wondered whether *Mettl9^{KO}* NSCs could undergo neural differentiation. By DIV10, *Mettl9^{KO}* NPCs became morphologically different from their *Mettl9^{WT}* counterparts, as they lacked neural projections and resembled fibroblast-like cells (Supplementary Fig. 4e). This strong phenotype was confirmed by the large number of mis-regulated genes found in both clonal *Mettl9^{KO}* lines at DIV10 (3298 down-regulated genes and 3232 up-regulated genes) compared to control *Mettl9^{WT}* NPCs (Supplementary Fig. 4f, g; Supplementary Data 1). Next, we inferred the identity of these aberrant *Mettl9^{KO}* cells, through the deconvolution of cell type composition using neural-networks[36] trained over single-cell atlas of developing mouse brain[37]. Importantly, this analysis revealed that *Mettl9^{KO}* cells at DIV10 had a relatively higher proportion of neural tube cells (Fig. 2k), usually present at earlier stages of neural development, a decrease in radial glial cells and a massive increase in other cell lineages such as mesoderm and ectoderm (e.g. fibroblast-related), as anticipated from the macroscopical observation of the cellular phenotype. Overall, constitutive loss of *Mettl9* prevents mESCs from generating bona fide NSCs and, at later stages, massively impairs the specification of NPCs, while promoting the aberrant acquisition of mesodermal and non-neural ectodermal identity.

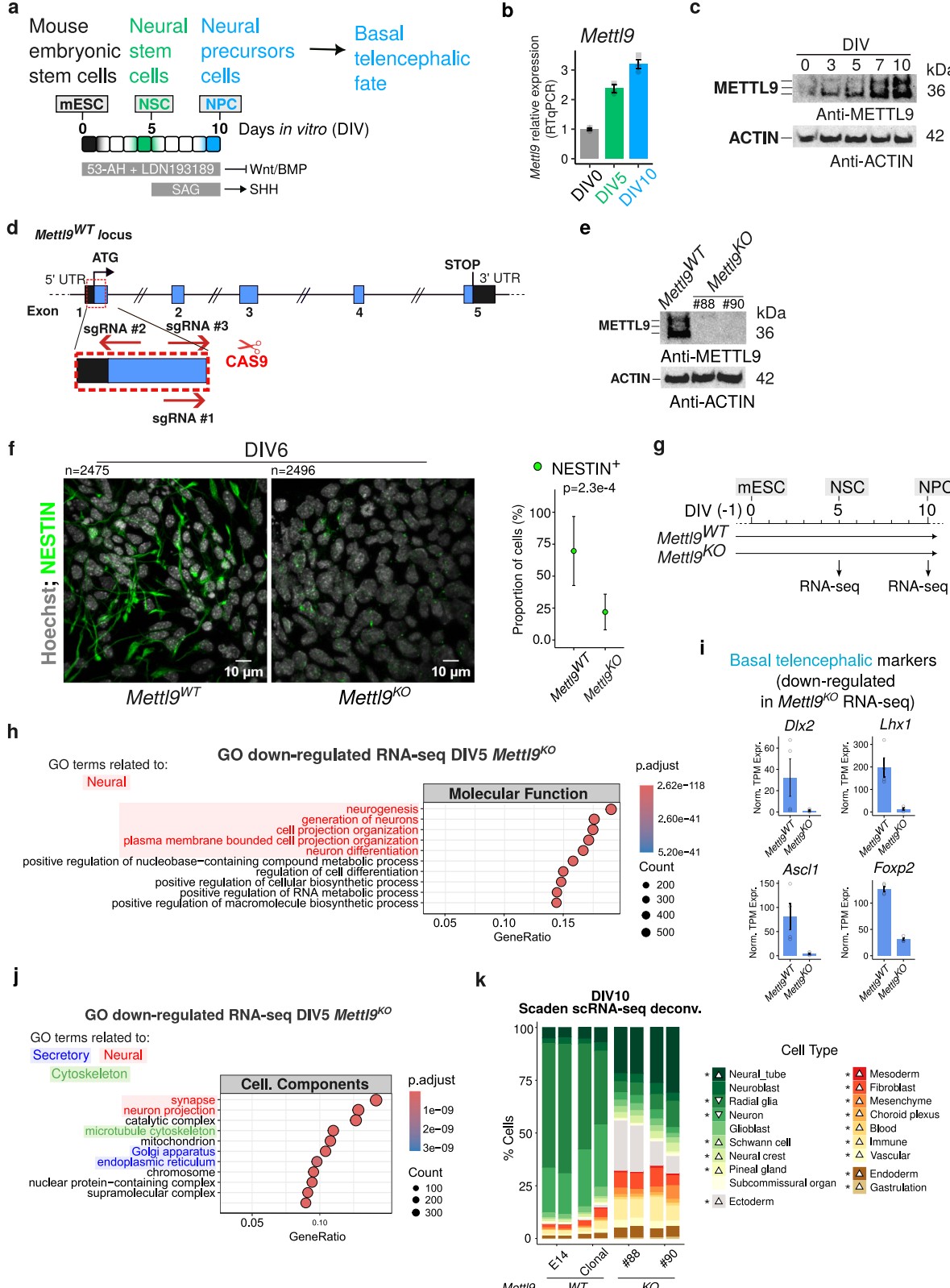

## Acute depletion of METTL9 from mESCs via Degron only partially mimics METTL9 constitutive depletion

To shed light on the biological and molecular processes more directly controlled by METTL9 during neural differentiation, we employed an inducible protein-Degron system[38,39]. We fused METTL9 to the mutated prolyl isomerase FKBP12[F36V], to induce its rapid degradation in live cells (Fig. 3a)[38,39]. Using CRISPR/Cas9, we generated an endogenously

tagged *Mettl9*[FKBP12-F36V-FLAG] mESCs line (in short *Mettl9*[Deg], which stands for *Mettl9 Degron*), by targeting the last exon of the *Mettl9* locus (Fig. 3b and Supplementary Fig. 5a) before the STOP codon. The resulting METTL9-FKBP12[F36V]-FLAG (METTL9-DEG) protein was expressed at the expected size in mESCs, as shown by WB (Fig. 3c). Upon cellular uptake, the dTAG[V]-1 drug specifically recognises the mutated FKBP12[F36V] (and not the endogenous, ubiquitous FKBP12) and

**Fig. 2 | Constitutive *Mettl9* depletion impairs mESCs neural priming and differentiation. a** Neural differential protocol adopted in this study. Shading indicates the acquisition of mESCs/NSC/NPC identity. **b** *Mettl9* mRNA expression normalised on *β-actin* (by qPCR), at DIV0, DIV5, DIV10. Error bars are mean ± SD; N = 3. **c** METTL9 protein expression shown by Western Blot (WB) from *Mettl9^WT^* mESCs DIV0 to DIV10 (anti-METTL9 antibody; kDa: kDalton; N = 1). **d** Mouse *Mettl9* locus and strategy to generate *Mettl9^KO^* mESCs via CRISPR/Cas9 sgRNAs targeting Exon1 (red). **e** WB from NSC extracts (DIV7) showing endogenous METTL9 expression with an anti-METTL9 antibody (WT; KO: #88 and #90). Blot representative of N = 3, from 3 differentiation experiments. **f** Representative immunofluorescence (IF) images of *Mettl9^WT^* and *Mettl9^KO^* NSCs (DIV6) with an anti-NESTIN antibody (green) and Hoechst (grey). Scale bar is 10 µm; relative quantification of NESTIN+ cells on the right (t-test, two-sided); error bars are mean ± SD. Quantified cell numbers (n) are shown above each panel (10 fields of view per condition; N = 2 differentiation experiments). **g** Timepoints of *Mettl9^KO^* mESC neural differentiation (DIV5 and DIV10) analysed by RNA-seq. **h** Top (10) GO Molecular function terms down-regulated in *Mettl9^KO^* RNA-seq (DIV5). In red, neural-related terms. Hypergeometric test: colour scale shows adjusted p values (Benjamini-Hochberg (BH) correction). **i** Normalised transcripts per million (TPM) expression of selected basal telencephalic markers. Error bars represent mean ± SE of N = 4. **j** Top (10) GO Cellular Component terms down-regulated in *Mettl9^KO^* RNA-seq (DIV5; see Methods). Hypergeometric test: colour scale shows adjusted p values (BH correction). **k** Cell type composition in control WT (E14 or clonal) and *Mettl9^KO^* (#88 and #90) lines inferred by SCADEN deconvolution analysis of scRNA-seq data (DIV10). Asterisk (*) is p = 0.029 (Wilcoxon test; N = 4).

recruits the E3 ubiquitin ligase CRBN to FKBP^F36V^, which in turns ubiquitylates METTL9-FKBP^F36V^ thereby promoting proteasomal degradation of the fusion protein (Fig. 3a).

Importantly, within 1 hour from dTAG^V^-1 administration, most of METTL9-FKBP-FLAG was degraded in mESCs, as shown by WB (Fig. 3d).

Therefore, we first assessed whether induced METTL9 protein depletion negatively impacted neural cell fate specification. To this end, we continuously supplied the dTAG^V^-1 drug, or DMSO for control, from DIV(−1) up until DIV5 (Fig. 3e) and then analysed the transcriptome of treated homozygous *Mettl9^Deg^* NSCs, by RNA-seq. At DIV5, we found 63 genes down-regulated and 126 up-regulated (Supplementary Fig. 5b; Supplementary Data 1) in the dTAG^V^-1-treated NSCs compared to the DMSO controls. GO analysis of the differentially regulated genes revealed that in the dTAG^V^-1-treated NSCs "tube development", "neurogenesis", "nervous system development" and "generation of neurons" were among the most down-regulated Molecular Function terms in comparison with DMSO controls (Fig. 3f; Supplementary Data 2), as exemplified by the neural markers *Celf2*, *Elavl2*, *Nrcam*, *Jag1* (Fig. 3g; Supplementary Data 1) as well as *Fzd1*, *Map2*, *Robo2*, *Zic1* and *Dll1* (Supplementary Fig. 5c). Moreover, among the most down-regulated GO terms concerning Cellular Component we found "neuron projection", "synapse", "plasma membrane region", "somatodendritic compartment", "dendrite" and "dendritic tree" (Fig. 3h; Supplementary Data 2), which were all down-regulated also in the *Mettl9^KO^* NSCs. Furthermore, among the up-regulated Molecular Functions (GO terms) we found many metabolic and biosynthetic processes (Supplementary Fig. 5d; Supplementary Data 2), similarly to *Mettl9^KO^* NSCs, and the up-regulated Cellular Component terms included "synapses" and "post-synapses" (Supplementary Fig. 5e). Therefore, the mis-regulation (mainly down-regulation) of neural-related genes as well as the up-regulation of metabolic-related genes in the *Mettl9^KO^* NSCs at DIV5 very likely represented bona fide specific effects of METTL9 depletion. Despite the similarity in the biological processes involved, the effects in the *Mettl9^Deg^* were very mild when compared with the *Mettl9^KO^*-transcriptomic analysis, both in terms of number of affected genes and in the strength of their mis-regulation. Consistent with this, no major macroscopic defects were observed upon neuralisation, as shown by a comparable number of Nestin positive cells between DMSO- or dTAG^V^-1-treated *Mettl9^Deg^* NSCs (Supplementary Fig. 5f).

The low levels of residual METTL9 observed in the dTAG^V^-1-treated *Mettl9^Deg^* mESC by WB (see Fig. 3d), as well as the transient presence of the protein from the time of its translation to its degradation, might be sufficient to sustain most of its biological functions in neural cells, resulting in the milder phenotypical and transcriptomic changes of dTAG^V^-1-treated *Mettl9^Deg^* lines compared to the *Mettl9^KO^*. Since the best characterised molecular activity of METTL9 is the catalysis of 1MH modification on target proteins containing the H[ANGST]H motif, we measured bulk 1MH levels in the dTAG^V^-1 treated cells (DIV6) and compared them with the paired control (DMSO) as well as with the *Mettl9^KO^* cells. Interestingly, while 1MH was efficiently reduced in the

KO cells, we detected no difference in the 1MH levels between the dTAG^V^-1-treated and DMSO (Ctrl) cells (Fig. 3i) and also the METTL9-independent 3MH levels, used as a control, were not affected (Supplementary Fig. 5g). Overall, these data suggest that the low levels of residual METTL9 protein in the dTAG^V^-1-treated *Mettl9^Deg^* line might be sufficient to sustain neural commitment of mESCs either via METTL9-dependent catalytic functions and/or through other non-catalytic activities.

## METTL9 supports neural commitment of mESCs largely independently of its catalytic activity

To investigate the contribution of METTL9 catalytic activity to its overall biological function, we abolished it by using CRISPR/Cas9. To this end, we targeted the endogenous *Mettl9* locus (Exon 3) in mESCs and edited 4 nucleotides to generate substitutions in 2 highly conserved amino acids within the SAM binding domain of the METTL9 protein (D151K and G153R) (Fig. 4a), which are known to abrogate METTL9 catalytic activity[14]. We selected homozygous *Mettl9^CatD^* ("CatD", standing for Catalytically Dead) and clonal *Mettl9^WT^* mESC lines (Supplementary Fig. 6a) and investigated their ability to sustain neural commitment and differentiation.

Surprisingly, *Mettl9^CatD^* NSCs did not display any major macroscopical defect, as shown by comparable NESTIN expression of both *Mettl9^WT^* and *Mettl9^CatD^* lines (Supplementary Fig. 6b), similarly to *Mettl9^Deg^* cells (Supplementary Fig. 5f). To understand whether *Mettl9^CatD^* NSCs exhibited any alteration at the molecular level, we characterised *Mettl9^CatD^* transcriptome by RNA-seq: at DIV5, we found only 54 genes mis-regulated compared with the clonal *Mettl9^WT^* line (29 down-regulated and 25 up-regulated) (Supplementary Fig. 6c; Supplementary Data 1). To exclude that the absence of an appreciable neural phenotype and the mild transcriptomic alterations were due to METTL9 residual catalytic activity in *Mettl9^CatD^* NSCs, we analysed bulk 1MH levels in *Mettl9^CatD^* at DIV6. *Mettl9^CatD^* NSCs displayed a significant reduction in 1MH levels (Fig. 4b), comparable to the 1MH decrease observed in *Mettl9^KO^* NSCs (Fig. 3i), despite having normal levels of the METTL9-independent 3MH modification (Supplementary Fig. 6d). This reduction is in agreement with previous data on METTL9-CatD in HEK293T cells[14] and indicates that *Mettl9^CatD^* mESCs express a bona fide catalytically dead METTL9 protein.

The mis-regulated genes found in the *Mettl9^CatD^* lines represent a much smaller proportion of those previously found in the *Mettl9^KO^* and also smaller than the dTAG^V^-1-treated *Mettl9^Deg^* lines at the same developmental stage (DIV5) (Fig. 4c). Overall, considering also 1MH levels in the 3 cell lines, these data indicate a lack of correlation between the molecular phenotype and the METTL9 catalytic activity (as evaluated by the reduction in 1MH levels). Conversely, METTL9 levels themselves (regardless of the presence of a WT or CatD protein) correlate with the severity of the phenotype; indeed, *Mettl9^KO^* lines which do not produce any METTL9 protein display a massively altered transcriptome; dTAG^V^-1-treated *Mettl9^Deg^* have residual METTL9 protein with normal 1MH levels and relatively mild transcriptomic

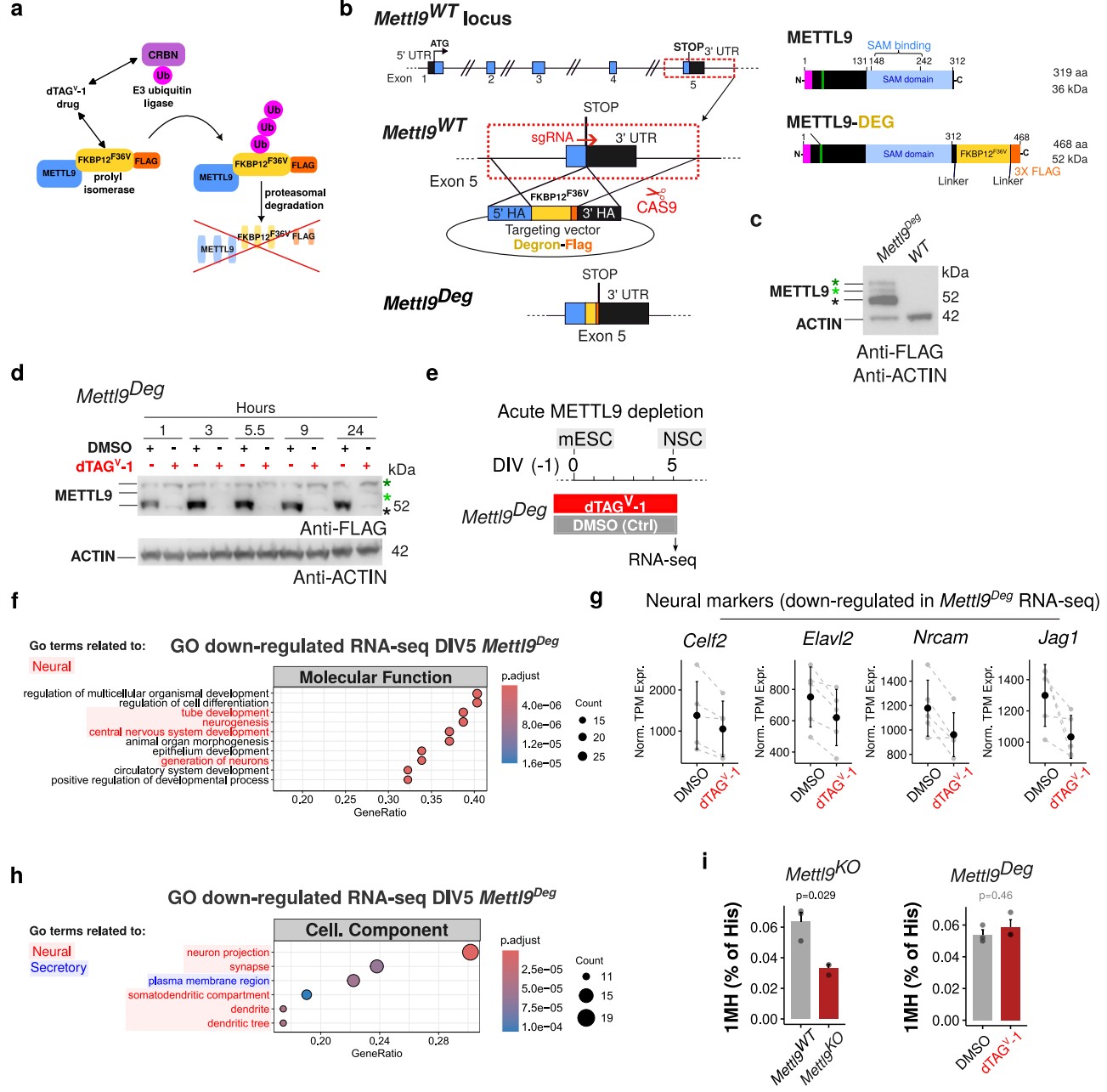

**Fig. 3 | Acute depletion of METTL9-DEGRON affects the gene expression profile of mNSCs. a** Schematic depicting METTL9-DEGRON system and relative mechanism of degradation upon dTAG$^V$-1 supply. **b** *Mettl9* genomic locus and its genetic targeting via CRISPR/Cas9 for *Mettl9$^{Deg}$* generation. On the right, the resulting C-terminally tagged METTL9-DEG protein is shown (aa: amino acid; kDa: kDalton). **c** Validation of METTL-DEGRON protein expression in *Mettl9$^{Degron}$* mESCs by WB (anti-FLAG and anti-ACTIN antibodies). Black, green and dark green asterisks (*) are the lowest, intermediate and top METTL9 bands, respectively. N = 3 WB (and differentiation) experiments. **d** Time course expression analysis of METTL9-DEG by WB, after supplying dTAG$^V$-1 (or DMSO, Ctrl) to mESCs for 1, 3, 5.5, 9 or 24 hours. Similar timepoints and drug concentrations used in N > 3 experiments. **e** Schematic

of experimental strategy for acute METTL9-DEG depletion (by dTAG$^V$-1) in *Mettl9$^{Deg}$* mESCs and molecular analysis at DIV5. **f** Top 10 Molecular function GO terms down-regulated in *Mettl9$^{Deg}$* RNA-seq (DIV5). Hypergeometric test: colour scale shows adjusted p values (Benjamini-Hochberg (BH) correction). **g** Normalised TPM expression of neural marker genes, from *Mettl9$^{Deg}$* RNA-seq (DIV5). Error bars represent mean ± SD of N = 5. **h** Top down-regulated Cellular Component GO terms in *Mettl9$^{Deg}$*. Hypergeometric test: colour scale shows adjusted p values (BH correction). **i** Relative bulk 1MH levels (% of total histidine) in *Mettl9$^{WT}$* and *Mettl9$^{KO}$* (left); DMSO- and dTAG$^V$-1-treated *Mettl9$^{Deg}$* (right), NSCs (DIV6), quantified by mass spectrometry. P values of the t-test (two-sided) are shown in black (P < 0.05) and grey (P ≥ 0.05), respectively; error bars show mean + SE, N = 3.

changes; whereas *Mettl9$^{CatD}$* have a mutant METTL9 protein without catalytic activity, and very mild transcriptomic alterations.

We next analysed the long-term consequences of METTL9 depletion or loss of catalytic activity by performing RNA-seq at DIV10 on *Mettl9$^{Deg}$* (DMSO-, for Ctrl, and dTAG$^V$-1-treated) and clonal *Mettl9$^{WT}$* and *Mettl9$^{CatD}$* NPCs (Supplementary Fig. 6e,f; Supplementary Data 1).

Consistent with our observations at DIV5, both dTAG$^V$-1-treated *Mettl9$^{Deg}$* and *Mettl9$^{CatD}$* cell lines displayed much milder transcriptomic alterations in terms of number and strength of mis-regulated genes compared to *Mettl9$^{KO}$* NPCs.

Interestingly, when comparing the number of mis-regulated genes between *Mettl9$^{CatD}$* and *Mettl9$^{WT}$* NPCs, we found a higher

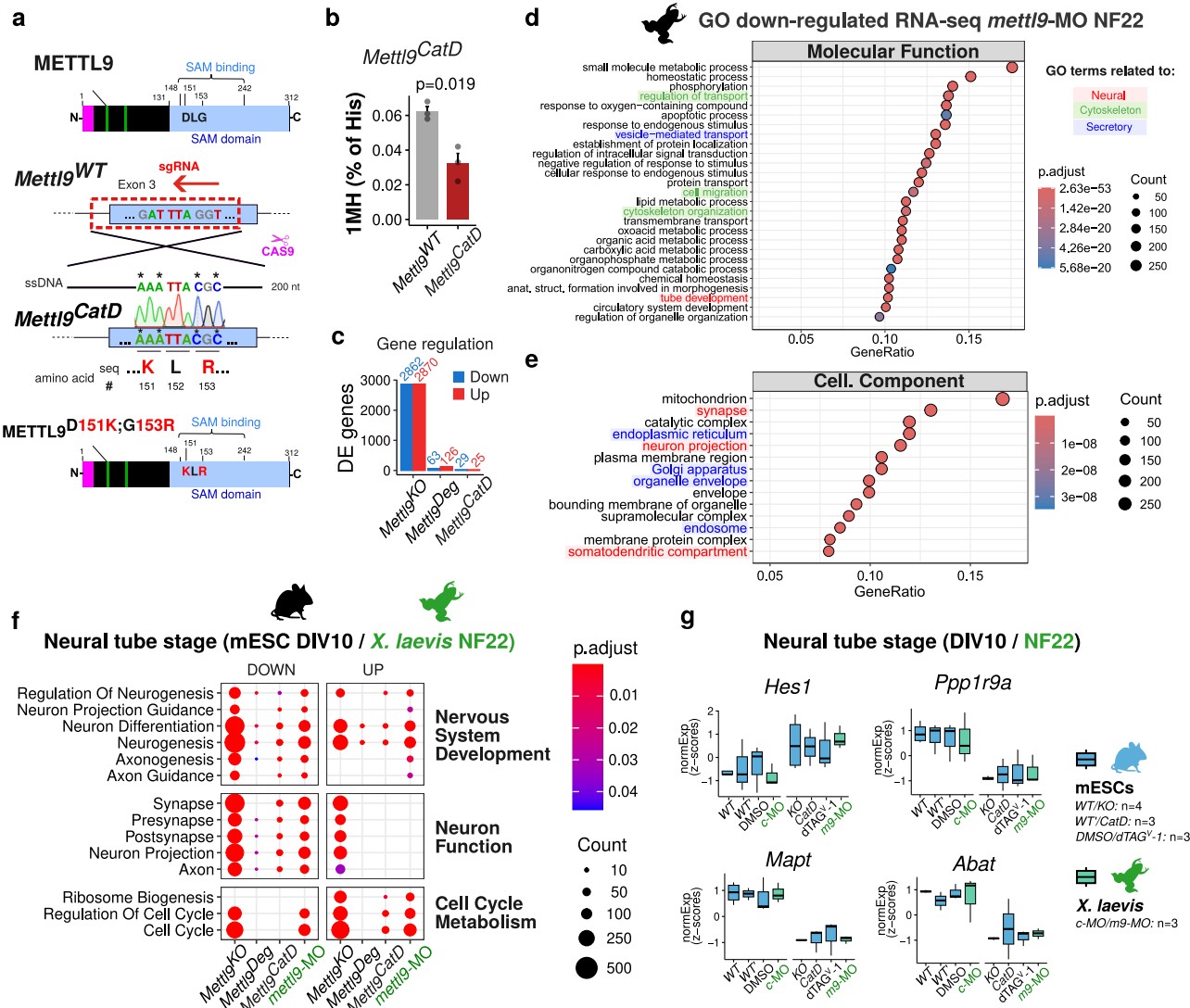

**Fig. 4 | METTL9 catalytic activity only mildly affects neural induction of mESCs.**
**a** METTL9 protein showing two key catalytic residues (D151, G153) of the SAM binding domain. Below, CRISPR/Cas9 targeting strategy to generate *Mettl9CatD* mESCs harbouring a mutated METTL9 protein with D151K and G153R. **b** Relative bulk 1MH levels (% of histidine) in *Mettl9CatD* and *Mettl9WT* NSCs (DIV6) quantified by mass spectrometry. P value of the t-test (two-sided) is shown; error bars indicate mean + SE, N = 3. **c** Number of differentially expressed genes found in the RNA-seq (DIV5) in *Mettl9KO*, *Mettl9Deg* and *Mettl9CatD* lines. **d** Top down-regulated Molecular function GO terms in *mettl9*-MO-NF22 *X. laevis* embryos. **e** Top down-regulated Cellular Component GO terms in *mettl9*-MO-NF22 embryos. **f** Neuronal and Cell

cycle/Metabolism-related GO terms commonly mis-regulated among *Mettl9* depleted NPCs (DIV10) and NF22 embryos. (**d**–**f**): hypergeometric test; colour scale shows adjusted p values (Benjamini-Hochberg correction). **g** Box plots showing examples of key neural marker genes consistently mis-regulated among the different datasets, included in the GO terms in panel (**f**). *WT'* is the clonal WT control for *Mettl9CatD*. *c*-MO and *m9*-MO are *ctrl*-MO and *mettl9*-MO respectively. Error bars show the mean ± SE of N = 4 (for *Mettl9KO* and Ctrl) and N = 3 (for *Mettl9CatD*, *Mettl9Deg* and *X. laevis* and respective Ctrls) experiments. Colour scale in (**d**–**f**) represents the Benjamini-Hochberg corrected p values from the hypergeometric test for enrichment of GO terms.

number of differentially expressed genes (DEG) at DIV10 compared to DIV5 (271 versus 54, respectively; Supplementary Fig. 6f, c). This indicates that the loss of METTL9 catalytic activity becomes more detrimental at later stages of neurogenesis, compared to the early phases of neural commitment. Conversely, dTAGV-1-treated *Mettl9Deg* NPCs (when compared to their paired DMSO-treated cells) displayed milder transcriptomic alterations at DIV10 (63; Supplementary Fig. 6e) than at DIV5 (189; Supplementary Fig. 5b), possibly due to compensatory mechanisms and/or differing molecular roles and interactions of the METTL9 protein during neural differentiation.

Given the striking neuralization defect of *Mettl9KO* mESCs at later stages of in vitro differentiation, we sought to complement and validate our observations in *Xenopus* embryo. Therefore, we profiled gene expression in whole *mettl9*-MO *Xenopus* tailbuds at stage NF22 (i.e.

after neural tube formation), a developmental timepoint roughly comparable to mouse NPCs at DIV10. Interestingly, we found 1617 down-regulated genes and 1454 up-regulated genes in the *mettl9*-MO embryos compared to *ctrl*-MO (Supplementary Fig. 7a). GO analysis revealed many Molecular Functions being down-regulated such as "tube development" as well as "vesicle-mediated transport", "protein transport", "cell migration" and "cytoskeleton organization" (Fig. 4d). Among the Cellular Component terms "synapses", "neuron projection", and "somatodendritic compartment" as well as "ER", "plasma membrane region", "Golgi" and "organelle envelope" were significantly down-regulated (Fig. 4e; Supplementary Data 2). Similarly to *Mettl9KO* NPCs, *mettl9*-MO embryos also showed up-regulation of many metabolic genes and interestingly, "central nervous system development" term (Supplementary Fig. 7b).

In conclusion, upon *mettl9* loss, key components linked to neural development are negatively affected at NF22 at a whole-embryo level.

Despite the evolutionary distance, we found many ontologies related to neurogenesis (Fig. 4f) and key neuronal genes consistently mis-regulated both across mouse RNA-seq datasets (*Mettl9^KO*, dTAG^V-1-treated *Mettl9^Deg* and *Mettl9^CatD*) at DIV10, and in NF22 *mettl9*-MO *Xenopus* embryos (Fig. 4g and Supplementary Fig. 7c). These data strongly support the notion that METTL9 is required throughout vertebrate neural development. However, the milder effects due to either the decreased bulk protein levels in *Mettl9^Deg* or the specific ablation of the enzymatic activity in *Mettl9^CatD* suggest that: (i) low levels of METTL9 are mostly sufficient to support neurogenesis and that (ii) this role occurs mainly through catalytic-independent functions.

## METTL9 modulates the secretory pathway in mouse NSCs

We next set out to investigate the molecular functions of METTL9, in NSCs. To this end, we first employed the *Mettl9^Deg* cell line. Although this system does not achieve a complete protein depletion, it enables rapid and inducible degradation of endogenous METTL9. With this approach we sought to characterise the immediate and direct consequences of METTL9 depletion, which might point at the specific biological roles exerted by METTL9 in NSCs. We characterised the proteome of NSCs by mass spectrometry after only 48 hrs of dTAG^V-1 treatment, at the onset of neural commitment (from DIV3 until DIV5) (Fig. 5a) and found 2 proteins significantly down-regulated and 11 up-regulated (as compared to DMSO-treated samples; adjusted p value < 0.05) (Fig. 5b and Supplementary Fig. 8a; Supplementary Data 3). These included the ER-to-Golgi and intra-Golgi trafficking protein USO1[40–42], the ER- and lipid-droplet- associated NSDHL enzyme involved in cholesterol biosynthesis[43], the plasma membrane-associated FXYD6 (sodium/potassium ATPase regulator)[44] and carboxypeptidase D, CPD[45], the ER-associated CYP51a1 involved in cholesterol biosynthesis[46,47] and the ER associated RDH11 retinol dehydrogenase[48], all associated with the endomembrane compartment of the secretory pathway. GO analysis on the mis-regulated proteome (encompassing a larger set of 115 up-regulated and 63 down-regulated proteins with a less stringent q value threshold of 0.2) revealed up-regulation of terms related to "ER membrane", "ER lumen" and "ER protein-containing complexes" (among which chaperones) and "ERGIC" (ER Golgi intermediate compartment) (Fig. 5c) as well as down-regulation of ribosomal-related genes (Supplementary Fig. 8b). We also found up-regulation of terms related to the "Golgi stack" and "Golgi membrane" and "Golgi associated vesicles", as well as "COPI-coated vesicles" and "exocytic", "transport" and "secretory vesicles" terms (including synaptic ones) (Fig. 5c). In summary, the proteomic analysis of NSCs after acute METTL9 loss reveals the up-regulation of the endomembrane/secretory pathway as an early molecular consequence.

To investigate whether similar biological processes were still affected after a complete and long-term METTL9 depletion, we characterised the proteome of *Mettl9^KO* NSCs by mass spectrometry. Interestingly, we found 395 proteins significantly down-regulated and 329 up-regulated over the Ctrl *Mettl9^WT* cells (q value < 0.01) (Supplementary Fig. 8c; Supplementary Data 3), highlighting a stronger proteome mis-regulation compared to that occurring in the dTAG^V-1 *Mettl9^Deg* line (34 proteins). This is consistent with a complete and prolonged absence of METTL9 during NSC differentiation. Interestingly, GO analysis of the mis-regulated proteins in the KO revealed down-regulation of terms related to "plasma membrane", "ER", "ER-Golgi intermediate compartment", as well as "nervous system development" among others (Supplementary Fig. 8d). These GO terms are consistent with those found at the transcriptomic level in the same *Mettl9^KO* NSCs (see Fig. 2j), and also with an impaired neuralisation. Among the up-regulated GO were found terms related to "mitochondrion", "cellular response to leukaemia inhibitory factor", "endosome

membrane". We then evaluated whether the proteins perturbed in the dTAG^V-1 *Mettl9^Deg* were similarly altered in the *Mettl9^KO*: interestingly we observed that the up-regulated proteins in the dTAG^V-1 were significantly down-regulated in the *Mettl9^KO*, and the down-regulated proteins in the dTAG^V-1 were significantly up-regulated in the *Mettl9^KO* (Supplementary Fig. 8e). These data indicated that the secretory pathway-related proteins are affected in both systems, although they are regulated in opposite directions and to a different extent (number of proteins and strength of the regulation). Therefore, the alteration of the secretory pathway could be an early cellular event directly linked to METTL9 acute loss, which is exacerbated in the long-term (*Mettl9^KO*), where complex indirect and/or compensatory cellular mechanisms might also be put in place by NSCs.

## METTL9 localises to the Golgi in mouse NSCs

METTL9 contains a predicted N-terminal signal peptide (SP)[49,50] corresponding to the first 18 amino acids (Supplementary Fig. 8f). Such peptide sequences target proteins to the secretory pathway via the ER[51]. Western blotting of both the endogenous wild-type METTL9 (Fig. 2c) and METTL9-DEG (Fig. 3c) proteins highlighted the existence of multiple METTL9 bands, suggesting the presence of post-translational modifications in both mNSCs and mESCs. METTL9 contains a canonical NMTS glycosylation sequon (amino acid # 35-38)[52] (Fig. 5d, green) with the Asn (N35) as a possible glycosylation site[53]. In silico prediction analysis[54], revealed N86 as an additional putative N-glycosylation site (Fig. 5d), although with lower probability than N35. We thus sought to determine whether the shifted METTL9 bands observed by SDS-PAGE are due to protein glycosylation in mESCs. We treated protein extracts with different glycosidases[55] in vitro and found that the 2 highest METTL9 bands completely disappeared upon N-glycosidase (Endo H or PNGase F) treatment, but not in the control or in O-glycosidase-treated samples (Fig. 5e). Given the sensitivity to Endo H[55], these data suggest that the two highest METTL9 bands corresponded to two high-mannose (or non-complex hybrid) N-glycans.

To directly prove this, we generated two mutagenised constructs encoding for a METTL9-FLAG coding sequence harbouring either one amino acid substitution (N35Q) or two (N35; N86Q), which prevent N-glycosylation (Fig. 5f and Methods), and transfected them in mESCs. Despite the presence of 3 bands in the control, in the N35Q mutant the highest METTL9 band was lost, whereas in the double mutant (N35Q;N86Q) only the lowest METTL9 band was observed (Fig. 5f). This confirms that both N35 and N86 are N-glycosylated in mESCs. In addition, we generated a fourth construct to alter the tripartite regions[56,57] of METTL9 signal peptide (SP*-METTL9-FLAG: R2A;W7A;-C9A;S11A; see Methods). Although no N-glycosylation site was edited in this construct, it showed only the lowest, unmodified METTL9 band by WB, similarly to the double mutant (N35Q;N86Q) (Fig. 5f and Supplementary Fig. 8g). This result suggests that disruption of the SP prevents the N-glycosylation of both Asn residues (N35 and N86), most likely by precluding METTL9 translocation into the ER.

These data were further corroborated through the inhibition of the first biosynthetic step of N-linked glycosylation in the ER by supplying tunicamycin[58–60] to mESCs: protein extracts were analysed by WB, which revealed the complete depletion of the two highest METTL9 bands in tunicamycin-treated samples (Supplementary Fig. 8h).

Overall, our experiments suggest that METTL9 N-terminal SP directs the protein to the ER lumen, where it acquires high mannose N-glycans on two distinct Asn residues (N35;N86). Indeed, this is also supported by available proteomic data[61], where METTL9 was found to co-fractionate mainly with the ER and other compartments of the secretory pathway in 5 different human cell lines (Supplementary Fig. 8i). Furthermore, since METTL9 lacks typical ER retention signals, such as KDEL[62], it likely exits the ER and proceeds further along the secretory pathway, through the Golgi.

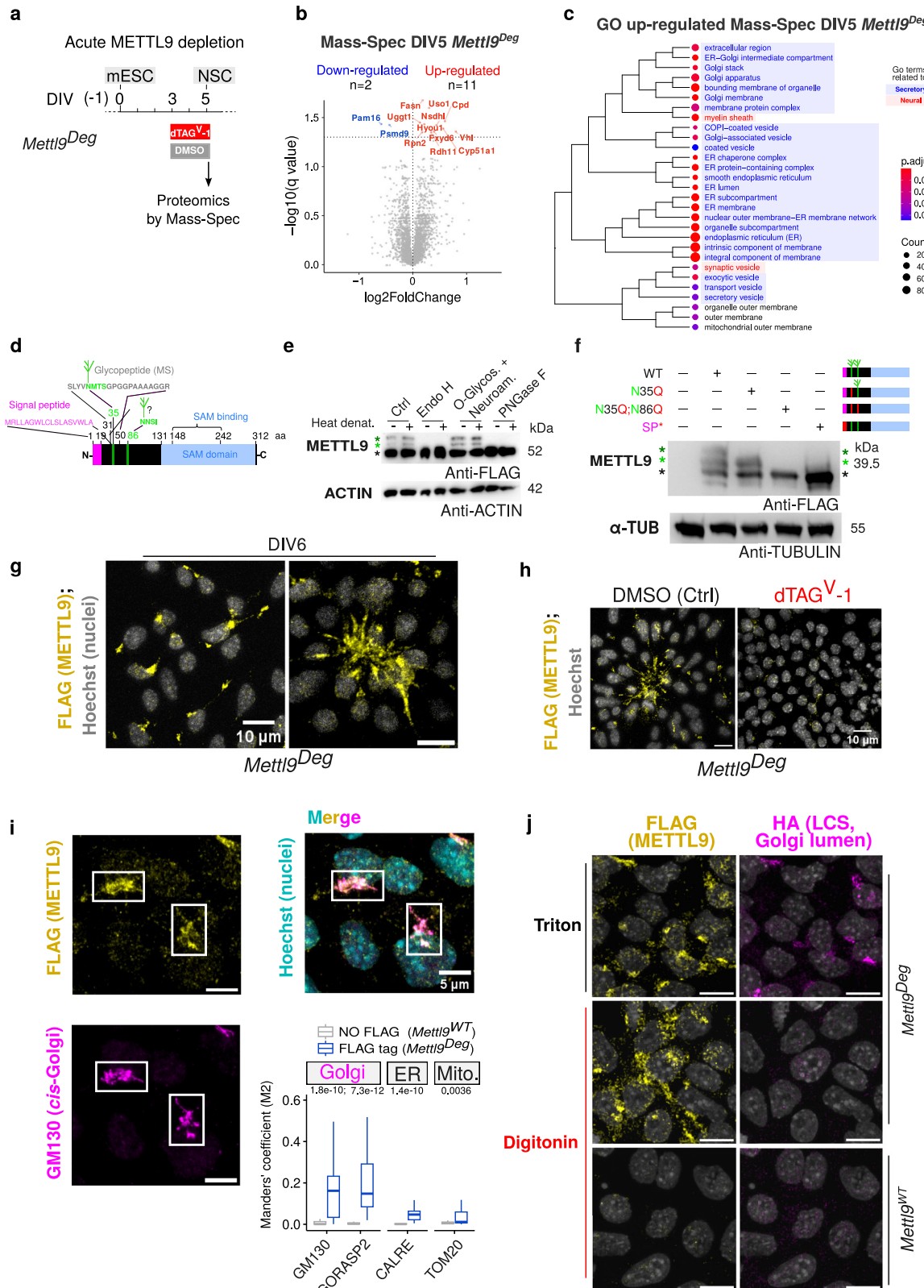

To further validate METTL9 association with the secretory pathway, we determined its endogenous sub-cellular localisation. Due to the lack of suitable antibodies, we took advantage of the FLAG tag of the *Mettl9Deg* cell line and performed immunofluorescence staining of NSCs: this revealed that almost all cells expressed METTL9 (Fig. 5g, left panel), indicating that METTL9 is homogenously expressed in NSCs.

Moreover, METTL9 displayed a distinct, asymmetrical cytosolic distribution (Fig. 5g), particularly evident in the apical part of neural rosettes (Fig. 5g, right panel); this signal was greatly reduced upon dTAGV-1 treatment (Fig. 5h), highlighting the specificity of the FLAG antibody in recognising the METTL9 protein. Since METTL9 distribution was reminiscent of the Golgi apparatus, which is asymmetrically localised in neural cells[63–66] and in particular within neural rosettes[67,68],

**Fig. 5 | METTL9 is associated with the secretory pathway and co-localises with the peripheral Golgi. a** Experimental strategy for acute METTL9 depletion in *Mettl9^Deg* mESCs. **b** Volcano plot of mis-regulated proteins (coloured dots and labels) in dTAG^V-1-*Mettl9^Deg* NSCs over Ctrl (DMSO-treated), by mass spectrometry (q value < 0.05). Y axis indicates log10 FDR-adjusted p values; two-tailed moderated t-statistics. **c** GO terms of the up-regulated proteins (with a q value < 0.2). Clustering showing significantly enriched GO terms with q value < 0.05. Hypergeometric test; colour scale shows adjusted p values (Benjamini-Hochberg correction). **d** METTL9 protein: signal peptide (magenta), the glycopeptide position[53] (grey) within sequon (green) and predicted sequon at N86 (aa is amino acid). **e** WB showing METTL9-DEG in N- (Endo H and PNGase) or O-glycosidase-treated mESCs extracts. Black, green and dark green asterisks (*) are the lowest, intermediate and top METTL9 bands, respectively. Antibodies: anti-FLAG and anti-ACTIN; N = 2 WB and enzymatic treatments. **f** WB (anti-FLAG, anti-alpha-TUBULIN) showing WT METTL9-FLAG or mutated METTL9-FLAG: N35Q, N35Q;N86Q, and SP* (mutated signal peptide), expressed in WT ESCs. N-glycosylated residues (green); mutated amino acid residues (red); SP (magenta). Asterisks near blot as in (**f**). N = 3 independent experiments. **g** IF images of *Mettl9^Deg* NSCs: anti-FLAG (METTL9) and Hoechst (nuclei). Right: a neural rosette. Scale bar: 10 μm. N = 8 fields of view. **h** IF images of DMSO- (Ctrl) or dTAG^V-1-treated *Mettl9^Deg* NSCs: anti-FLAG antibody showing METTL9. Scale bar: 10 μm. N ≥ 6 fields of view per condition. **i** IF images of *Mettl9^Deg* NSCs: anti-FLAG antibody (yellow) for METTL9, GM130 (magenta) for *cis*-Golgi and Hoechst (cyan) for nuclei. Scale bar: 5 μm. On the right, co-localisation between anti-FLAG and: (i) anti-GM130, (ii) anti-GORASP2 (i and ii for Golgi), (iii) anti-CALRE (ER) or (iv) anti-TOM20 (mitochondria). Boxplot: Manders' coefficient (M2) indicates co-localization (slices analysed in NO FLAG and FLAG tag, respectively: N = 32, N = 31 (CALRE); N = 35, N = 40, (GM130); N = 35, N = 31 (GORASP2); N = 30, N = 30 (TOM20). P values: above each marker; Wilcoxon test, two-sided. **j** IF images of *Mettl9^Deg*/LCS-HA or WT E14 NSCs (DIV5), after Triton or digitonin permeabilization: anti-FLAG antibody (METTL9, yellow), anti-HA (LCS, *trans*-Golgi lumen, magenta) and Hoechst (nuclei, gray). Scale bar: 10 μm. IFs in (**g–j**) were representative of N ≥ 3 differentiation experiments.

we assessed whether METTL9 co-localised with it. Indeed, co-staining of NSCs with the *cis*-Golgi markers GM130[69] and GORASP2[70] confirmed an extensive overlap between METTL9 and the Golgi (Fig. 5i and Supplementary Fig. 8j) in DIV6 cells, whereas a very low proportion of METTL9 signal was found to co-localise with other cytosolic organelle markers such as CALRE for the ER[71] or TOM20 for mitochondria[72] (Fig. 5i and Supplementary Fig. 8j). Consistently, METTL9 subcellular distribution drastically changed in DIV6 NSCs after an hour treatment with golgicide (Supplementary Fig. 8k), a drug which specifically induces Golgi fragmentation[73]. Overall, these data indicate that METTL9 is a Golgi-associated protein in mouse NSCs, whose subcellular localisation depends on the integrity of the Golgi.

Eventually, we assessed whether METTL9 localisation at the Golgi occurred on its peripheral (cytosolic) or luminal side (or both). To test this, we performed immunofluorescence on NSCs after selective plasma membranes permeabilization with digitonin, which leaves cholesterol-poor membranes, such as that of Golgi, intact[74,75] and thus not accessible to antibodies. A tagged-version of the *trans*-Golgi Lactosylceramide synthase (LCS-HA) was used as a Golgi luminal control (see Methods). As expected, the LCS-HA signal was detected only in the Triton-NSC samples (control), but not in the digitonin one (Fig. 5j); on the contrary, the FLAG signal of the endogenously tagged METTL9 was detected in the digitonin-permeabilised sample and comparable to the Triton-permeabilised, indicating that METTL9 is positioned on the cytosolic face of the Golgi.

Overall, this data showed a tight association of METTL9 with the secretory pathway since i) it is N-glycosylated in this compartment, ii) it localises to the peripheral (cytosolic) Golgi side and iii) it modulates the secretory pathway-related proteome. This suggests that METTL9 could exert molecular functions linked to the homeostasis of this cellular compartment.

## METTL9 interacts with key secretory pathway and transport regulators, independently of its catalytic activity

To investigate the molecular pathways regulated by METTL9, we characterised the METTL9 interactome in NSCs (DIV4), by performing immunoprecipitation (Supplementary Fig. 9a) coupled to mass spectrometry (IP-MS; Supplementary Fig. 9b). This identified 71 proteins enriched in the Anti-FLAG-METTL9 IP (Fig. 6a; Supplementary Data 3). Among the top interactors were the microtubule destabiliser Stathmin1, (STMN1), which is highly expressed in the nervous system also during development[76–78] and the E3-ufmylation adapter DDRGK-domain containing protein 1, DDRGK1, anchored to the cytosolic side of the ER[79] and involved in reticulophagy[80]. Gene Ontology (GO) analysis (Fig. 6b) revealed that METTL9 interactome is enriched in Golgi membrane, ER protein-containing complexes, synaptic vesicles, transport vesicles and lysosomal and endosomal membrane factors.

Importantly, besides other know METTL9 interactors like CANX and FAF2, we found many RAB proteins, including the pre-Golgi and cis-Golgi RAB1A and RAB2A (both involved in pre-Golgi trafficking and whose knock-down or over-expression cause Golgi fragmentation[81–84]), and the late-endosome and lysosomal GTPase RAB7[85,86]. We validated METTL9-STMN1 and METTL9-RAB2a physical interactions after co-expressing METTL9-FLAG and either STMN1-HA or HA-RAB2a in mESCs. METTL9 was co-immunoprecipitated by STMN1-HA or HA-RAB2a, as shown by WB (Fig. 6c, d).

To further characterise the METTL9-STMN1 interaction, we predicted the structure of this complex in silico by using AlphaFold[87] (Fig. 6e). Interestingly, the METTL9 amino acid residues mainly involved in the binding coincide with its catalytic pocket (Fig. 6e). It is noteworthy that STMN1 amino acid sequence, and in particular the residues predicted to participate in METTL9 binding contain no bona fide 1MH motif and thus are unlikely to be methylated. Overall, these data suggest that METTL9 might engage STMN1 in a catalytic-independent interaction.

We tested these predictions by measuring METTL9 methyl-transferase activity (MTase) in a biochemical assay (Supplementary Fig. 9c,d), in the presence of S-adenosylmethionine (SAM) and the SLC30A5$_{163-180}$ synthetic peptide, which is a known METTL9 target[14]. When using purified recombinant GST-STMN1 (GST-STMN1-HA) as a substrate instead of SLC30A5$_{163-180}$, GST-METTL9 MTase activity was almost undetectable, and comparable to the negative GST control (Fig. 6f). This strongly supports the notion that STMN1 is not a METTL9 substrate in vitro. We next assessed METTL9 MTase activity on the canonical SLC30A5$_{163-180}$ target peptide, in the presence of increasing concentrations of STMN1. Remarkably, we observed that METTL9 activity was impaired by STMN1 (but not by GST) in a dose-dependent manner (Fig. 6f). These data indicate that STMN1 is able to outcompete a known METTL9 substrate in vitro and are in agreement with our structural models showing STMN1 in the catalytic pocket of METTL9.

AlphaFold predictions also showed that METTL9 embraces a few amino acids of the STMN1-Tubulin binding repeat I, including Ser63 (Fig. 6e), whose phosphorylation regulates tubulin binding[88–90]. Therefore, METTL9 binding may modulate STMN1 function, similarly to the way in which STMN1 regulates METTL9 catalytic activity. A comparable mechanistic model might be extended to other members of the METTL9 interactome, like RAB2A and RAB7A (Supplementary Fig. 9e, f), which share with STMN1 i) a predicted binding to the METTL9 catalytic pocket, ii) the lack of H[ANGST]H motifs and iii) the presence of regulatory domains (i.e. switch II GTPase domain) contacting METTL9. Thus, similarly to STMN1, also the activity of these RAB proteins could be potentially regulated upon METTL9 binding.

Interestingly, the substitutions D151K and G153R within METTL9-CatD are predicted to have a minimal impact on the overall METTL9

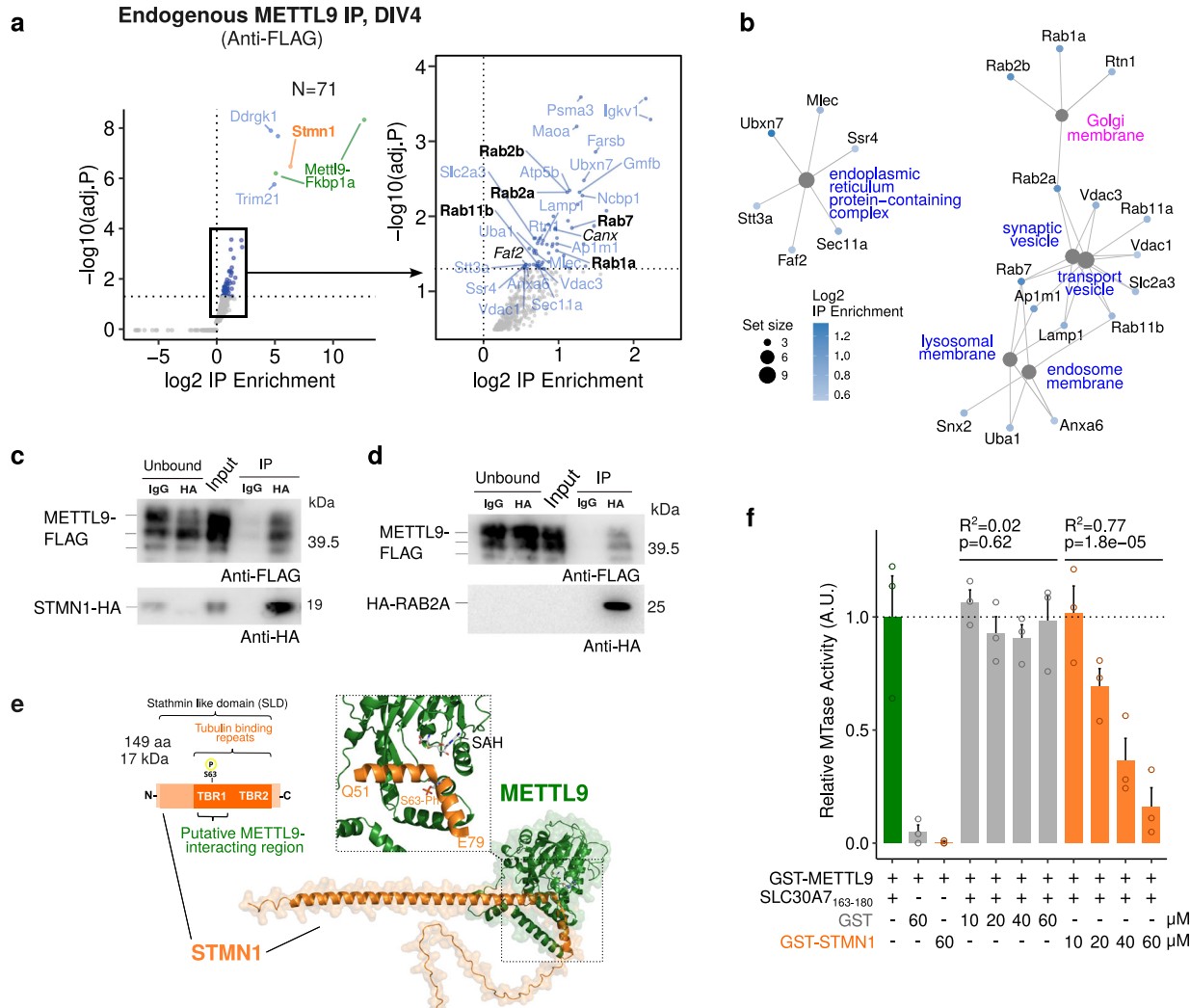

**Fig. 6 | METTL9 interacts with secretory pathway-related proteins in mNSCs.**
**a** Volcano plot showing the proteins enriched upon METTL-IP (anti-FLAG) over the control-IP (IgG) in *Mettl9^Deg* NSCs (q value < 0.01); on the right, zoom-in on the enriched interactors, among which the known METTL9-interactors *Faf2* and *Canx* (in black, italics). Y axis represents the -log10 p value after Benjamini-Hochberg multiple test correction (one-sided moderated t-statistics; P[X > x]). **b** Network showing the genes belonging to the top GO terms enriched in METTL9 interactors relative to the secretory pathway. **c, d** WB showing the immunoprecipitation of STMN1-HA (IP) (**c**) or HA-RAB2a (**d**) with anti-HA beads (HA) or IgG (ctrl), after co-expression of STMN1-HA or HA-RAB2a and METTL9-FLAG in WT mESCs. WB signal: anti-FLAG and anti-HA. N = 2 co-IP (and differentiation) experiments, run in N = 4 WB. **e** AlphaFold modelling prediction of STMN1-METTL9 protein complexes (STMN1, in orange; METTL9 in green). **f** Bar graph showing in vitro METTL9 methyltransferase activity (MTase) of recombinant GST-METTL9-FLAG (GST-METTL9) protein with the SLC30A7$_{163-180}$ peptide or with GST-STMN1-HA (GST-STMN1) and GST (first, second and third bar, respectively); GST-METTL9 activity was also measured with increasing concentrations (μM) of GST or GST-STMN1. The correlation between MTase activity and concentration of GST or GST-STMN1 is expressed by the R$^2$ and p values (linear regression). Error bars represent mean + SE of N = 3 independent experiments. A.U. is arbitrary unit.

protein structure (Supplementary Fig. 9g,h). The mutated residues are in close proximity to the most buried and solvent-inaccessible end of the catalytic site, where the SAM molecule is accommodated. Notably, they are positioned far from the bulk of the predicted interaction surface between METTL9 and STMN1 or RAB2A (Supplementary Fig. 9i), suggesting that METTL9 could preserve these protein-protein interactions regardless of its catalytic activity. Consistent with these models, METTL9-CatD-FLAG could be immunoprecipitated by STMN1-HA or HA-RAB2a, after an Anti-HA-IP in mESCs, as shown by WB (Supplementary Fig. 9j,k, respectively).

Overall, the fact that the protein-protein interactions between the catalytically inactive METTL9 and STMN1 or RAB2a are likely preserved in *Mettl9^CatD* mESCs might explain why they show a mild molecular phenotype and highlights the importance of these physical interactions in NSCs.

## The catalytic-dependent function of METTL9 has a secondary but convergent role in neural development and secretory system function

METTL9 catalytic activity exerts a secondary role compared to the main non-catalytic functions in NSCs, as shown by the less severe neural differentiation defects of *Mettl9^CatD* compared to *Mettl9^KO*. In agreement with this, there is a striking difference between *Mettl9^KO* and *Mettl9^CatD* or *Mettl9^Deg* in the number and strength of mis-regulated genes. However, the three cell lines also share a high overall level of consistency among their transcriptomic alterations. Indeed, cumulative plots of the Fold Changes (Supplementary Fig. 10a) showed that many hundreds of the most up- (or down-) regulated genes in the *Mettl9^KO* were consistently up- (or down-) regulated, respectively, in the other lines, although to a milder extent. In addition, integrative re-analysis of all three mESC experiments at DIV5 confirmed

"neurogenesis", "neuron/cell projections", "synapses", "somatodendritic" compartment and "Golgi" terms among the top gene ontologies coherently down-regulated in all the cellular models (Supplementary Fig. 10b,c). Moreover, we found that many genes encoding for Golgi-resident enzymes or structural Golgi-proteins, transport- and secretory pathway-related proteins (also involved in neural processes) were consistently mis-regulated, albeit to different extents, among all mouse RNA-seq datasets (Mettl9$^{KO}$, Mettl9$^{Deg}$, Mettl9$^{CatD}$) at DIV5 (Supplementary Fig. 10d).

Therefore, notwithstanding the less severe neural phenotype of the Mettl9$^{CatD}$, the high degree of similarity of the affected cellular pathways among the different cell lines suggests that 1MH-dependent and independent activities might converge onto the same molecular processes in NSCs. A prerequisite for this model would be that METTL9 substrates of methylation might be directly involved in secretory-related and neuronal pathways. To test this hypothesis, we scanned the sequences of the mouse proteome and identified potential METTL9 substrates (i.e. all H[ANGST]H-containing proteins). Interestingly, GO analysis revealed that secretory, neuronal, and transport-related processes accounted for almost 50% of the categories significantly enriched in putative 1MH targets (Supplementary Fig. 10e). Moreover, some of these proteins have already been demonstrated to be methylated by METTL9[14] (Supplementary Fig. 10f). These included MYO18A, known to exert important roles in Golgi positioning in neural cells[65] and many zinc transporters (SLC30A1/5/7 and SLC39A7), which are critical for controlling zinc levels within the cell, and some also within organelles such as the Golgi compartment[91,92]. Therefore, while METTL9 has potentially hundreds of targets expressed in any cell type, their 1MH-modification (and hence regulation) might be more critical for neural stem cells, which heavily rely on the secretory pathway to sustain directional trafficking towards the apical part of the cell, where the growth cone emerges[93,94].

Overall, these data suggest that, while the enzymatic and catalytic-independent functions of METTL9 act through distinct molecular mechanisms (1MH-methylation of protein substrates and protein-protein interactions, respectively) and that their relative contribution greatly varies in NSCs, both functions might impinge on the same cellular machinery of the secretory system, sustaining proper neural development.

## METTL9 depletion affects cellular trafficking kinetics and disrupts Golgi integrity in mNSCs

We showed that the expression level of hundreds of early secretory pathway proteins was altered in Mettl9$^{KO}$ NSCs (Supplementary Fig. 8c,d). A similar (although to a much smaller extent) effect was observed upon METTL9 acute depletion (Fig. 5b,c); this suggested that the homeostasis of the secretory pathway could be directly controlled by METTL9. Moreover, many METTL9 interactors control key cellular processes such as macromolecular motility and cargo engagement; for instance, STMN1 controls microtubules growth[76,77] and cellular trafficking along the secretory system also relies on this cytoskeleton component[95]. RAB2a is a key regulator of ER to Golgi trafficking, and its GTPase domain is potentially modulated by METTL9 binding (Supplementary Fig. 9e, f).

Thus, we investigated the ER to Golgi trafficking kinetics of a Golgi resident enzyme, the α-mannosidase II (ManII), by using the retention using selective hooks (RUSH) method[96], in Mettl9$^{KO}$ NSCs (Fig. 7a). This state-of-the-art system consists of two components, a hook and a reporter: at the steady state, the hook, which is an ER-localised Streptavidin protein (Str-KDEL), anchors the cargo, ManII, fused to the streptavidin-binding peptide-EGFP (ManII-SBP-EGFP) to the ER (donor compartment), via the strong Str-SBP interaction. Upon biotin supply to live cells (Time 0, $T_0$), the ManII-SBP-EGFP is displaced from the ER-Str hook, enabling its synchronous release from the ER to the Golgi (final, acceptor compartment).

We engineered stable Mettl9$^{WT}$ and Mettl9$^{KO}$ cell lines expressing both the ER hook and the ManII-SBP-EGFP cargo and performed a time course live imaging experiment at NSCs stage. At $T_0$, EGFP signal was mainly localised at the ER (green ring around nuclei) in both Mettl9$^{WT}$ and Mettl9$^{KO}$ cells (Fig. 7b); however, 5 and 10 mins after Biotin addition the EGFP signal was significantly more retained in the ER for the Mettl9$^{KO}$ compared to the Mettl9$^{WT}$ cells, which had already started to export ManII-SBP-EGFP to the ER exit sites (i.e. more signal as dots and granules compared to the Mettl9$^{KO}$). This defect was fully recovered after 20 minutes (endpoint), as most EGFP signal was detected in the Golgi (dots) both in the Mettl9$^{WT}$ and Mettl9$^{KO}$ cells (Fig. 7b and Supplementary Fig. 11a–c). Overall, this data indicates a slight but significant delay in the kinetics of the cellular trafficking from the ER to the Golgi in Mettl9$^{KO}$ NSCs; this is consistent with the altered proteomic data in the same cells and could be explained by the disruption of the regulatory functions normally exerted by METTL9 on STMN1 and RAB2 activities.

Besides regulating cellular trafficking, METTL9 co-localises with the Golgi (Fig. 5i) and many of its protein interactors, particularly, STMN1 and RAB2, are essential for maintaining the structural integrity of the Golgi[82,97]. Thus, we investigated whether the absence of METTL9 could have a detrimental effect on the Golgi apparatus morphology. By performing immunofluorescence staining of Mettl9$^{KO}$ cells with an anti-GM130 antibody, we classified Golgi morphology in three categories[98]: compact (i.e. with a dense and/or elongated shape typical of neural stem cells), mildly fragmented and completely scattered (Fig. 7c), and we enumerated cells according to these 3 categories. Interestingly, we observed that Mettl9$^{KO}$ NSCs displayed a significantly higher proportion of scattered and mildly fragmented Golgi compared to Mettl9$^{WT}$ cells at DIV6 (Fig. 7d).

It is noteworthy that both dTAG$^V$-1-treated Mettl9$^{Deg}$ and Mettl9$^{CatD}$ NSCs also showed a significant, albeit milder Golgi fragmentation pattern (Supplementary Fig. 11d), further supporting the convergence of METTL9 catalytic-dependent and independent functions on the maintenance of the secretory system.

Golgi integrity and positioning presides cell polarization, axon elongation and intracellular trafficking, which are pivotal processes in the induction and maturation of a neuronal cell[63,93,94,99]. In light of this, our data suggest that, upon METTL9 depletion, Golgi morphology and cellular trafficking are negatively affected, probably due to the lack of METTL9 regulatory activity on important interactors like STMN1 and RAB2; these cellular defects, in turn, might prevent neural differentiation of mESCs.

## The catalytic independent roles of METTL9 in the secretory pathway and neural development are evolutionary conserved

To enable the molecular dissection of amphibian NSCs and get insight into the conservation of the molecular pathways affected by mettl9 loss, we took advantage of Xenopus animal caps (a.c.) neuralised with noggin mRNA. Thus, we assessed the effect of mettl9 knock-down on early neural induction by analysing the proteome and transcriptome of a.c. at stage 12.5 (Fig. 8a; Supplementary Fig. 12a). Strikingly, proteomic analysis by mass spectrometry revealed the "Golgi" GO term amongst the most down-regulated biological processes (Fig. 8b; Supplementary Fig. 12b; Supplementary Data 3). Furthermore, RNA-seq analysis of the same samples (Supplementary Fig. 12c; Supplementary Data 1) identified amongst the most down-regulated GO terms "projection organization", as well as "cell motility" and "vesicle-mediated transport" (Fig. 8c; Supplementary Data 2). Moreover, among the down-regulated cell components, we found the "ER", "cytoskeleton" and "Golgi" (Fig. 8d; Supplementary Data 2). Interestingly, among the most up-regulated Molecular Function terms we found "generation of neurons", "neurogenesis" and "neuron differentiation" (Supplementary Fig. 12d). These data strongly suggest that Mettl9 exerts neurodevelopmental

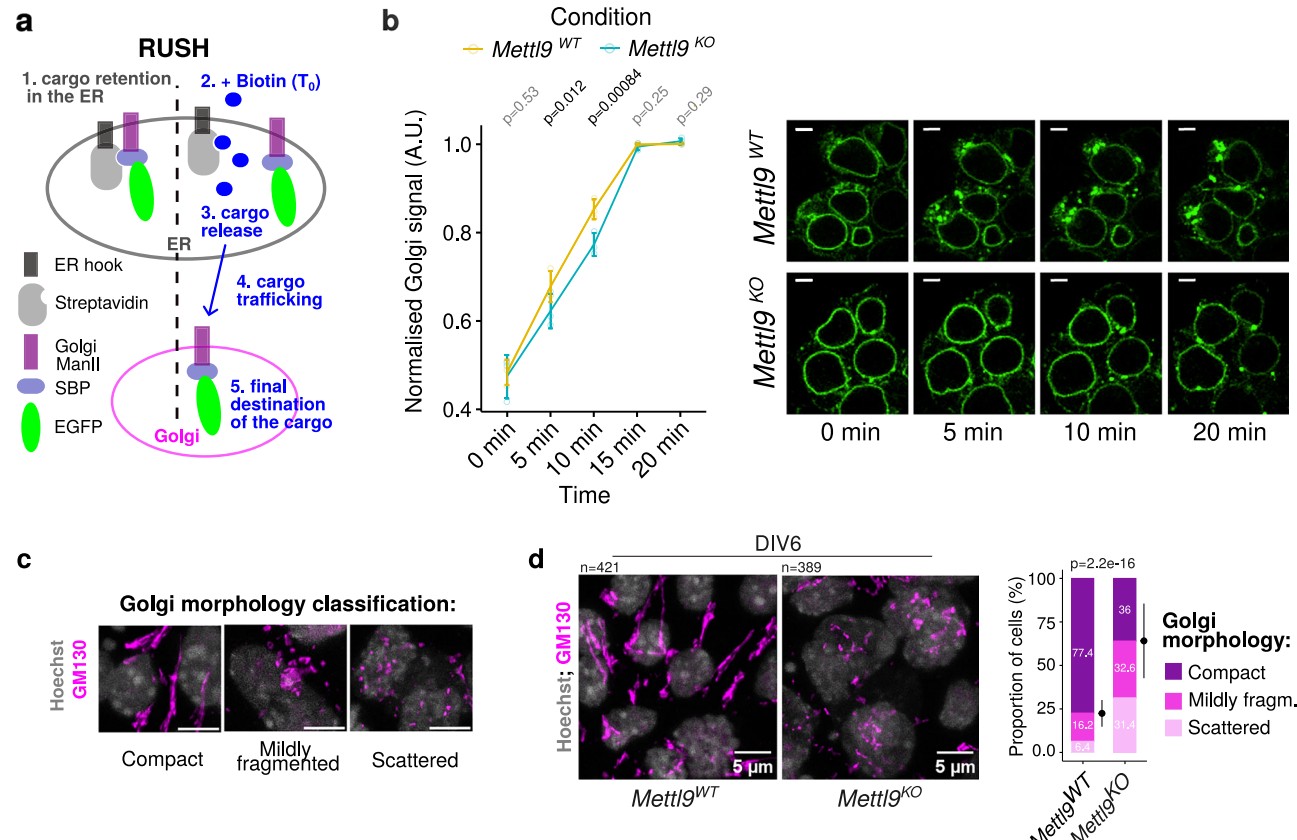

**Fig. 7 | METTL9 depletion impairs cellular trafficking kinetics and Golgi morphology in mNSCs. a** Schematic showing the RUSH system used to study cellular trafficking in this work. The ER hook here is the Streptavidin protein anchored to the Endoplasmic Reticulum (ER). The reporter includes the streptavidin-binding peptide (SBP) fused to the cargo which is the α-mannosidase II (ManII) enzyme resident in the Golgi and to the fluorescent EGFP protein. Upon Biotin addition to cell media, $(T_0)$, the reporter is released from the ER hook and its export from the ER starts. Its trafficking is followed until it reaches the Golgi, where ManII is delivered to. **b** Normalised ManII-SBP-EGFP signal in the Golgi of *Mettl9^{WT}* and *Mettl9^{KO}* NSCs, at different timepoints after Biotin addition (paired t-test, two-sided, p values are shown above each time point); error bars represent mean ± SE of N = 3 independent experiments. On the right, representative close-ups of live microscopy images of MannII-SBP-eGFP signal used for quantification. Scale bar is 5 μm. **c** Qualitative classification of Golgi (anti-GM130) morphology in mNSCs into 3 categories, used for cell counting. Scale bar is 5 μm. **d** Representative IF images of *Mettl9^{WT}* and *Mettl9^{KO}* NSCs stained with anti-GM130 (Golgi) and Hoechst (nuclei). Scale bar is 5 μm. On the right, corresponding quantifications of Golgi morphology (categorised in compact, mildly fragmented or scattered). Error bars represent mean ± SD; number of cells counted are above each panel. (χ2 test). IF performed from N > 3 differentiation experiments.

functions by modulating similar pathways (i.e. secretory and cytoskeleton) throughout vertebrates. Given the high level of conservation between *Xenopus* and mNSCs (both in terms of neural phenotype and cellular pathways regulated by METTL9, e.g. Golgi), we leveraged *Xenopus* embryos to evaluate the contribution of METTL9 catalytic activity to neural development in vivo. To this end, we co-injected a *Xenopus laevis mettl9^{CatD}* mRNA (harbouring the same mutations of mouse *Mettl9^{CatD}*) (Supplementary Fig. 12e) with *mettl9*-MO in embryos and assessed the potential recovery of the neural phenotype by analysing *elcR* mRNA expression by WISH. Remarkably, *mettl9^{CatD}* injected embryos could partially rescue the neural defects ascribed to *mettl9* knock-down. Indeed, a significantly lower proportion of both *mettl9^{WT}* and *mettl9^{CatD}* embryos showed a perturbed neuralization pattern when compared to the *mettl9*-MO injected embryos alone, rescuing *elcR* expression in the intermediate (i) stria, as well as in the trigeminal placode (tp) (Fig. 8e, f). Importantly, the extent of the recovery in the *mettl9^{CatD}* was comparable to that of *mettl9^{WT}*, suggesting that *mettl9^{WT}* or *mettl9^{CatD}* can similarly rescue the neural phenotype in *Xenopus*.

Overall, these complementation assays confirmed the direct involvement of Mettl9 in neural development in vivo and corroborated our previous data about the primary role of Mettl9 catalytic-independent functions in mNSCs.

## Discussion

In this work, we identified a developmental role for Mettl9 in the context of neural differentiation. By exploring human and mouse scRNA datasets and using 2 different model systems (namely *X. laevis* embryos and mouse neural stem cells cultures), we showed that Mettl9 is highly expressed during vertebrate neurogenesis. Importantly, we discovered that neural fate specification is consistently impaired by perturbing Mettl9 expression or function with different genetic systems, resulting in aberrant developmental trajectories. Overall, our data demonstrate that Mettl9 requirement for early neurogenesis is a conserved feature of vertebrates.

Our study highlights that this neural function is likely mediated by direct involvement of METTL9 in the secretory pathway: indeed ER-, vesicle- and Golgi-related pathways are mis-regulated upon METTL9 depletion, as shown by proteomic and transcriptomic data, with the latter indicating possible regulatory feedback loops[100–102]. These molecular perturbations are consistent with the altered cellular trafficking kinetics, observed by RUSH, in mNSCs.

Importantly, we found that METTL9 co-localises with the Golgi apparatus in mNSC. We suspect that this might have been overlooked by other works because of a lack of good anti-METTL9 antibodies, as well as the exclusive use of METTL9 over-expression systems in cancer cell lines.

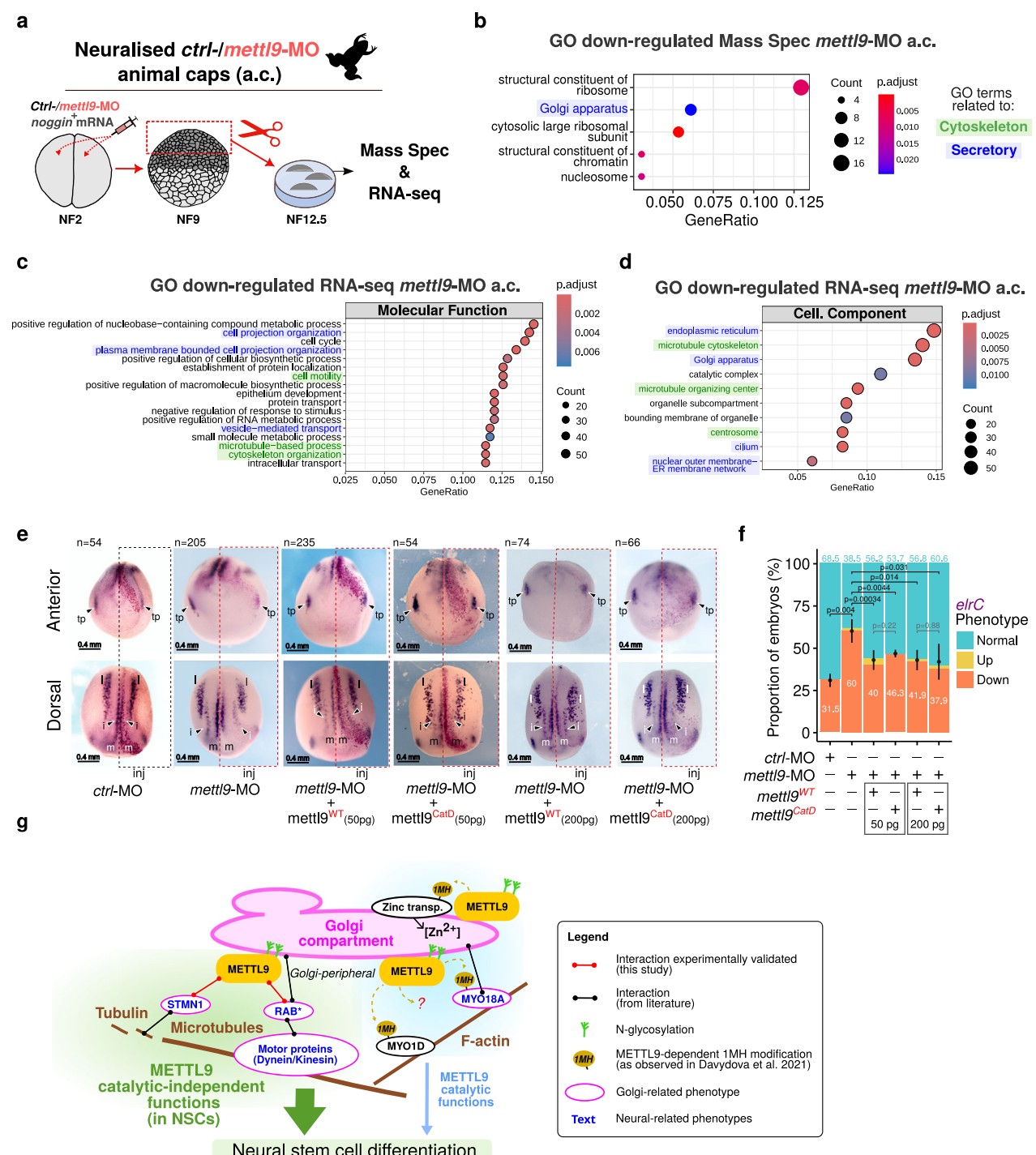

METTL9 association with the Golgi is per se very intriguing, since this organelle underlies multiple aspects of neural fate specification and maturation. Golgi is asymmetrically distributed in neural cells[63–66] and its position determines neural cell polarity and axonal growth[63]. Indeed, Golgi polarisation underlies asymmetric membrane trafficking, which is key for dendrite and axon growth[93,94,99,103]. Golgi is also present in the form of outposts within dendrites, where it sustains dendrite growth[99,104]. Therefore, Golgi fragmentation in METTL9-deficient NSCs might have broad consequences on neuronal function, such as disruption of polarised trafficking which in turns could prevent axon growth.

The close association of METTL9 with the secretory pathway is also supported by biochemical data. Indeed, we showed that METTL9, besides harbouring a signal peptide (SP), is modified with high mannose N-glycans in two Asn residues of its amino acid sequence. Thus, METTL9 enters the secretory system through the ER where it is N-glycosylated. Our mass spectrometry data in mNSCs highlighted many METTL9 interactors known to be associated with the peripheral (cytoplasmic) side of Golgi as well as with endosomes and vesicles. This is consistent with METTL9 being detected by IF on the cytosolic side of the Golgi. Interestingly, the METTL9 SP might escape cleavage and could be retained, as a buried helix, within the mature METTL9 protein (Supplementary Fig. 8f). Thus, its SP might act as an anchor to tether METTL9 to the Golgi membrane (similarly to other proteins[105–107]). Alternatively, this localization pattern might be accomplished by specific protein interactors. While our data clearly

**Fig. 8 | Conserved alteration of the secretory pathway in *mettl9*-MO *X. laevis* neuralised animal caps (a.c.). a** Schematic depicting the preparation of neuralised animal caps (a.c.) from *mettl9*-MO or *ctrl*-MO embryos in *X. laevis*. **b** Most down-regulated GO terms in the *mettl*-MO a.c. proteome. **c, d** Most down-regulated Molecular Function (**c**) and Cellular Component (**d**) GO terms in *mettl9*-MO neuralised a.c. RNA-seq data (see Methods). GO terms related to Cytoskeleton and Secretory pathways were highlighted in green and blue, respectively. (**b–d**): hypergeometric test; colour scale shows adjusted p values, Benjamini-Hochberg correction. **e** Representative anterior and dorsal views of *X. laevis* embryos injected with either *ctrl*-MO or *mettl9*-MO, or co-injected with *mettl9*-MO and either *mettl9$^{WT}$* or *mettl9$^{CatD}$* mRNAs, showing *elrC* mRNA expression (purple), by whole-mount RNA in situ hybridisation. Arrowheads indicate intermediate (i) neuron precursors and trigeminal placodes (tp) affected in the treated side of *mettl9*-MO injected embryos. Inj.: Injected side. (m) medial and (l) lateral stria. Embryos numbers are shown above each panel. **f** Bar graph showing the quantification of embryos: *ctrl*-MO, *mettl9*-MO or *mettl9*-MO co-injected with either *mettl9$^{WT}$* or *mettl9$^{CatD}$* mRNAs,

screened for altered *elrC* expression (% embryos are reported in the graph; $\chi 2$ test). Error bars: mean ± SD. **g** Schematic depicting our working model: METTL9 has an evolutionary conserved role in vertebrates in sustaining early neural development, mainly through catalytic independent functions (in green). Among these, we identified one related to the maintenance of the secretory pathway. This function is mediated by protein-protein interactions occurring most likely at the peripheral side of the Golgi (magenta), where METTL9 is localised in mNSCs. We envisage that METTL9 binding to STMN1 and RAB2 regulates their functions related to the cytoskeleton, cargo motility and Golgi structure. Therefore, in METTL9-deficient NSCs, cellular trafficking and Golgi morphology are perturbed, and this is detrimental to the establishment of neural polarity, cell signalling, axon development and ultimately to neural differentiation. METTL9 methyltransferase activity may cooperate with the maintenance of the secretory pathway through histidine methylation (and thus, probable regulation) of Golgi-related substrates, like MYO18A or zinc transporters like SLC30A5/7 and SLC39A7 but might be marginal for neural development.

indicate that METTL9 co-localises with the Golgi apparatus, further work is required to dissect the biological significance of METTL9 N-glycosylation on its catalytic and non-catalytic functions, particularly in the context of neural development.

The focus of METTL9 studies so far has been on its methyltransferase activity[14,18]. The 1MH profiles by Davydova et al.[14] document particularly high levels of METTL9 activity in the brain. Here, we described that proteins involved in the secretory pathway and Golgi homeostasis are enriched in METTL9 substrates, either experimentally validated[14] or predicted by the presence of the canonical H[ANGST]H motif. Many zinc transporters are targets of METTL9-dependent methylation (including SLC39A7, SLC30A1/5/7), and the zinc-binding affinity of SLC39A7 is directly regulated by 1MH[14]. These zinc-transporters are embedded into the membrane of the Golgi *cisternae* (except for the plasma membrane-associated SLC30A1), where they regulate zinc transport and concentration[91,108,109]. This function, in turn, is crucial to support the activity of numerous Zn-dependent enzymes of the secretory pathway (e.g. α-mannosidase II and β−4-galactosyltransferase[92,110–112]). The unconventional myosin protein MYO18A is another experimentally validated METTL9 target[14]: it exerts important roles in bridging the Golgi to the actin cytoskeleton via GOLPH3, by shaping Golgi morphology and positioning as well as affecting NSC polarity and apically-targeted membrane trafficking during mouse corticogenesis in vivo[65,113–116]. These insights have been highlighted in our study also thanks to the pivotal role of the secretory pathway in highly polarised cells, such as developing neurons.

Notwithstanding the above, we found that METTL9 exerts a significant role independently of its catalytic activity. By comparing different cellular models, we showed that ablation of METTL9 catalytic activity caused much milder neural commitment defects than full genetic abrogation of *Mettl9*. The METTL9 interactors identified in mNSCs do not contain any H[ANGST]H motifs and may form inactive complexes with the catalytic pocket of METTL9. This was confirmed by our in vitro methylation assays on STMN1. Taken together, these data suggest that the adequate METTL9 protein levels which preserve these protein-protein interactions might be more important than its catalytic activity, at least during early neural development. This is further documented by the observation that in *Xenopus* embryos a mutant *mettl9* can rescue the neuralization phenotype with an efficiency similar to wild-type *mettl9*.

Moonlighting is a fascinating, emerging aspect of molecular biology. The concept is that individual proteins can possess more than one physiologically significant role. For example, in addition to their long-known and well-characterised structural role, histone octamers have also retained an evolutionarily ancient catalytic function devoted to the reduction of copper ions[117]. Moreover, several nuclear modifying enzymes also possess non-catalytic functions[118]. So far few but intriguing works have reported the existence of non-enzymatic roles for

METTL proteins. For instance, N-terminus of METTL3 is sufficient to promote translation by mRNA circularization[6,119] and METTL11A/METTL13 function as regulatory subunits of each other's catalytic activity[120]. Recently, METTL1 was found to exert an oncogenic role in sarcomagenesis independently of its tRNA methyltransferase activity, via modulation of tRNA aminoacylation and protein synthesis[121]. Thus, we hypothesise that having both catalytic dependent and independent functions could represent a more generalised feature of METTL proteins; this could have arisen from the convergent evolution of different molecular mechanisms impinging on similar biological processes, such as the secretory pathway in the case of METTL9. Thus, our findings support the investigation of additional functions of RNA-modifying enzymes in physiologically relevant contexts.

Our protein interactome study in mNSCs showed that METTL9 binds to STMN1 and many RABs proteins, amongst the top-ranking hits. STMN1 belongs to the Stathmin family of tubulin-binding proteins and is highly expressed in the nervous system[76,77,122]. Perturbation of STMN1 impairs dendrite growth and axonal arborisation[123,124] and is involved in regulating synaptic plasticity, memory[125] and behaviour[126]; moreover, STMN1 levels are altered in neurodegenerative diseases and its gene is linked to a variety of psychiatric disorders[76,127]. Amongst the other METTL9 interactors, we found RAB1, RAB2, RAB7, and RAB11, which are all important for transport and secretory-related functions in neurons[128–131]; most of them are also associated with neurodegenerative or neurodevelopmental disorders[132,133]. Importantly, altered expression of STMN1, RAB1, and RAB2 has been shown to cause Golgi fragmentation[81–83,97], similarly to METTL9 loss of function. Neural induction, axonal elongation, cell projection growth, and the establishment of somatodendritic compartments heavily rely on Golgi functionality, and these processes were recurrently altered upon METTL9 down-regulation. Interestingly, alteration of Golgi morphology is often linked to impaired ER to Golgi trafficking[83,134,135], which might explain the mis-regulation of the ER protein levels in the proteome of METTL9-deficient NSCs as well as their defective trafficking.

Proteins travelling along the secretory pathway are modified by post-translational modifications, e.g. glycosylation, whose apposition requires the precise spatio-temporal control of specific subsets of enzymes within each sub-compartment (e.g. specific Golgi cisternae)[136]. Both an intact Golgi structure and functional cellular trafficking underpin this complex process[137]. Thus, the Golgi fragmentation and altered trafficking kinetics of Golgi enzymes like ManII that we observed upon METTL9 loss could result in an aberrantly modified proteome output (mis-glycosylation)[136,137], with a negative impact on the function of plasma membrane or secreted proteins, involved in cell-cell communication[138,139]. The delay in the ER to Golgi trafficking kinetics in *Mettl9$^{KO}$* NSCs could also cause transient protein accumulation in the ER with negative consequences on the ER homeostasis, at the level of protein misfolding and altered protein

modifications. Other parts of the secretory pathway as well as the retrograde trafficking, which are also key for recycling enzymes and sorting them correctly in each cisterna[136,137], could be affected in *Mettl9*[KO] NSCs. Future work will be required to assess in detail the impact of METTL9 loss on cellular trafficking by testing other RUSH cargos in METTL9-deficient mNSCs.

Our working model proposes that METTL9 sustains neural differentiation, mainly in a catalytic-independent manner, by safeguarding the secretory pathway. Mechanistically, this might occur through protein-protein interactions: upon binding to key secretory pathway regulators, like STMN1 and RABs, METTL9 might modulate them by competing with their partners and/or allosterically modifying their structure. METTL9 MTase activity has a smaller contribution to neural development, but amongst its substrates there are several Golgi-associated proteins. We showed that STMN1 outcompetes known METTL9 substrates in vitro and we know that STMN1 is particularly highly expressed in the developing brain[76,78]; thus, as neural development progresses in vivo, METTL9-STMN1 interaction could prevail over METTL9 MTase activity.

Stemming from these observations, further investigation will clarify the pleiotropic molecular roles of METTL9 in the secretory pathway, particularly during neural development.

Over 40% of the Golgi genes mutated in monogenic disorders, causing the so-called *Golgipathies*, affect the nervous system, a very high proportion compared to other tissues, and they often impair neurodevelopment[140]. Furthermore, Golgi fragmentation has been described as one of the first cellular alterations present in the neurons of patients at early stages of neurodegenerative conditions such as Parkinson's, Amyotrophic Lateral Sclerosis and Alzheimer's diseases, among others[141,142], although it is not clear the extent of its contribution to the development of these diseases.

Interestingly, METTL9 expression in the adult brain peaks in the striatum, which is responsible for some of the cognitive and behavioural functions impaired in neurodegenerative diseases.

Importantly, we have highlighted a putative link between heterozygous human *METTL9* deletions and the impairment of neurodevelopment or cognitive functions in 6 patients; nevertheless, further investigation is needed to directly assess the causal relationship between *METTL9* gene loss and the onset of these neurodevelopmental disorders' symptoms.

In conclusion, elucidating the relationship between METTL9 and Golgi function in mammalian neurogenesis will help understanding the cellular and molecular mechanisms underlying Golgi integrity and homeostasis; this, in turn, might be leveraged to amend Golgi fragmentation and dysfunction and represent an important step forward, towards the treatment of both neurodevelopmental and neurodegenerative diseases.

## Methods

Animals' use followed recommendations of the European Community (2010/63/UE) and protocol approved by Italian Ministry of Public Health (Auth. #501/2021-PR). Sex determination of the *Xenopus* embryos used in this study was not conducted due to the technical constraints associated with sex identification at early developmental stages; however, this limitation is not expected to impact the results, as the large sample sizes employed minimize potential sex-related variability.

The genetic sex composition of the 6 patients (described in the Decipher project[21]) was of 2 females (327478,276830) and 4 males (276011, 412447, 266291, 322729).

### Xenopus laevis maintenance

*Xenopus laevis* frogs were purchased from Nasco, Fort Atkinson, Wisconsin (USA) and kept in aquatic housing systems (Aquaneering) with continuously recirculating water (16-18 °C) passed through a fluidised bed biofilter and final UV light steriliser. Animals were fed on pelleted diet.

### *Xenopus laevis* and RNA in situ hybridization

Embryos were obtained and staged as previously described[143,144]. Whole-mount RNA in situ hybridization (WISH) was performed as described by Harland et al[145]., with the following modifications: pigment bleaching was performed following Mayor et al[146]. Digoxygenin-labeled antisense RNA probes were generated by in vitro transcription from pMA-*mettl9* (containing the *X. laevis mettl9.L* CDS, obtained by dsDNA synthesis and cloning; Thermo Fisher Scientific), or from *elrC, notch, neurog2,* and *twist1* templates[30–32]. β-gal staining was performed using Salmon-Gal as a chromogenic substrate (Sigma Aldrich, #96317). Briefly, embryos were transferred into glass tubes and fixed in 1X MEMFA solution (100 mM MOPS pH 7.4, 2 mM EGTA, 1 mM MgSO$_4$, 3.7% Formaldehyde) with horizontal stirring for 30 minutes, then washed twice in 1X PBS for 5 minutes each. 500 μl of β-galactosidase detection solution (PBS, 30 mM K$_3$Fe(CN)$_6$, 30 mM K$_4$Fe(CN)$_6$*3H2O, 2 mM MgCl$_2$, 400 ug/ml Salmon-Gal) was added to the embryos, and they were incubated at 37 °C with vertical stirring in the dark until the staining became evident (30-60 minutes). The reaction was then stopped with four washes in 1X PBS for 5 minutes each. Embryos were fixed again in 1X MEMFA for 30 minutes with horizontal stirring, and gradually dehydrated through sequential 5 minute washes in 50%, 70%, and 100% ethanol solutions. The embryos were stored in 100% ethanol at −20 °C.

*Xenopus* embryos at stage 30 were cut in half horizontally to expose the pharyngeal and branchial pouch region; half embryos were then processed for WISH using *mettl9* and *twist1* (expressed in neural crest cells[147]) probes. The Accession ID or references of the RNA probe sequences have been reported in the Supplementary Data File 4.

### Synthesis of capped mRNAs for *Xenopus* injection

The pCS2-*mettl9*[WT] plasmid was engineered by cloning the *Xenopus* mettl9.L CDS into the pCS2 plasmid by restriction digestion of the pMA-*mettl9* insert from the pMA-QR plasmid using EcoRI (New England Biolabs). For the generation of the mutant pCS2-*mettl9*[CatD] plasmid (encoding for D149K and G151R, corresponding to mammalian D151K and G153R), the pCS2-*mettl9*[WT] backbone together with a portion of the *mettl9*-wt gene was amplified by PCR from the pCS2-mettl9-wt plasmid (Phusion™ Plus DNA Polymerase with GC enhancer (ThermoFisher Scientific, #F630S) and gel purified. To introduce the four point-mutations in the *mettl9* gene, the DNA sequence encoding for D149K and G151R was synthesised as a double-stranded DNA fragment (Genewiz) and cloned into the above-mentioned PCR-amplified and linearised plasmid using In-Fusion cloning according to the manufacturer instructions (In-Fusion snap assembly master mix, Takara, #638948). The sequences of the final constructs were confirmed by Sanger sequencing. The DNA sequences of PCR and Sanger primers and dsDNA used in the *mettl9*[CatD] cloning are reported in Supplementary Data File 4.

Capped *beta-galactosidase* (β-gal), *noggin, mettl9*[WT] and *mettl9*[CatD] mRNAs were synthesised from linearised templates obtained by digesting plasmids pCS2-*nuc β-gal*, pCS2-*noggin* (kind gift of R. Harland), pCS2-*mettl9*[WT] and pCS2-*mettl9*[CatD] with NotI (New England Biolabs). The mMESSAGE mMACHINE™ SP6 Transcription Kit (ThermoFisher, #AM1340) was employed according to the manufacturer's instructions to generate capped mRNAs in vitro.

### Microinjections in *X. laevis* embryos

10-20 ng of morpholino antisense oligonucleotides (MOs, purchased from GeneTools, Inc) were injected into the animal region of one of the two dorsal blastomeres of 4-cell stage embryos. *mettl9*-MO targets the exon1/intron1 boundary and leads to mis-splicing by retaining the first intron, generating a predicted truncated protein made of the 53 initial

aa residues (encoded by the first exon) and 13 additional residues (dictated by the unspliced first intron before a premature stop codon). As a control MO (ctrl-MO) we used the same amount of the universal GeneTools control MO (ctrl-MO) that targets a human beta-globin intron mutation causing beta-thalassemia. For rescue experiments, 4-cell stage embryos were injected in the animal pole of one of the two dorsal blastomeres, with 20 ng of mettl9-MO or with: 20 ng of mettl9-MO + 50 (or 200) pg of mettl9$^{WT}$ or mettl9$^{CatD}$ mRNAs, encoding Mettl9$^{WT}$ or Mettl9$^{CatD}$ proteins, respectively. For control, 20 ng ctrl-MO was similarly injected. In all these injections, 400 pg of β-gal mRNA were co-injected as a tracer. For transcriptomic analysis, embryos were bilaterally injected in the animal pole of both dorsal blastomeres at the 4-cell stage either with mettl9-MO or with ctrl-MO (10 ng total injected amount). For each condition, pools of 5 embryos were used for total RNA extraction for each stage.

To confirm the mis-splicing event caused by mettl9-MO, we extracted 2 μg of total RNA from embryos at NF14 and NF22 after bilateral injection at 4-cell stage with mettl9-MO and ctrl-MO, and reverse transcribed it using the High-Capacity cDNA Reverse Transcription Kit (Thermo Fisher Scientific, #4368814), according to the manufacturer's instructions. Then, we quantified mettl9 expression using a forward primer (M9F1) matching the 1$^{st}$ exon and a reverse primer (M9R1) matching the 2$^{nd}$ exon of mettl9.L/mettl.S genes and employed slc35b1.L as a reference transcript[148]. Morpholino oligos and qPCR primers' sequences are listed in Supplementary Data File 4.

For neural phenotype classification, the labels of each individual test tube containing a batch of embryos were scrambled prior to manual microscopical evaluation under the stereomicroscope. Representative images were acquired and displayed in the figures (used for qualitative evaluation) after processing in GIMP using the brightness/contrast, curves and nitidise functions, applied to both controls and experimental samples.

### X. laevis neuralised animal cap preparation

For neuralised animal cap (a.c.) assays, 2-cell stage embryos were co-injected at the animal pole of both blastomeres with either noggin mRNA + mettl9-MO, or with noggin mRNA + ctrl-MO (each blastomere received 23 pg of noggin mRNA + 10 ng of MO). At stage NF9, a.c. were dissected from injected embryos and grown in 1XMBS (88.0 mM NaCl, 1.0 mM KCl, 2.4 mM NaHCO$_3$, 15.0 mM HEPES pH 7.6, 0.3 mM CaNO$_3$-4H$_2$O, 0.41 mM CaCl$_2$-6H$_2$O, 0.82 mM MgSO$_4$) with gentamycin (at 50 mg/ml final concentration) until sibling embryos reached stage NF12.5. For the proteomic and transcriptomic analysis, about 20-60 a.c. were pooled for each replicate.

### mESCs maintenance, neural differentiation and drug treatments

ES-E14TG2a (E14) mouse embryonic stem cell line (mESC; ATCC), derived from a male 129/Ola mouse blastocyst, were used as wild-type (WT), parental cells. Mettl9$^{KO}$, Mettl9$^{Deg}$, Mettl9$^{CatD}$ mESCs lines were generated through genetic engineering of ES-E14TG2a WT mESCs by CRISPR/Cas9[149], as explained below.

mESCs were cultured in Serum/LIF, composed by Dulbecco's Modified Eagle's Medium - high glucose (Sigma Aldrich, #D6546), 15% Fetal bovine serum (Thermo Fisher Scientific, #10500064), 2 mM L-Glutamine (Thermo Fisher Scientific, #25030024), 1X MEM Non-Essential Amino Acids Solution (Thermo Fisher Scientific, #11140050), 1 mM Sodium Pyruvate (Thermo Fisher Scientific, #11360039), 50 μM 2-Mercaptoethanol (Thermo Fisher Scientific, #21985023), 5 ng/ml (1000U/ml) Leukemia Inhibitory Factor (Thermo Fisher Scientific, #A35934). mESCs were detached with StemPro™ Accutase™ Cell Dissociation Reagent (Thermo Fisher Scientific, #A1110501) and seeded on Gelatin-coated plates (EmbryoMax® 0.1% Gelatin Solution, Merck Millipore, #ES-006-B).

For neural differentiation, mESC were cultured for one day (from DIV(−1), 1 day before DIV0, to DIV0) in 2i/LIF medium, composed by

DMEM/F-12 w/o Glutamine (Thermo Fisher Scientific, #21331020), 1X B-27 Supplement minus vitamin A (Thermo Fisher Scientific, #12587010), 1X N-2 Supplement (Thermo Fisher Scientific, #17502048), 0.1X MEM Non-Essential Amino Acids Solution (Thermo Fisher Scientific, #11140050), 2 mM L-Glutamine (Thermo Fisher Scientific, #25030024), 1 mM Sodium Pyruvate (Thermo Fisher Scientific, #11360039), 50 μM 2-Mercaptoethanol (Thermo Fisher Scientific, #21985023), 1 μM MEK inhibitor PD0325901 (Sigma Aldrich, #PZ0162), 3 μM GSK3 inhibitor CHIR99021 (Sigma Aldrich, #SML1046). After 24 hrs (DIV0), 2i/LIF media was replaced by WiBi medium, composed by DMEM/F-12 w/o Glutamine (Thermo Fisher Scientific, #21331020), 1X B-27 Supplement minus vitamin A (Thermo Fisher Scientific, #12587010), 1X N-2 Supplement (Thermo Fisher Scientific, #17502048), 0.1X MEM Non-Essential Amino Acids Solution (Thermo Fisher Scientific, #11140050), 2 mM L-Glutamine (Thermo Fisher Scientific, #25030024), 1 mM Sodium Pyruvate (Thermo Fisher Scientific, #11360039), 50 μM 2-Mercaptoethanol (Thermo Fisher Scientific, #21985023), 2.5 μM Wnt pathway inhibitor 53AH, (Cellagen Technology, #C5324-2) and 0.2 μM TGF-β/BMP inhibitor LDN193189 hydrochloride (Sigma Aldrich, #SML0559). WiBi medium was provided fresh daily until DIV10 (endpoint). From DIV5 until DIV10, WiBi media was supplemented with 0.1 μM Shh agonist SAG dihydrochloride (Sigma Aldrich, #SML1314) to induce a basal telencephalic fate[33]. During neural differentiation (from DIV3 onwards), cells were cultured on Poly-L-ornithine and Laminin-coated plates, with 16 μg/ml Poly-L-ornithine hydrobromide (Sigma Aldrich, #P3655) diluted in H$_2$0 and 3.2 μg/ml Laminin Mouse Protein (Thermo Fisher Scientific, #23017015) diluted in PBS. The coating incubations were performed overnight at 37 °C. All cells were cultured at 37 °C with 5% CO$_2$.

To induce acute METTL9 degradation, 500 nM of dTAG$^{V}$-1 drug (Tocris, #6914/5) or 0.05% DMSO (Merck, #D2650) as a control were supplied to the media of Mettl9$^{Deg}$ mESC cultures, for 2 days (from DIV3 until DIV5), 6 days (from DIV(−1) until DIV5) or 11 days (from DIV(−1) until DIV10), as specified in the respective Results sections and Figures.

Mettl9$^{Deg}$ mESCs were plated the day before tunicamycin (Tocris, #3516/10) treatment. 0.5 μg/ml tunicamycin was added to the media and after 9 hrs cells were collected for SDS-PAGE and western blot analysis.

NSCs (at DIV6) were treated with 10 μM of golgicide A (Merck Millipore, #G0923) for 1 hr and then they were fixed for immuno-fluorescence staining.

### Generation of Mettl9$^{KO}$ mESC lines by CRISPR/Cas9 and selection of homozygous clones

To generate Mettl9$^{KO}$ mESCs lines, CRISPR/Cas9 lentivectors were constructed using the pLentiCRISPRv2 backbone (Addgene, #52961). Single guide RNA (sgRNA) sequences targeting two regions spanning the first exon of Mett9 were designed using CHOPCHOP[150] (Supplementary Data File 4) and synthesised as single-stranded DNA oligonucleotides (Integrated DNA Technologies). After annealing, they were cloned into the BsmBI site of the pLentiCRISPRv2 plasmid according to the provider's protocol[151]. Lentiviral particles were produced by co-transfecting HEK293T cells (ATCC) with the pLentiCRISPRv2-sgRNA plasmid(s), psPAX2 (Addgene, #12260), and pMD2.G (Addgene, #12259) using Lipofectamine 2000 (Thermo Fisher Scientific, #11668019). In brief, 5×10$^5$ HEK293T cells were seeded in a 6-well dish and transfected the next day with 1 μg of pLentiCRISPRv2-sgRNA, 1.5 μg of psPAX2, 0.5 μg of pMD2.G, and 5 μl of Lipofectamine 2000 in 4 mL of Opti-MEM (Thermo Fisher Scientific, #51985034). The medium was changed 6 hours post-transfection with 1 mL of fresh Opti-MEM, and lentiviral supernatant was collected after 24 hours and filtered through a 0.45 μm PVDF filter (Sigma Aldrich, #SLHV033N). E14 mESCs were transduced with the fresh lentivector supernatant by spinfection (45 min, 900 x g, 32 °C), then selected with 2 μg/mL puromycin (Sigma-Aldrich) for 5 days, starting 48 h post-transduction. Surviving cells

were expanded and single-cell clones were isolated by limiting dilution. Clones were amplified and we conducted a first screen for knockout efficiency by measuring the levels of *Mettl9* mRNA expression by RT-qPCR. Then, for clones with >20-fold reduction, we confirmed protein knockout by Western blot analysis using an anti-METTL9 antibody. The clonal control (WT) cell line was obtained from one puromycin-resistant colony generated after transduction with the lentivectors. This clonal WT line displayed normal *Mettl9* mRNA and METTL9 protein levels and was used to control for possible off-target effects of the sgRNAs employed.

### *Mettl9$^{KO}$* cell line validation and off-target analysis by long read sequencing

To characterise the biallelic deletions in the *Mettl9* locus we extracted the genomic DNA from the two *Mettl9$^{KO}$* clones used in the study (#88 and #90) and their parental E14Tg2a cell line with Monarch HMW DNA Extraction Kit for Cells & Blood (New England Biolabs, # T3050L) and performed long-read whole-genome sequencing. In brief, we prepared 3 Oxford Nanopore Technologies (ONT) libraries using the Ligation Sequencing Kit V14 (ONT, # SQK-LSK114), starting with 1 µg of genomic DNA and following manufacturer's instructions. Each library was loaded on a PromethION flow cell (ONT, FLO-PRO114M) and sequenced for 72 h. Reads were basecalled using Dorado (0.3.0, Oxford Nanopore Technologies) and aligned to the mouse genome (GRCm38) using minimap2[152]. Alignment files were visualised using the Integrative Genome Viewer[153].

To assess the presence of potential off-target effects of the CRISPR/Cas9 sgRNA sequences used to generate the *Mettl9$^{KO}$* cell lines, we predicted all putative off-target sites using Cas-OFFinder[154], allowing up to five mismatches across the 20-base target sequence (which is the highest value permitted by the program). We then leveraged the long-read whole genome sequencing data to identify genetic variants in the two Mettl9KO clones (KO#88 and KO#90) as well as their parental line (E14). Specifically, we performed single-nucleotide variant (SNV) calling using DeepVariant[155] and aggregated the resulting VCF files with glNexus[156]. Structural variant (SV) detection was conducted with Sniffles2[157] in multi-sample mode. High-confidence variants were extracted using vcftools[158] with the following parameters: "--minGQ 20 --minQ 30 --max-missing 1 --maf 0.2". To assess the overlap between putative off-target sites and SNV/SV variants, we used bedtools[159], allowing distances of 100, 1k, and 10k nucleotides flanking each side of the potential off-target cuts (resulting in 0SNV + 0SV, 4SNV + 0SV, and 20SNV + 0SV overlapping potential off targets, respectively). Finally, we interrogated the "GT" genotype attributes of each cell line with bcftools[160], confirming that none of the variants overlapping with putative off-target sites were indeed uniquely present in the KO cell lines (Supplementary Fig. 3e), but in fact were also present in the parental cells.

### Generation and validation of *Mettl9$^{Deg}$* mESC lines

For the generation of the *Mettl9$^{Deg}$* mESC lines, CAS9 was guided to the last exon of the endogenous *Mettl9* locus, near the stop codon, by using an sgRNA designed with CRISPOR[161] and cloned into the px330-related px459 (Addgene)[149] gRNA/CAS9 vector. To generate an in frame and C-terminally tagged METTL9-FKBP12$^{F36V}$-FLAG, a pBluescript II KS(-) vector containing the Homologous Directed Repair (HDR) template was used. The HDR sequence consisted of a 5' Homology Arm complementary to ~1 Kb of *Mettl9* Exon 5; the FKBP12$^{F36V}$-3XFLAG-STOP sequence, followed by a 3' Homology Arm complementary to a ~1 Kb region starting from *Mettl9* 3'UTR. The px459 and pBluescript vectors were co-transfected in mESCs and sgRNA/CAS9-containing cells were selected with 0.001 µg/µl puromycin (Merck, #P8833) after 2 days. Cells were plated at single cell density to allow expansion and selection of clonal cell lines. mESC clones were genotyped to identify homozygous *Mettl9$^{Deg}$* clones.

Genomic DNA was extracted from putative *Mettl9$^{Deg}$* mESCs clones, grown on a 96-well plate until confluency (1 well per clone). Cell media was discarded and replaced with 100 µl Lysis Buffer per well (Lysis Buffer: 100 mM Tris pH 8.5, 5 mM EDTA, 0.2% SDS, 200 mM NaCl) supplemented with fresh Proteinase K (Thermo Scientific, # EO0491), 0.1 mg/ml. After lysing cells overnight at 37 °C, lysates were collected and DNA precipitated by adding an isovolume of propan-2-ol (Carlo Erba, #415184) and by mixing them. Samples were centrifuged at 845 x g for 15 minutes and DNA pellets resuspended in nuclease-free H$_2$0. Purified DNA was used as template for PCR. The first genotyping PCR was performed with DreamTaq Green PCR Master Mix (Thermos Fisher, #K1081). For *Mettl9$^{Deg}$* clones, it resulted in the amplification of two bands (233 bp; 758 bp) in the case of homozygous *Mettl9$^{Deg}$* mESCs and a single band (311 bp) for the corresponding untargeted *Mettl9$^{WT}$* locus. Sanger Sequencing was performed on PCR amplicons to confirm the absence of mutations in homozygous *Mettl9$^{Deg}$* cell lines. Then, a second PCR was performed to validate the correct genomic integration of the transgene (degron sequence) using primers over 1 kb upstream and downstream the homology arms (outside the region corresponding to the targeting vector) and by using a Phusion™ Plus DNA Polymerase (Thermo Fisher, #F630S). The resulting PCR product (3116 bp) was further purified and Sanger sequenced. Eventually, the homozygous mESC clones #6, #14, and #17 (all generated with sgRNA #2) were selected and expanded to establish cell lines for experimental use. PCR and Sanger primers, dsDNA and sgRNA sequences are available in Supplementary Data File 4.

The Degron construct did not interfere with METTL9 cellular methyltransferase activity, as confirmed by comparable levels of 1MH levels between untagged (WT) and (uninduced) *Mettl9$^{Deg}$* lines (Figs. 3i and 4b).

### Generation and validation of *Mettl9$^{CatD}$* mESC cell lines

To generate *Mettl9$^{CatD}$* mESCs lines, CAS9 was guided to the third exon of the endogenous *Mettl9* locus, in the region encoding the SAM binding domain, by using an sgRNA transcribed from a px459 vector (as explained before for the *Mettl9$^{Deg}$*). As HDR template we used a 200 nt single-stranded DNA (ssDNA) (synthesised by Integrative DNA technologies, IDT) designed with 4 point-mutations (generating 2 amino acids substitutions: D151K and G153R) which abrogate METTL9 enzymatic activity[14]. The sgRNA-containing px459 vector and the ssDNA were co-transfected into mESCs and clonal cell lines were selected as explained before. Genotyping screening was performed by extracting genomic DNA, as explained in the previous "Mettl9-Degron" section, followed by PCR.

To identify homozygous *Mettl9$^{CatD}$* or untargeted clonal *Mettl9$^{WT}$* mESCs, a 501 bp product was PCR amplified from DNA of putative *Mettl9$^{CatD}$* clones. This PCR was digested with HhaI and resulted in 2 DNA fragments (258 bp and 243 bp) when containing the fourth desired mutation included in the CatD-ssDNA (T > C, see Supplementary Fig. 6a and legend); or in an uncut product (501 bp) when *Mettl9* locus did not contain mutations and came from WT, untargeted mESCs. This identified the homozygous *Mettl9$^{CatD}$* clone #B26 and the untargeted *Mettl9$^{WT}$* clone #B28 ("B" refers to the sgRNA B used); the presence of the other 3 point-mutations in #B26 was further confirmed by Sanger Sequencing. A longer PCR of the *Mettl9* locus was performed to exclude potential genomic re-arrangements following CRISPR/Cas9, in the #B26 and #B28 clones. PCR primers outside the region of the ssDNA donor generated a 1261 bp product for both *Mettl9$^{WT}$* and *Mettl9$^{CatD}$* mESCs, indicating that the CRISPR/Cas9 manipulation did not affect the locus. The sequences of these PCR products were further confirmed by Sanger sequencing. PCR and Sanger primers, ssDNA and sgRNA sequences are available in Supplementary Data File 4.

## Immunofluorescence (IF) and confocal imaging acquisition and analysis

*Mettl9^WT* (E14) and *Mettl9^Deg* (clones #6 and #17) NSCs were grown on ibiTreat slides (8 well chambered, Ibidi, #80806), previously coated with Poly-L-ornithine and Laminin. At DIV6, NSCs were fixed for 10 minutes with 4% paraformaldehyde (Electron Microscopy Sciences, #157-8) diluted in Dulbecco's Phosphate Buffered Saline (PBS) (Thermo Fisher, # 14190094), then washed in PBS, permeabilised with PBS + 0.3% Triton X-100 for 10 minutes and incubated with blocking solution (PBS + 0.1% Triton-X100, 3% goat serum, 1% BSA) for 1 h. Fixed cells were incubated with primary antibodies diluted in blocking buffer at 4 °C, overnight. Afterwards, cells were washed 4 times with PBS + 0.1% Triton X-100 (5 minute each wash), incubated with secondary antibodies diluted in blocking buffer. After 3 washes, nuclei were stained with Hoechst 33258 (Thermo Fisher Scientific, #H3569) diluted 1:5000 in PBS + 0.1% Triton X-100. Samples were stored in Aqua-Poly/Mount mounting media (Polysciences, #18606-20) at 4 °C. All incubations were performed on a rocker at room temperature (unless specified) with gentle agitation. Primary antibodies used: anti-FLAG (Sigma, #F1804), anti-GOLGA2/GM130 (Proteintech, #11308-1-AP), anti-NESTIN (Cell Signaling Technology, E4O9E XP® BK73349S), anti-TOM20 (Proteintech, #11802-1-AP), anti-GORASP2 (Proteintech, #10598-1-AP), anti-CALRETICULIN (Proteintech, #27298-1-AP). Secondary antibodies (used at 1:500 dilution): Goat anti-Mouse Alexa Fluor™ Plus 488 (Thermo Fisher Scientific, #A32723TR), Goat anti-Rabbit Alexa Fluor™ Plus 488 (Thermo Fisher Scientific, #A32731TR), Goat anti-Mouse Alexa Fluor™ Plus 647 (Thermo Fisher Scientific, #A32728TR). Anti-tag antibody specificity was always confirmed by the lack of signal by IF and WB in non-tagged cell lines as well as their widespread use in the literature. Other primary antibodies used for IF were chosen for their use in the specialised literature and confirmed by their expected, well-defined subcellular localization pattern.

Imaging was performed on a Confocal microscope (Nikon A1 MPrs, Nikon instruments, Yokohama, JP). Galvano scan mode was used, with sequential scanning allowing the switch starting from the longest to shortest wavelengths lasers (640 nm, 488 nm and 405 nm) to avoid crosstalk among different fluorophores. 40X-water-1.15 Numerical Aperture (NA), 60X-oil-1.49 NA or 100X-oil-1.45 NA objectives were used. Image analysis and counts was performed using Fiji[162]. Only for Nestin staining of *Mettl9^Deg* line, the microscope and acquisition parameters are as specified for *Mettl9^Deg*/LCS-HA (following section).

All manual image counts, such as the Golgi morphology analysis and the quantification of NESTIN-positive (NES^+) cells, were performed in a blind after scrambling the image order and assigning random labels. The Fiji Cell Counter plugin was used to keep track of the counts. To prevent bias in counting NES^+ cells in the *Mettl9^Deg* line, a composite image with only 2 channels (Hoechst and Nestin) was used for counting (the FLAG channel was excluded for the counting).

To quantify the extent of the co-localisation between METTL9 (anti-FLAG) and different subcellular markers, confocal images were acquired from immune-stained wild-type (background control) and *Mettl9^Deg* mESCs lines: each field of view contained approximately 45 cells with 31-36 optical slices (Z-stacks). Each slice was analysed with JACoP plugin (https://imagej.net/plugins/jacop)[163] in Fiji[162]. Otsu and Yen were used as Auto Thresholds A and B and the M2 Manders' coefficient[164] was used for the analysis.

## *Mettl9^Deg*/LCS-HA line generation and immunofluorescence (IF) with digitonin

A C-terminally HA-tagged version of the lactosylceramide synthase (encoded by the human *B4GALT5* gene) LCS-HA, was stably expressed in the *Mettl9^Deg* (clone #6) line by using the *piggyBac* (PB) system[165], to provide a positive Golgi luminal control in digitonin-IF, as in ref.[75], since its C-terminal HA epitope is exposed to the Golgi lumen. A minimal

*piggyBac* transposon plasmid (PB), available in the lab, was used as backbone for LCS-HA cloning. Briefly, this vector was derived from a commercially available plasmid (VectorBuilder, Cat #VB010000-9782ymj) from which the f1 origin of replication and the lac operon system were removed to decrease the plasmid size. The correct assembly was verified by whole-plasmid Oxford Nanopore Sequencing (GENEWIZ). This minimal vector was linearised by PCR (Phusion™ Plus DNA Polymerase (ThermoFisher Scientific, #F630S)) and gel purified. LCS-HA-IRES-hygromycin sequence was amplified from the pyCAG-LCS-HA plasmid (generated in the lab by amplifying LCS-HA sequence from LCS-HA-pcDNA4b[166]) by Phusion™ Plus DNA Polymerase with two primers, each containing a 15 nt homology region with the PB plasmid and gel purified. This PCR-amplicon was cloned into the linearised PB plasmid using In-Fusion cloning according to manufacturer instructions (In-Fusion snap assembly master mix, Takara, #638948). All PCR and Sanger primers are listed in Supplementary Data File 4. mESCs were co-transfected with this LCS-HA-PB plasmid and with a hyperactive *piggyBac* transposase[167] (VectorBuilder, Cat# VB900088-2874gzt) in a 3:1 mass ratio by jetOPTIMUS® DNA Transfection, (Sartorius, # 101000051). To select *Mettl9^Deg* cells stably expressing LCS-HA, mESCs were cultured for 2 passages with 0.17 µg/µL Hygromycin B (Thermo Fisher Scientific, #10687010). *Mettl9^Deg*/LCS-HA cells were differentiated and processed for IF.

The digitonin-IF protocol was adapted from Ricciardi et al., 2022[75]. *Mettl9^Deg*/LCS-HA and E14 WT (control) NSCs were differentiated on Ibidi slides (#80426) and at DIV5 they were fixed for 15 minutes with fresh 2% paraformaldehyde (Electron Microscopy Sciences, #157-8) diluted in Dulbecco's Phosphate Buffered Saline (PBS) (Thermo Fisher, # 14190094). After three washes in Buffer A (20 mM PIPES pH 6.8, 137 mM NaCl, 2.7 mM KCl), cells were permeabilised with 10 µM Digitonin (Merck Sigma-Aldrich, #D141) (diluted in Buffer A), for 5 mins. Cells were incubated with blocking buffer (5% FBS [vol/vol], 50 mM NH₄Cl in Buffer A), without permeabilising agents, for 1 h. Then, samples were incubated with primary antibodies diluted in blocking solution at 4 °C, overnight. Afterwards, cells were washed 4 times with Buffer A and then incubated with secondary antibodies diluted in blocking buffer. After 3 washes, nuclei were stained with Hoechst 33342 (Thermo FisherScientific, # H3570) diluted 1:5000 in Buffer A. Samples were stored in Aqua-Poly/Mount mounting media (Polysciences, #18606-20) at 4 °C. All incubations were performed on a rocker at room temperature (unless specified) with gentle agitation. Primary antibodies used: mouse anti-FLAG (Sigma, #F1804) (1:600) and rabbit anti-HA-Tag (C29F4) (Cell Signaling, #3724) (1:700). Secondary antibodies (used at 1:500 dilution): Goat anti-Rabbit Alexa Fluor™ Plus 488 (ThermoFisher Scientific, #A32731TR), Goat anti-Mouse Alexa Fluor™ Plus 647 (Thermo Fisher Scientific, #A32728TR).

Imaging was performed on a Nikon AX-NSPARC, (confocal mode). Resonant scan mode was used, with sequential scanning allowing the switch from the longest to the shortest wavelength (640 nm, 488 nm, 405 nm). 40x SiliconOil (NA: 1.25) objective was used. Since only a subset of *Mettl9^Deg* expressed also LCS-HA at DIV5 (probably due to construct silencing) microscopy acquisition was performed on fields of view displaying LCS-HA positive cells (in Triton samples), and randomly in the digitonin samples, as the latter did not have any detectable signal in the 488 channel (LCS-HA).

Digitonin-permeabilised *Mettl9^Deg*/LCS-HA samples were processed in parallel with the Triton-permeabilised *Mettl9^Deg*/LCS-HA samples (standard IF protocol as in the previous Method section). Antibodies concentrations and microscopy acquisition parameters were kept the same for both Triton- and digitonin-permeabilised cells. For image processing (in Fiji[162]), the brightness and contrast values of digitonin-permeabilised *Mettl9^Deg*/LCS-HA samples were set on their control (the blank digitonin-permeabilised WT E14 line). A median filter with 0.5 px radius was applied for images visualization of both Triton and digitonin samples.

## RUSH Methods

The sequence encoding Str-KDEL-IRES-ManII-SBP-EGFP from the #65252 Addgene plasmid was PCR amplified and cloned into a pyCAG-hygro vector as described for LCS-HA (see *Mettl9Deg*/LCS-HA line generation). The resulting Str-KDEL;ManII-SBP-EGFP-pyCAG plasmid was used as a template for molecular cloning of the Str-KDEL;ManII-SBP-EGFP sequence into the *piggyBac* (PB) plasmid. PCR and Sanger primers are in Supplementary Data File 4. mESCs were co-transfected with this (Str-KDEL;ManII-SBP-EGFP)-PB plasmid and with a hyperactive *piggyBac* transposase in a 3:1 mass ratio. 48 hrs later GFP positive cells were FACS sorted in a SH800S instrument (Sony) and expanded in culture. mESCs stably expressing the GFP were then differentiated in WiBi media until DIV6/7, before performing live imaging microscopy.

After observing leakiness of the GFP from the ER to the Golgi at the steady state, due to the presence of free biotin coming from DMEM/F-12 and B27-Supplement, we sequestered it by supplying 0.1 mg/ml of NeutrAvidin Protein (Thermo Scientific™, #31000) to the WiBi media for 2-3 days before performing live imaging. We chose this strategy as a compromise ensuring the lowest GFP leakiness while maintaining optimal cell viability, as NSCs did not tolerate the presence of NeutrAvidin for the entire duration of the differentiation (probably due to the requirement of low amounts of biotin to support neural development). Before live imaging, DIV6/DIV7 NSCs were washed twice and incubated with DMEM/F-12 without phenol red (Thermo Fisher Scientific, #21041025), fresh 0.1 mg/ml NeutrAvidin, in a live imaging chamber (at 37 °C and 5% CO2), of an AX-NSPARC microscope. After identifying GFP-positive cells, their spatial coordinates were saved for later automatic timelapse acquisition. The timecourse started as soon as old media was replaced with pre-warmed 80 µM Biotin-media (without NeutrAvidin). Live-imaging was performed on a Nikon AX-NSPARC (confocal mode). Resonant scan mode was used, with a 488 nm laser and a 40x-Water Immersion (NA: 1.25) objective.

RUSH live imaging data were analysed following a procedure adapted from refs. 96,168,169 using a macro in Fiji[162]. Briefly, *Mettl9WT* and *Mettl9KO* time series were randomised before starting the analysis, in order to avoid any cell selection bias. Cells expressing the RUSH constructs were manually chosen using individual polygonal selections. The selection process allowed to avoid cells with an excessive or very dim fluorescence or presenting significant leakage from the ER at the beginning of the time course. Then, for each cell in every temporal slice, we measured the average background-corrected integrated pixel density within a region of interest (ROI) corresponding to the Golgi complex. This ROI was determined from a threshold mask obtained at a later time point and applied throughout the time-lapse analysis. In addition, using the same approach we quantified the signal within the ER region, obtained from an initial ROI mask. We quantified the time courses data of more than 390 cells per condition across 3 independent experiments (*Mettl9WT* n = 133, 156, 107; *Mettl9KO* n = 210, 88, 212). We then showed the distribution of both 1) normalised Golgi signal, 2) normalised ER signal and 3) Golgi/ER ratio over time and compared the median *Mettl9WT* and *Mettl9KO* values at each time point by paired t-test.

## Total proteome preparation from acutely depleted *Mettl9Deg* and *Mettl9KO* NSCs

Differentiating *Mettl9Deg* (clone #6) mESCs were seeded on Poly-L-ornithine and Laminin coated plates (12-well plates) at DIV3 and cultured with WiBi media supplemented with 500 nM of dTAGV-1 drug or 0.05% DMSO. Media was supplied daily with fresh DMSO or dTAGV-1 drug until DIV5 (for 2 days), when neural stem cells (5 wells per treatment) were collected. Cells were washed with DMEM/F-12 (#21331020), and each well was lysed on ice with 300 µl of Lysis buffer: 250 mM NaCl, 50 mM Tris-HCL pH 8.00, 1% IGEPAL® CA-630, 0.5% Sodium Deoxycholate, 0.1% Sodium dodecyl sulfate (SDS), EDTA-free protease inhibitors (Roche, #11873580001). Lysates were transferred into protein low binding tubes and sonicated for 10 cycles (30 seconds ON, 30 seconds OFF) at 4 °C with the Bioruptor® Pico instrument (Diagenode). Protein extracts were centrifuged at 16200 x g and clear supernatants were quantified with the BCA Protein Assay (Pierce, #23227), on the NanoQuant Plate™ Infinite F PLEX spectrometer (Tecan). 100 µg of total protein extracts were precipitated with 4 volumes of ice-cold acetone overnight at −20 °C. After centrifuging at 16200 x g, 4 °C, precipitated proteins were washed with ice-cold acetone and centrifuged again. Protein pellets were dried at R.T. and frozen in dry ice. Protein pellets were further processes to obtain peptides suitable for Mass Spectrometry analysis (see LC-MSMS analysis section). Pellets were used as input for the PreOmics iST Sample Preparation Kit iST 8x (#P.O.00001 or #P.O.00027) and proteins were extracted and digested following the manufacturer's guidelines.

WT E14 and *Mettl9KO* (#88) mESCs were differentiated until DIV5 as described above and protein extracted as explained before. 100 µg of total protein extracts were precipitated as previously described and processed with the PreOmics iST Kit (see before) for Mass Spectrometry.

## METTL9 immunoprecipitation for Mass Spectrometry

Differentiating *Mettl9Deg* (clone #6) mESCs were collected at DIV4: 9 ×10 cm plates of cells were washed with DMEM/F-12, detached by StemPro™ Accutase™ (Thermo Fisher Scientific, #A1110501) and cell pellets lysed with 0.05 M Tris-HCL pH 8.0, 0.1 M KCl, 20% Glycerol, 0.3% v/v IGEPAL® CA-630 for 10 minutes on ice. After centrifugation at 16200 x g for 10 minutes, clear supernatants were collected, and protein extracts were quantified by BCA Protein Assay (see above). The lysate was diluted 1:1 with IP Lysis Buffer (0.05 M Tris-HCL pH 8.0, 0.1 M KCl, 0.3% v/v IGEPAL® CA-630) and then split into 8 tubes, with 1681 µg of protein extracts each. Each individual IP reaction was performed by incubating a single tube with 50 µl of Dynabeads™ Protein G (Thermo Fisher Scientific, #10004D) pre-crosslinked with either normal mouse IgG (Santa Cruz Biotechnology, #sc-2025) or with anti-FLAG antibody (Sigma Aldrich, #F1804), corresponding to 5 µg of antibody for each IP. Cross-linking was performed as following: anti-IgG and anti-FLAG were incubated with pre-washed Dynabeads™ Protein G (ratio 50 µg antibody with 500 µl ProteinG) for 2 hrs at R.T., after washes, beads were resuspended in 20 mM DMP (dimethyl pimelimidate) (Thermo Fisher Scientific, #21666) in Borate Buffer (40 mM Boric acid, 40 mM Sodium tetraborate decahydrate) at R.T. for 30 mins. Afterwards, they were washed and crosslinked AB-beads resuspended in double the amount of lysis buffer (50 mM Tris-Hcl pH 8.0, 150 mM KCl, 0.1% v/v Triton X-100). This Buffer was also used for previous washes.

After 2 hrs at 4 °C, immunocomplexes on the beads were separated from the unbound fraction on a DynaMag™−2 Magnet (Thermo Fisher Scientific, #12321D). Immunocomplexes were washed 3 times with IP Lysis Buffer. Elution of protein complexes from antibody-beads was performed with 0.5% SDS, 0.05 M Tris-HCL pH 8, at 60 °C for 15 min, with gentle agitation. 45 µl of the eluted fraction were precipitated with 4 volumes of acetone at −20 °C overnight whereas 5 µl were used for quality control (Western Blot). Dried protein pellets were digested to obtain peptides for Mass Spectrometry, by using the PreOmics iST Sample Preparation Kit iST as explained before for the Total proteome preparation.

## Total proteome preparation from *X. laevis* neuralised *ctrl*-MO or *mettl9*-MO animal caps

5 Pools of 20-60 neuralised *ctrl*-MO and *mettl9*-MO animal caps were collected at stage 12.5 from different experimental batches, as described in (Supplementary Data 5). They were snap frozen in dry ice and lysed with 100 µl of JS Buffer (0.25 M HEPES pH 7, 0.15 M NaCl, 1% Glycerol, 1% Triton X-100, 0.0015 M MgCl2, 0.005 M EGTA) supplemented with EDTA-free protease inhibitors (Sigma Aldrich, #11873580001). 90 µl were further processed for protein extraction

and quantified by BCA Protein Assay. 44 μg of protein extracts were precipitated with 4 volumes of ice-cold acetone and concentrated protein pellets were used as input for PreOmics iST Sample Preparation to generate peptides for Mass Spectrometry analysis, as explained in the section "Total proteome preparation from acutely depleted *Mettl9^Deg* NSCs".

## LC-MS/MS analysis

In all cases, peptide mixtures were separated by reversed-phase chromatography using an EASY-nLC 1200 ultra-high-performance liquid chromatography system with an EASY-Spray column (Thermo Fisher Scientific), 25 cm in length (inner diameter 75 μm, PepMap C18, 2 μm particles). For the total proteome experiments, a Q Exactive Plus was used, while a Q Exactive HF was used for the immunoprecipitation experiment. Purified peptides were loaded in buffer A (0.1% formic acid in water) at a constant pressure of 980 bar. For total proteome experiments, digested peptides were separated using the following gradient: 70 minutes from 5% to 20% buffer B (0.1% formic acid, 80% acetonitrile), 15 minutes from 20% to 30% buffer B, 5 minutes from 30% to 65%, and 5 minutes from 60% to 95%, at a constant flow rate of 300 nl/min. For the IP experiment, a slightly shorter gradient was used (from 5% to 20% buffer B over 50 minutes). The column temperature was maintained at 45 °C using EASY-Spray oven control.

The mass spectrometry instruments were operated in "top-15" data-dependent acquisition (DDA) mode. MS spectra were collected in the Orbitrap mass analyser within a range of 375 to 1650 m/z with an automatic gain control (AGC) target of 3e6 and a maximum ion injection time of 20 ms, at a resolution of 70,000 for the Plus and 60,000 for the HF. The 15 most intense ions from the full scan were sequentially fragmented with an isolation width of 1.8 m/z for the Plus and 1.4 m/z for the HF. Normalised collision energy (NCE) was set to 28%. The resolution used for MS/MS spectra collection in the Orbitrap was 17,500 for the Plus and 30,000 for the HF instrument, with an AGC target of 1e5 and a maximum ion injection time of 100 ms. Precursor dynamic exclusion was enabled with a duration of 20 seconds.

## Mass Spectrometry (MS) data analysis

MS raw files were processed using MaxQuant version 2.1.2.0. The extracted MS/MS spectra were matched by the Andromeda search engine against tryptic peptides (allowing a maximum of two missed cleavages) derived from the Uniprot UP000000589 mouse database (54,679 entries) for the total proteome experiments and the Uniprot UP000186698 database (61,614 entries) for the IP experiment. The search included cysteine carbamidomethylation as a fixed modification and methionine oxidation and N-terminal protein acetylation as variable modification. The required minimum peptide length was seven amino acids, with maximum mass tolerances of 4.5 ppm for precursor ions after nonlinear recalibration and 20 ppm for fragment ions. The MAXLFQ algorithm was enabled with a minimum ratio count as described in Cox et al., 2014[170]. If applicable, peptide identifications were transferred between samples by 'match between runs' within a 0.7-minute window after retention time alignment. Raw and processed data, including proteinGroups MaxQuant output files, are available via ProteomeXchange with identifier PXD053437.

Exploratory analysis and quality control were performed using the Perseus platform[171]. Differential protein abundance was computed using the DEqMS Bioconductor library[172], which accounts for technical and biological variance in label-free mass spectrometry datasets. In brief, the proteinGroups output table from MaxQuant was filtered for "reverse", "only identified by site," and "contaminants." LFQ intensities were log-transformed and median-centered across the replicates. Protein hits with values missing (NA) in more than 50% of the samples were discarded. Remaining missing values were imputed using gamma regression fit of the mean-variance trend with the single_imputation function from the PaiR library (github.com/PhilipBerg/pair). Finally, a

linear model fit was obtained using the limma *lmFit* function[173] and computed moderated t-statistics using *eBayes* function. Whenever possible, batch correction was enforced by providing the experimental group information in the model matrix, similarly to the RNA-seq analysis (Supplementary Data 5).

For the IP-mass spectrometry data, a one-sided (P[X > x]) Student's t distribution function was used for the moderated t-statistics, focusing solely on genes enriched in anti-FLAG compared to control IgGs. All the upstream steps were performed as described above.

## Immunoprecipitation (IP) with anti-HA beads in mESCs

2 ×106 mESCs (E14) were seeded in a 10 cm plate and transfected in suspension by jetOPTIMUS® DNA Transfection (Sartorius, # 101000051), with 5 μg of DNA. DNA included 2 plasmids for each of the 4 conditions: (i) 2.5 μg of pyCAG-METTL9-FLAG (WT) and either 2.5 μg of pyCAG-STMN1-HA or (ii) 2.5 μg of pyCAG-HA-RAB2a; (iii) 2.5 μg of pyCAG-METTL9-CatD-FLAG and with either 2.5 μg of pyCAG-STMN1-HA or (iv) 2.5 μg of pyCAG-HA-RAB2a. All the pyCAG constructs were engineered by cloning synthetic dsDNA encoding for METTL9-FLAG; METTL9-CatD-FLAG; STMN1-HA and HA-RAB2a into pyCAG-hygromycin vectors by Gibson assembly (see also the "In silico prediction of N-glycosylation sites and generation of METTL9 mutant constructs (N35Q, N35Q;N86Q and SP*)" section). Sequences of dsDNAs and Sanger primers are provided in Supplementary Data File 4. 48 hrs after transfection cells were harvested and lysed as described in the "METTL9 immunoprecipitation for Mass Spectrometry" section, with the addition of EDTA-free protease inhibitors (Roche, #11873580001). Around 600 μg of protein extracts were used for each IP, which was incubated with either 85 μl of pre-crosslinked Pierce™ Anti-HA Magnetic Beads (Thermo Fisher Scientific, #88836) or 85 μl of pre-crosslinked Dynabeads™ Protein G (Thermo Fisher Scientific, #10004D) with normal mouse IgG (Santa Cruz Biotechnology, #sc-2025). Cross-linking of both types of beads was performed as described in the "METTL9 immunoprecipitation for Mass Spectrometry" section). The IP was left over-night and the unbound fractions were collected for WB analysis. Washes and elution of IP-beads were performed exactly as described for the IP for Mass Spectrometry. Samples were analysed by WB, as in the "Western blot" section.

## In vitro Methyltransferase assays

The DNA sequences corresponding to mouse *Mettl9*-FLAG and *Stmn1*-HA were PCR amplified by a Phusion™ Plus DNA Polymerase (Thermo Fisher, #F630S) from pyCAG-METTL9-FLAG and pyCAG-STMN1-HA plasmids respectively (pyCAG plasmids generated in this study, as explained in other sections) with PCR primers (Supplementary Data File 4) including EcoRI and XhoI restriction sites. PCR products were cloned in frame with the GST-3C (as in Davydova et al., 2021[14]) of EcoRI-XhoI-digested pGEX-6P-1 plasmid (available in T. Kouzarides Lab), resulting in the generation of recombinant GST-3C-METTL9-FLAG and GST-3C-STMN1-HA proteins, simply referred to GST-METTL9 and GST-STMN1.

Expression of GST-METTL9, GST-STMN1, and GST-control (GST) recombinant proteins was induced by addition of 0.1 mM Isopropyl-β-D-thiogalactoside (IPTG) to BL21 *Escherichia coli* (Invitrogen, # C600003) cultures and incubation overnight at 18 °C. Proteins were affinity purified using Glutathione Sepharose 4B (Sigma-Aldrich) according to manufacturer's instruction, eluted in TRIS Base Sodium (TBS) buffer (50 mM TRIS pH=7.5 with NaOH, 150 mM NaCl) + 50 mM reduced glutathione) and dialyzed over-night against TBS + 10% glycerol, followed by snap-freezing. The concentration of the recombinant proteins was calculated by Bradford assay (Biorad, #5000113 and #5000114) and their purity was evaluated by SDS-PAGE (GenScript, SurePAGE™, Bis-Tris, #M00655) followed by Coomassie staining (Abcam, #ab119211). The methyltransferase reactions were performed in 20 mM Tris-HCl pH 8; 50 mM NaCl; 1 mM EDTA; 3 mM

MgCl$_2$; 0.1 mM DTT; 0.1 mg/ml BSA buffer, with 10 µM S-adenosyl Methionine (SAM) as methyl donor and 1 µM of GST-METTL9. As substrate, we used the 5 µM of peptide "GHGHSHGSGHGHSHSLFN" (from mouse SLC30A7 protein, amino acids 163-180, purchased from ProteoGenix) (SLC30A7$_{163\text{-}180}$), as in Davydova et al., 2021[14]. The reactions were incubated for 1 hour at 37 °C and stopped by addition of trifluoroacetic acid (TFA) to 0.125 % final concentration. The methyltransferase activity was measured by the non-radioactive MTase-Glo™ Methyltransferase Assay (Promega, #V7601) following manufacturer instructions. To rule out any possible interference of either the enzyme or competitors with the detection method, the conversion of S-adenosyl-homocysteine (SAH) into luciferase activity was assessed in the presence of each individual protein (Supplementary Fig. 9d) and it was found to be unaffected. All the experiments were performed three times, including technical triplicates in each of them.

### RNA extraction and quality control
Total RNA for reverse transcription and qPCR or for RNA-seq library preparation was extracted using the TRIzol Reagent (Thermo Fisher Scientific, #15596026) and chloroform, following manufacturer's instructions. RNA was further purified with RNA Clean & Concentrator-5 (Zymo Research, #R1016) and quantified by using Qubit™ Broad Range or Qubit™ RNA HS Assay Kit (Thermo Fisher Scientific, #Q33266 and #Q32855) and its integrity assessed with an Agilent 2100 Bioanalyzer system, by using the Agilent RNA 6000 Pico or Nano Kits (Agilent Technologies, #5067-1513 and #5067-1511).

In the case of *X. laevis* animal caps, 10 µl of the tissue lysates from the proteomic preparation were used for total RNA extraction immediately after lysis, adding 390 µl of TRIzol Reagent and following the same procedure.

Total RNA was extracted from *mettl9*-MO or *ctrl*-MO *X. laevis* NF22 embryos (5 embryos per pool per each stage) using TRIzol/chloroform.

### Reverse transcription and qPCR
Total RNA was reverse transcribed into cDNA with the High-Capacity cDNA Reverse Transcription Kit (Thermo Fisher Scientific, #4368814). cDNA was used as a template for qPCR by using the iTaq™ Universal SYBR® Green Supermix (Biorad, #1725124) and appropriate primers and a C1000™ Touch Thermal Cycler (Biorad, CFX96™ Real-Time System). qPCR primers' sequences are listed in Supplementary Data File 4. When sequences were taken from PrimerBank[174,175] a PrimerBank ID is indicated.

### RNA sequencing
mRNA sequencing libraries from mESCs (E14; clonal *Mettl9$^{WT}$* and clones #88 and #90 for *Mettl9$^{KO}$*; clones #6, #14 and #17 for *Mettl9$^{Deg}$*; clones #18 *Mettl9$^{WT}$* and #26 for *Mettl9$^{CatD}$*) and *X. laevis* a.c. experiments were prepared using the SMART-Seq® HT PLUS Kit (Takara #R400748 & #R400749) and multiplexed with the Unique Dual Index Kit – 48U (Takara, #R400744), according to manufacturer's instructions. 1 ng of total RNA was used as an input.

For *X. laevis* whole embryo experiments (NF22), transcriptomic libraries were generated using the Stranded mRNA Prep Kit (Illumina, #20040532) starting from 200 ng of total RNA.

Equimolar amounts of libraries were loaded and sequenced on a NovaSeq 6000 (Illumina).

### Differential gene expression analysis
Transcript counts from RNA-seq data were obtained using Salmon[176] in quasi-mapping mode using *M. musculus* (GRCm38, GENCODE version M22 - Ensembl 97) or *X. laevis* (Xenbase v10.1) transcriptomes and decoys and aggregated at the gene level using the tximport package[177].

Hierarchical clustering analysis was performed on normalised global gene expression data using a dissimilarity metric defined as (1 − Pearson correlation coefficient), with Ward's method employed for clustering.

Differential gene expression analysis was conducted using DESeq2[178]. In brief, for total RNA-seq data, we tested the fold-change and significance between each pair of conditions using a negative binomial Wald test, and genes with a Benjamini and Hochberg-corrected p value lower than 0.05 were considered statistically significant, unless stated otherwise.

In some experiments, Principal Component Analysis revealed that a high proportion of the variance was explained by the variability among different experimental replicates. In order to factor these effects out, we performed a batch effect correction by using a generalised linear model (glm) to fit count distributions taking into consideration both the treatment and experimental batch (whenever this was meaningful and allowed by the experiment layout; see Supplementary Data 5).

For the analysis of the global effects of Mettl9 depletion across the 3 mESC cell lines (*Mettl9$^{KO}$*, *Mettl9$^{Deg}$* and *Mettl9$^{CatD}$*) at DIV5, we used as a factor of the glm fit the presence/absence of physiological Mettl9 and the different experiments across all cell lines.

GO enrichment analysis was performed using the clusterProfiler package[179], extracting the genes with a significant up- or down-regulation (i.e. with an adjusted p value < 0.05, unless stated otherwise). The enrichment of each term was deemed significant when its q value was <0.01 (Benjamini-Hochberg corrected p value of the hypergeometric test), and significant terms were ranked by set size.

### Western Blot
Protein extracts were resuspended in Laemmli Sample Buffer (Biorad, #1610747), boiled for 5 min at 95 °C and subjected to SDS-PAGE: proteins were loaded into NuPAGE™ 10%, Bis-Tris, 1.0–1.5 mm, Mini Protein Gels (1.5 mm/10wells, Thermo Fisher Scientific, #NP0315BOX) with a protein ladder (Biorad, #1610374) in SDS running buffer (Invitrogen, #NP0001), at 140 Volts for ~1.5 hrs. Proteins were transferred into a nitrocellulose membrane (Cytiva, #GE10600002) (or into a Immun-Blot PVDF membrane (Biorad, #1620177) only for RAB2a and STMN1 anti-HA-IP), with Transfer Buffer (Thermo Fisher Scientific, #NP00061) and 10% MeOH (20% MeOH was used only for transferring RAB2a and STMN1 anti-HA-IP), at 90 Volts for 1 hr on ice. Membranes were blocked in PBS 0.1% Tween 3% Nonfat milk (Applichem, #APA08301000) and incubated with the appropriate primary antibodies diluted in PBS-0.1% Tween-3% Nonfat milk at 4 °C overnight. Primary antibodies: mouse anti-FLAG (Merck Millipore, #F3165) (1:1000 dilution), rabbit anti-METTL9 (Custom antibody, see next section) (1:400), rabbit anti-HA (BioLegend, #902301) (1:2000) and rabbit anti-Alpha Tubulin antibody (Proteintech, #11224-1-AP). Membranes were washed 3 times with PBS-0.1% Tween and incubated 1 hr at room temperature with the secondary antibodies (1:10000 dilution): HRP-conjugated Goat anti-Mouse IgG (Proteintech, #SA00001-1) and HRP-conjugated Goat anti-Rabbit IgG (Thermo Fisher Scientific, #G-21234), or directly with anti-β-Actin−Peroxidase antibody Mouse monoclonal (Merck Millipore, #A3854) (1:10000). After 3 washes, membranes were incubated with a chemiluminescent HRP substrate (Thermo Fisher Scientific, #34578 or #A38554) and protein signal detected using a ChemiDoc™ MP apparatus (Biorad).

### Custom anti-METTL9 antibody production
A custom antibody targeting mouse METTL9 was generated by Biomatik LLC (US). In brief, the peptide sequence RMWTLRSPLSRSLYVN-Cys corresponding to amino acids 20-35 of METTL9 protein was synthesised and conjugated to Keyhole Limpet Hemocyanin (KLH). Subsequently, a New Zealand rabbit was immunised with the antigen, initially receiving Complete Freund's Adjuvant on days 0 and 14, followed by Incomplete Freund's Adjuvant on days 28 and 35. Additional

immunizations were administered with 0.9% NaCl on days 42, 49, 56, and 63. The final bleed was obtained on Day 70, and the resulting immune serum underwent affinity purification. The purified antibody was validated by ELISA, yielding a titer greater than 1:32,000.

### Protein hydrolysis for methyl-histidine content analysis

The following mESC lines: $Mettl9^{WT}$ (E14) and $Mettl9^{KO}$ (clone #88); $Mettl9^{Deg}$ (clone #17) DMSO- or dTAG$^V$-1-treated (from DIV(−1) until DIV6); $Mettl9^{WT}$ (clone #18) and $Mettl9^{CatD}$ (clone #26) were grown and differentiated until DIV6. Each replicate consisted of pools of 2 wells of a 6-well-plate. At DIV6 NSCs were trypsinised, centrifuged and pellets were washed with PBS, before being snap-frozen in dry ice. Cell pellets were lysed with RIPA Lysis Buffer (0.150 M NaCl, 0.05 M Tris-HCl pH 8.0, 1% (v/v) IGEPAL® CA-630, 0.5% Sodium Deoxycholate, 0.1% SDS) and protease inhibitors-EDTA free to obtain protein extracts. Subsequently, proteins were precipitated in ice-cold acetone and pellets were subjected to acidic hydrolysis, mainly as performed by Davydova et al., 2021[14]. The frozen dry pellets were transferred into vacuum hydrolysis tubes (Thermo Fisher Scientific, 1 mL, 8 mm×60 mm; #29570), 200 µL of 6 N HCl Sequencing Grade solution (Thermo Fisher Scientific, #24308) was added, and the pellets were dissolved by sonication. The reaction tubes were subjected to three evacuation-refill (vacuum-argon) cycles using a Schlenk line. The pellets were then hydrolysed under vacuum at 110 °C (Labnet - AccuBlock Digital Dry Bath). After 48 h, the reaction mixtures were cooled to room temperature, diluted with $H_2O$ (200 µL), and the hydrolysis reagents were removed by lyophilisation (Scanvac- CoolSafe Pro). The pellets were then suspended in 900 µL of $H_2O$ and filtered through 0.20 µm Titan3™ PVDF syringe filters (Thermo Fisher Scientific, #42204-PV) to remove insoluble material. The aqueous solutions were lyophilised to obtain the samples, which were then stored at −80 °C until LC-MS/MS analysis (performed by BEVITAL AS) for quantification of methylhistidine content (1MH, 3MH and Histidine).

### Glycosidase treatment of mESCs protein extracts

Protein extracts from $Mettl9^{Deg}$ mESCs were prepared by lysing cells with 0.25 M NaCl, 0.05 M Tris-HCl pH 7.4, 1% NP40, 0.005 M EDTA, supplemented by Protease inhibitors. 9 µg of protein extracts were used for Endo $H_f$ (New England Biolabs, #P0703S), rPNGase F (New England Biolabs, #P0704S) or O-Glycosidase (New England Biolabs, #P0733S) and Neuromidase treatment, for 2 hrs at 37 °C. Additional reactions were prepared in parallel (as described above) and denatured with Glycoprotein Denaturing Buffer (New England Biolabs) at 100 °C for 10 minutes prior to enzymatic treatment.

### In silico prediction of N-glycosylation sites and generation of METTL9 mutant constructs (N35Q, N35Q;N86Q and SP*)

The in silico prediction of putative N-glycosylated residues was performed on the NetNGlyc-1.0[54] server, by inserting METTL9 amino acid sequence as input. The pyCAG-METTL9-FLAG plasmid, and those with mutated amino acids within $Mettl9$ coding sequence (METTL9-N35Q-FLAG (N35Q), METTL9-N35Q;N86Q-FLAG (N35Q;N86Q), and SP*-METTL9-FLAG (SP*), as described below), were generated by cloning the $Mettl9$-FLAG sequence into a pyCAG-Hygromycin plasmid by Gibson assembly (Thermo Fisher Scientific, #A46627). WT or mutated $Mettl9$-FLAG sequences were synthesised as dsDNA (see Supplementary Data File 4) harbouring homology arms to the digested XhoI-NotI pyCAG-hygromycin vector. Mutated $Mettl9$-FLAG dsDNAs encoded one of the following amino acid substitutions: either N35Q or N35Q;N86Q, where the choice of Glutamine (Q) is widely accepted in the glycobiology field[180,181]. Another construct was engineered with a mutated Signal peptide (SP*) encoding (R2A;W7A;C9A;S11A), which abolishes the positive charge of the N domain (R2) and the polar C domain (C9;S11), and that mildly affects the central hydrophobic region (W7) (see Results). 80 000 E14 mESCs (WT) were transfected in

suspension with 500 ng of each construct by using Lipofectamine™ 3000 (Thermo Fisher Scientific, #L3000001). After 48 hrs transfected cells were harvested and protein extracts were prepared as for total proteomics and analysed by Western Blot.

### Molecular complex predictions

The structures of the heterodimeric protein complexes were predicted using AlphaFold (2.3.2)[87] in multimer mode, using the reference protein sequences obtained from Uniprot[182] (METTL9: Q9EPL4; STMN1: P54227; RAB2a: P53994; RAB7a: P51150). To estimate either the structural conservation of METTL9 between mouse and Xenopus (Mettl9.L: Q7ZXA4), or the impact of the D151K and G153R mutations (METTL9-CatD) on the folding of mouse METTL9, each structure was predicted using AlphaFold in monomer mode and superimposed with TM-align[183]. The comparative analysis metrics (i.e. Root Mean Squared Deviations, Root Mean Squared Fluctuations and absolute deviations of the pseudo-torsion angles) were calculated using the R package bio3d[184]. The structures were rendered with Pymol Molecular Graphics System (Version 1.8, Schrödinger, LLC), and are available as Pymol session files in Supplementary Data 6.

### Data parsing and visualisation

Data formatting, statistical tests and visualization throughout this work were obtained using the R packages tidyverse[185], ggpubr (rpkgs.datanovia.com/ggpubr/) and cowplot (github.com/wilkelab/cowplot). Visualisation of the top enriched GO terms was performed using the dotplot, cnetplot and treeplot functions from the enrichplot package (Bioconductor[186]).

### Reporting summary

Further information on research design is available in the Nature Portfolio Reporting Summary linked to this article.

## Data availability

Raw RNA sequencing data from mESCs and X. laevis experiments were submitted to the SRA archive with project identifiers PRJNA1111296 and PRJNA1111433, respectively. Long-read whole genome sequences of the parental cell line and $Mettl9^{KO}$ clones #88 and #90 are deposited under SRA project PRJNA1242812. The mass spectrometry proteomics data (both raw and protein groups tables) have been deposited to the ProteomeXchange Consortium[187] via the PRIDE[188] partner repository with the dataset identifier PXD053437. Source data are provided with this paper as a Source Data File, for each graph and Figure panel shown in Main and Supplementary Figs. Mouse brain scRNA-seq data in Supplementary Fig. 1c were generated by re-analysis of dataset GSE116470. Statistics and Reproducibility. Statistical tests were unpaired, unless explicitly stated, and performed on N independent biological replicates (as detailed in each Fig. legend). The boxplots in Figs. 4g and 5i and in Supplementary Figs. 1a, 1b, 2b, 7c, 8e, 9h and 10a display the lower (Q1) and upper (Q3) quartiles at the ends of each box, with the thick line indicating the median of the distribution. The whiskers extend to the range between $Q1 − 1.5 \times IQR$ and $Q3 + 1.5 \times IQR$, where IQR (interquartile range) is the difference between Q3 and Q1. Data points outside this range are shown individually, representing potential outliers. Images shown in Fig. 1 c and Supplementary Fig. 2d are representative of WISH staining performed on at least N = 30 X. laevis embryos, for each stage shown. For each blot, gel, microscopy image and graph the underlying data (e.g. uncropped image or tables) are provided in Source Data file. Source data are provided with this paper.

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

## Acknowledgements

We thank the members of the IIT Genomics facility (Diego Vozzi, Yeraldin Chiquinquira Castillo De Spelorzi, and Edoardo Henzen), the HPC Team (Sergio Decherchi, Alessandro Parodi and Mattia Pini,) and the imaging facility (Michele Oneto and Marco Scotto) for their support to the research activities of the project. We are indebted to all the members of the RNA Initiative@IIT community for nurturing a stimulating scientific environment. The pyCAG-Hygromycin plasmid was kindly gifted by Ian Chambers' Lab, in Edinburgh. We greatly thank Professor M. Andreazzoli, E. Ferraro and M. Ori for advice and reagents for some of the procedures. This study makes use of data generated by the DECIPHER community. We are very grateful to Dr Julia Foreman, Dr Rachel Irving, Dr Vani Jain, Dr Katherine Neas, Dr Soo-Mi Park, Dr Alison Ross and Dr Vinod Varghese for kindly providing feedback on the clinical data. A full list of centres that contributed to the generation of the data is available from deciphergenomics.org/about/stats and via email from contact@deciphergenomics.org. DECIPHER is hosted by EMBL-EBI and funding for the DECIPHER project was provided by the Wellcome Trust [grant number WT223718/Z/21/Z]. The project was supported with intramural IIT funding (A. Codino, L.S., S. Gustincich and L.P.). IB and SLL were funded by the Cancer Research UK (grant reference RG86786) and by the Joseph Mitchell Fund. E.C. and F.C. were supported by the PRIN grant #2022M95RC7 from the Italian Ministry of University and Research (MUR) and the Tuscany Health Ecosystem - THE grant from MUR.

## Author contributions

A. Codino designed, performed, and analysed the experiments and wrote the manuscript. L.S. contributed to the execution of neural differentiation protocols, immunofluorescence and molecular cloning experiments. C.O. performed the manipulations of *Xenopus laevis* embryos, with the help and under the supervision of R.V.A. Cuomo performed the mass spectrometry analysis for proteomics experiments. H.S.R. performed the in vitro methyltransferase assays. M.P. and N.M. performed the acidic hydrolysis experiments for the detection of 1MH, 3MH, and His in NSCs extracts, under the supervision of S. Girotto and R.S. S.L. performed the experiments for the generation of the *Mettl9*^KO mESC line under the supervision of I.B. E.C. performed some of the neural differentiation experiments under the supervision of F.C. P.B. supervised the microscopy analysis. R.R. co-supervised A. Codino in the RUSH and digitonin-IF experiments. A.J.B., S. Gustincich and T.K. provided help and support for the molecular biology techniques of the project. L.P. conceived and supervised the study, performed the bioinformatics analysis and wrote the manuscript. All the authors revised and approved the manuscript.

## Competing interests

T.K. is a co-founder of Abcam Plc and Storm Therapeutics Ltd, Cambridge, UK. The other authors declare no competing interests.
