## [Transparent Peer Review file · Nature Communications]

METTL9 sustains vertebrate neural development primarily via non-catalytic functions

Corresponding Author: Dr Luca Pandolfini

Version 0:

Reviewer comments:

Reviewer #1

(Remarks to the Author)

The authors investigate the role of Mettl9 in neural development by induced mNSC/mNPC from mES and *X. laevis* embryos. They found that neural fate specification is impaired when deplete METTL9. The authors proposed that METTL9 regulates neurogenesis via Golgi apparatus in a catalytically independent manner. However, SLC39As, known substrates of METTL9, are zinc transporters of ER and Golgi (PMID 33563959, 34218407 and 37015930). The phenotype here could also be explained by the known substrates. Therefore, careful validation and comparison between the new and known interactors should be conducted. Unfortunately, the study ignores this key point. The animal models are intriguing but preliminary, especially in the connection to the proposed new mechanism. Much of the data is of poor quality and should be organized more appropriately, and the mechanism requires more evidence. In my view, this version is not suitable for publication.

Major concerns:

1. For IP-MS identifying interacting proteins, the authors used quite excessive amount of material (9x 10 cm plates were spitted into two groups: IgG control and FLAG IP), however, there is no description how much was used for the subsequent sample preparation and MS analysis. The authors should include the Coomassie brilliant blue or silver staining (together with IgG control) to show the quality of the IP complex and estimate the amount used in the next step. Only blotting METTL9 itself of IP (Fig. S6A) is not enough for IP-MS quality control, they should also blot CANX/SLC39As as positive controls. GFP-METTL9 immunoprecipitation and MS analysis was nicely performed before (PMID: 31036863), and clearly enriched for SLC39As, which should be set as a standard for this study.
2. The authors proposed METTL9 could bind STMN1 and RAB2/RAB7, and function in a catalytically independent fashion. But the data provided is not solid at all. Alpha-fold predication can only be used as supplementary data. The authors should also provide more controls in cell-based experiments, like co-IP between METTL9 (WT and catalytically dead) and STMN1/RAB. The authors should also check if the known substrates were enriched in their IP material, and rank the interaction strengths. The authors should also provide data demonstrating whether the binding of STMN1/RAB affects the enzymatic activity of METTL9 on known substrates.
3. The authors should also elucidate the mechanism more clearly: If enzymatic activity needs to be excluded, they should parallelly introduce METTL9 WT and catalytically dead to METTL9-depleted cells and compare the biological and molecular phenotype. Also, the authors didn't provide enzymatic function of METTL9 in this study, so the title shouldn't be "METTL9 sustains vertebrate neural development via both catalytic and non-catalytic functions". It would be clearer if they only claim to identify new binding proteins of METTL9, and how these interactions impact the function of METTL9.
4. Fig. 3G (Mettl9Deg-DIV5) showed different genes compared with Fig. 2I, and the related RNA-seq showed poor correlation between replicates (very large padj), and thus the comparison of DEGs among different cell lines was meaningless (Fig. 4C). The same problem exists in Fig. S3E (Mettl9Deg-DIV10). The reproducibility of Mass-Spec of DIV5 Mettl9Deg appears to be poor (Fig. S5A).
5. I'm confused that on Page 7 Line 24 they state "Overall, these data suggest, that residual METTL9 catalytic activity in the Mettl9Deg line might be sufficient to sustain neural commitment of mESCs.", while the subtitle is "Acute depletion of METTL9 from mESCs via DEGRON partially mimics METTL9 constitutive depletion". The two statements are contradictory

and can't be both right.

Minor Concerns:

1. Page 9 Line 24, "and found 10 proteins significantly down-regulated and 24 up-regulated over the Ctrl (DMSO-treated samples; adjusted p value < 0.05) (Supplemental Fig. S5A, B; Supplemental Table S3)." The adjusted p value is different from the figure legend and Sup Table. Where the statement is "GO analysis on the misregulated proteome (154 proteins with adjusted p-value < 0.1)". The authors should also describe the criteria for picking up the "154" proteins. Circle size indicating protein numbers in Fig. 5C is unproportioned.
2. The quality of WB in Fig. 2C is very poor.
3. Page 10 Line 10: "To this end, we inhibited the first biosynthetic step of N-linked glycosylation in the ER by supplying tunicamycin49–51 in live mESCs: protein extracts were analyzed by WB, which revealed the complete depletion of the two highest METTL9 bands (Fig. 5E)." It looks like the opposite. Remove the glycosylation part if it has no impact on METTL9 function.
4. Page 11 Line7: should be "Fig. 6A"
5. Some recent researches on METTL9 should be referenced and discussed, particularly different substrates may or may not be involve in (PMID: 34562450, 37015930, 37522633, 37398635, 38017014). The authors should have a comprehensive understanding of prior works before concluding their study.

Reviewer #2

(Remarks to the Author)

The manuscript by Codino et al, reveals that the METTL9 methyltransferase known to be linked to cancer biology plays an important role during embryonic development in neurogenesis.

Here the authors show that Mettl9 is highly expressed during neural differentiation in the *Xenopus* embryo and in mESCs. They then show that Mettl9 knock-down in the frog alters neurogenesis and that its constitutive depletion impairs mESCs to acquire a neural fate and generate differentiated neural cells. To dissect Mettl9 role during neural development, they generated and assessed by RNA-seq the ability to undergo neural differentiation of two additional mESC lines (an inducible METTL9-DEGRON, and a line endogenously expressing a catalytically inactive protein). They also profiled gene expression in mettl9 *Xenopus* tailbud morphants. The results obtained confirm the implication of Mettl9 in vertebrate neurogenesis and suggest catalytic-dependent but also catalytic-independent functions. By characterizing the proteome of METTL9-DEG NSCs, they obtained evidence that METTL9 carries out its function through modulation of the secretory pathway. They further show that METTL9 co-localizes with the Golgi, that its depletion leads to Golgi fragmentation in NSCs and the downregulation of Golgi-related mRNAs and proteins in neuralized *Xenopus* animal caps.

It is an important study in developmental neurobiology, as it establishes METTL9, currently only known to be linked to cancer biology, as a conserved key regulator of vertebrate neurogenesis. It is also of significance as it highlights that METTL9 likely exerts significant roles independently of its catalytic activity and that it contributes to the Golgi function and the secretory system. This is an interesting observation given that Golgi fragmentation has been observed in neurodegenerative diseases.

The studies are comprehensive, using multi-species paradigms: the data are of high quality and the conclusions convincing.

Some major points that could be addressed to further strengthen the study:

- 1) Page3, line 39, Figure 1C: the authors state that Mettl9 in *Xenopus* is expressed in neural crest cells, based on the in situ hybridization analysis of NF31 tadpole embryos. The provided picture shows staining in the branchial arches. As branchial arches consist of ectoderm, mesoderm, and endoderm, it is more appropriate to label the image "branchial arches" rather than "neural crest". The authors could also provide horizontal sections of the head of a tadpole to show more convincingly that Mettl9 is expressed in neural crest derivatives.
- 2) To assess whether Mettl9KO and Mettl9CatD mESC lines can differentiate into NSCs, the authors performed immunostainings with a nestin antibody. In Mettl9 KO mESCs, the staining is quantified (figure 2F), which is not the case for the Mettl9CatD (figure S4B). No staining is shown for the Mettl9Deg mESC line. Consistently showing such stainings and quantifications would provide the reader with a mean to reliably compare the phenotypes of the three mESC lines together with the wild-type condition but also between themselves. It may serve as a readout to demonstrate and quantify the in(ability) of these lines to generate neural cells.
- 3) To further investigate the requirement of mettl9 for neurogenesis in vivo, the authors may try to perform a rescue of the mettl9 MO experiment in *Xenopus* (using i.e. ElrC as a readout) by coinjection of mRNA encoding Mettl9 WT (and perhaps also Mettl9CatD to further provide evidence that its action is independent of its catalytic activity).

Minor points:

- 4) Sup Fig 1F: "ld" abbreviation is missing in the legend

5) Page 7, line 11-12. The authors state based on the results obtained using the Mettl9DEG NSCs that “the downregulation of neural-related genes as well as the up-regulation of metabolic very likely represented bona fide direct effects of METTL9 depletion”. Can the authors really conclude that it is a direct effect?

6) The cumulative plots in Figure S6F is difficult to interpret for non-expert readers. The authors could provide an expanded explanation in the associated legend.

7) In Fig S6, panel E, the colors used in the graph do not correspond to the indicated color code (grey?). Annotation of the Y axis is also missing.

8) Page 10. In the sentence “Consistently, METTL9 expression pattern drastically changed in DIV6 NSCs after a short treatment with golgicide”, “expression pattern” could be replaced by “subcellular distribution”.

Reviewer #3

(Remarks to the Author)

Reviewer #4

(Remarks to the Author)

In this article, the authors investigated METTL9, an enzyme responsible for catalyzing histidine N1-methylation. They provide compelling evidence of METTL9's role in the development of the nervous system in both *Xenopus* and mouse models. However, a key limitation of the study lies in the incomplete understanding of METTL9's cellular function. While proteomic and transcriptomic analyses, along with the observed Golgi dispersion in knockout (KO) models, were utilized, they fall short of definitively demonstrating METTL9's involvement in the secretory pathway.

A significant unresolved issue (in previous articles as here) pertains to the precise localization of METTL9 within neurons. Specifically, it remains unclear whether METTL9, which carried a signal peptide, resides in the lumen of the endoplasmic reticulum (ER) and Golgi apparatus, in the cytosol, or both. This distinction is crucial, as it would suggest that the cytosolic interaction partners identified in the study may not be true physiological partners. Furthermore, the substrates and biological functions of METTL9 would differ considerably depending on whether it operates in the Golgi lumen or the cytosol.

The authors should also directly assess the secretory function of METTL9 in their KO cells. This could be effectively accomplished using the RUSH assay, which would provide clearer insights into whether METTL9 is involved in the secretory pathway and further clarify its cellular role.

Reviewer #5

(Remarks to the Author)

The manuscript by Pandolfini and collaborators studies the role of METTL9 in neural development. They found that the loss of Mettl9 essentially impairs differentiation at least of the ventral neuronal lineage which is both delayed and misrouted in other lineages. Then the authors proposed that the main effect that METTL9 is exerting is through regulation of its interactors within the secretory pathway to the Golgi leading to the maintenance of this latter organelle. I found the topic and part of the results really intriguing. However, I think that the manuscript in this present form is both misleading and difficult to follow. Moreover, despite some exciting data, I believe that it is (i) very preliminary (many hypotheses are of interested but not demonstrated) and (ii) based on not-solid-enough approaches (e.i. degron line). Thus, I suggest to reorganize the manuscript to underline properly the proteomic data and follow this flow for the story and provide data from newly designed experiments to functionally challenge the hypothesis (e.g. independency of catalytic activity).

Below I listed some points that I noted in the manuscript in this present form:

- Fig S2B: please show the sequences that you get in the mutants with the “new” reading frames.
- A better WT control, than the parent line, would be a clone which underwent the same passages of the mutant ones which resulted mutation free.
- An off target analysis of the ESCs should be provided
- Transcriptomic data from the two KO clones should be independently indicated with the indication of the unique and common DEGs.

- Uninduced Mettl9-Deg line (i.e. with Mettl9 expressed) should be tested for the METTL9 protein functionality: for instance, methylation of targets containing the H[ANGST]H motif, in order to confirm that the chimeric METTL9 protein is equal to WT.
- The experiment using degron version of METTL9 are theoretically elegant and orthogonal to the complete KO. However, data pointed out that the model is basically not working properly. In fact, the transcriptomic data of Mettl9-Deg NSCs continuously supplied the dTAGV-1 drug is puzzling: only few DEGs retrieved in comparison with the full KO line. The authors then explained this with residual protein activity that indeed leads to the presence of 1MH level in the degron degraded cells compared to the DMSO treated (via bulk analysis). Then the question is: how these results are relevant?
- Complementation assay in MO-treated embryos with either WT or CatD Mettl9 should indicate the requirement of the enzymatic activity of Mettl9 for neural development in vivo.
- If the Mettl9 degradation is leaving residual protein able to essentially blunt the transcriptomic effect after prolonged treatment, why we should rely on data deriving from a 2 day treatment (Fig.5)?
- Are the differentially enriched proteins found in the degron line, also affected in the full KO?

Version 1:

Reviewer comments:

Reviewer #1

(Remarks to the Author)

The authors have addressed all of my concerns. The revised version is much improved. However, it remains too comprehensive and detail-heavy, which limits readability. I recommend simplifying the structure and clearly highlighting the key point -non-catalytic function - prior to publication.

Reviewer #2

(Remarks to the Author)

The concerns we raised have been satisfactorily addressed, and we now endorse the publication of this manuscript.

Reviewer #3

(Remarks to the Author)

Reviewer #4

(Remarks to the Author)

The authors have satisfactorily answered the reviewers' comments.

Reviewer #5

(Remarks to the Author)

I found the revised version of the manuscript to be significantly improved in terms of clarity, coherence, and overall structure. The authors have made a commendable effort in addressing the main concerns raised during the previous review round. The manuscript now presents its objectives and findings in a more precise and accessible manner, which greatly enhances its readability and scientific impact.

Both the strengths and the limitations of the work are now more explicitly acknowledged, which demonstrates a mature and balanced approach to the research. While some aspects still remain open or insufficiently explored, the integration of new methodologies and analyses has provided a more robust foundation for the results. These additions have not only reinforced the main conclusions but also contributed to a clearer and more convincing set of take-home messages.

Overall, the manuscript has advanced considerably, and although certain points may benefit from further clarification in future work, the current version marks a substantial step forward in terms of scientific quality and communicative effectiveness.

Reviewer #1 (Remarks to the Author)

The authors investigate the role of Mettl9 in neural development by induced mNSC/mNPC from mES and X. laevis embryos. They found that neural fate specification is impaired when deplete METTL9. The authors proposed that METTL9 regulates neurogenesis via Golgi apparatus in a catalytically independent manner. However, SLC39As, known substrates of METTL9, are zinc transporters of ER and Golgi (PMID 33563959, 34218407 and 37015930). The phenotype here could also be explained by the known substrates. Therefore, careful validation and comparison between the new and known interactors should be conducted. Unfortunately, the study ignores this key point. The animal models are intriguing but preliminary, especially in the connection to the proposed new mechanism. Much of the data is of poor quality and should be organized more appropriately, and the mechanism requires more evidence. In my view, this version is not suitable for publication.

We are sorry that the previous version of the manuscript did not meet the Reviewer's expectations. During the revision we took the utmost care to consolidate many of the key points of our work, and we extensively edited the manuscript to improve its logical flow. We now also included SLC39A7 in the model figure (**Fig. 8g**) and mentioned the previous relevant work concerning METTL9 substrates and/or known interactors. We have also explained better throughout the text the possible relative contribution of the catalytic functions on neurogenesis. We think that the effort made to respond to the many points raised by Reviewer has substantially changed the quality of the paper and we really thank the reviewer for helping us to better address the scope of the numerous new experiments performed in the revision.

Major concerns:

1^[a]. For IP-MS identifying interacting proteins, the authors used quite excessive amount of material (9x 10 cm plates were spitted into two groups: IgG control and FLAG IP), however, there is no description how much was used for the subsequent sample preparation and MS analysis.

We would like to clarify that 9x 10 cm plates have been used for 8 different IP reactions, so approximately **1x 10 cm plate (around 15 x 10⁶ cells), corresponding to 1681 ug were used as input for each IP (as now better described in the Methods section)**. The amount of IP starting material is dependent on the expression level of the target protein: in this instance METTL9-FLAG is not over-expressed but it is present at endogenous levels; moreover, it also depends on the size of the cells because neural stem cells are very small (around 8 μm) and with low protein content, thus they might require higher cell numbers compared to the much bigger HEK293t (11-15 μm) which are almost double in size. To justify our experimental design strategy, here **we also provide references of papers which used similar, if not higher, amounts of proteins for each IP, from mouse or human ESC/NSCs/brain tissue:**

(Basu et al. 2020) *Genes & Dev*, 2 mg from mouse embryonic Cortex lysate
(Pintacuda et al. 2023) *Cell Genomics*, 1-2 mg or 15x10⁶ iPSC-derived neurons
(Liu et al. 2016) *J Bio Chem*, 1mg mouse brain lysate
(Morrow et al. 2020) *Cell Stem Cell*, 20x10⁶ neural stem cells
(Uzbas and O'Neill 2023) *Bio Protoc*, 4-5mg hiPS-derived neural cells
(Urbán et al. 2016) *Science*, 2x10⁸ neural stem cells

For the above reasons we believe that the IP-MS has been performed in a canonical way, at least for the NSCs field.

^[1b] The authors should include the Coomassie brilliant blue or silver staining (together with IgG control) to show the quality of the IP complex and estimate the amount used in the next step.

In order to address the Reviewer's concern and to show that the initial amount of material used for the IP was not excessive, but in line with good practice for semi-quantitative mass spectrometry, we have included **here the Coomassie gel of a representative IP** (corresponding to the same samples loaded in the WB shown in Supplementary Fig. 9a). **In all the loaded IP, no protein is visible and, more importantly, no IgG bands are detected at 50 and 25 kDa** ("prec." stands for precipitated with Acetone). After showing this Coomassie gel quality control to the MassSpec facility we proceeded with the sample preparation for Mass Spectrometry using the PreOmics kit (as specified in the methods). **This kit is designed to work with 1-100 µg of protein pellets**. We used 90% of the eluted IP as starting material for the PreOmics. At the end of this standard protocol, the peptides are directly loaded into the machine following standard procedures of the MassSpec Facility.

^[1c] Only blotting METTL9 itself of IP (Fig. S6A) is not enough for IP-MS quality control, they should also blot CANX/SLC39As as positive controls. GFP-METTL9 immunoprecipitation and MS analysis was nicely performed before (PMID: 31036863), and clearly enriched for SLC39As, which should be set as a standard for this study.

We thank the Reviewer for pointing out this oversight in the previous manuscript. In principle, it might not be straightforward to perform a direct comparison with (Ignatova et al. 2019). In fact, **our data deal with a very different cellular model (neural stem cell cells versus cancer cell lines) and target different baits: METTL9 expression at physiological levels harbouring a C-terminal long, flexible linker followed by a Degron-FLAG tag (total molecular weight of the tag 15.7 kDa), versus over-expression with a bulky N-terminal GFP tag (molecular weight of the tag 28 kDa)**. In addition, the relative extraction/IP buffers and experimental conditions may favour more stable (e.g. protein binding partners) versus transient interactions (e.g. enzyme+substrate). In agreement with this, most of the peptides belonging to the published METTL9 substrates/interactors, namely SLC39A7, SLC39A10, ARMC6, ITPRIPL2 and SLC39A6 were detected *neither in the input nor in the IP spectra* in DIV4 NSCs, hindering a direct comparison. Notwithstanding this, **the key METTL9 interactors Canx (log2Enrichment 0.95, p.adj=0.01), and Faf2 (log2Enrichment 0.57, p.adj=0.04) were also confirmed in our IP-MS experiment in mouse NPCs, thus**

supporting the strength of our data. This is now clearly stated in the text (Figure 6a Legend) and Figure 6a. Finally, in the revised version we also experimentally validated a few interactions by co-IP experiments, using another antibody (Anti-HA), thus confirming 2 biological relevant interactors in our system (STMN1 and RAB2a; **Fig. 6c, d**). Taking together all these data, we would like to defend the solidity of our METTL9 IP-MS interactome.

^[2^a] The authors proposed METTL9 could bind STMN1 and RAB2/RAB7, and function in a catalytically independent fashion. But the data provided is not solid at all. Alpha-fold predication can only be used as supplementary data.

As documented in the replies below, we addressed the important point highlighted by the Reviewer. Also, we **moved the AlphaFold prediction regarding RAB2a-M9, old Fig. 6D, to the Supplementary Fig.9e**, as suggested by the Reviewer. We believe that the STMN1-M9 complex prediction should remain in main **Fig. 6e** because it is **now validated by the co-IP and backed by more mechanistic data (as detailed below)**, therefore it might be important for the reader to visualise this key interaction.

^[2^b] The authors should also provide more controls in cell-based experiments, like co-IP between METTL9 (WT and catalytically dead) and STMN1/RAB.

To address this important point indicated by the Reviewer, and to further strengthen the IP-MS data, we assessed some of the interactions detected by IP-MassSpec by **performing co-immunoprecipitation experiments in a cell-based system**. These experiments **confirmed that METTL9 interacts physically with STMN1 and with RAB2a (new Fig. 6 c,d) in mESCs**, using a different antibody (Anti-HA-beads). Importantly, these results also showed that these interactions are independent from the presence of the DEGRON tag, since the Mettl9 constructs for this assay only harbored a C-terminal FLAG tag. The same experiment was also performed using METTL9-CatD-FLAG, demonstrating that the interaction is not affected by the mutations suppressing the enzymatic activity.

^[2^c] The authors should also check if the known substrates were enriched in their IP material, and rank the interaction strengths.

Unfortunately, **we did not detect any known substrate** of METTL9's catalytic activity (see the reply to Reviewer #1 major point 1c), **but two important known interactors (i.e. CANX and FAF2) have now been highlighted in Fig. 6a and Text**. Log2 enrichment values are indicated in the plot in **Fig. 6a**, are also reported in Supplementary Data 3.

^[2^d] The authors should also provide data demonstrating whether the binding of STMN1/RAB affects the enzymatic activity of METTL9 on known substrates.

Prompted by the Reviewer's pertinent suggestion, **we have tested whether STMN1 affects METTL9 Catalytic activity *in vitro* and the results are now shown in Fig. 6f (and Supplementary Fig. 9c,d)**. Importantly, we found that **STMN1-HA competes with a known METTL9 substrate in a dose-dependent manner**. Consequently, we also verified by co-IP that STMN1-METTL9 interaction is independent of METTL9 catalytic activity (as also reported in the reply to a previous point.) Consistently, **STMN1-HA (or HA-RAB2a) was able to immunoprecipitate METTL9-CatD (results are now in Supplementary Fig. 9 j,k)**. This is further supported by our new *in silico* predictions (**Supplementary Fig. 9 g,h,i**), which show the overall similarity between METTL9-WT and METTL9-CatD structure, particularly at the interface with STMN1/RAB2a. Overall the new pieces of evidence indicate that STMN1 might exert important regulatory activities of METTL9 function and that this regulation could be

reciprocal: even more relevant perhaps, the binding of METTL9 to STMN1 may contribute to modulate STMN1 cellular functions, by modulating tubulin binding and microtubules growth.

3. The authors should also elucidate the mechanism more clearly: If enzymatic activity needs to be excluded, they should parallelly introduce METTL9 WT and catalytically dead to METTL9-depleted cells and compare the biological and molecular phenotype.

During the revision, we tried to re-introduce WT or CatD METTL9 in *Mettl9*^{KO} mESCs and assess whether the phenotype (e.g. Golgi fragmentation) was at least partially recovered. We tried two independent strategies, including the use of a transient transfection with pyCAG plasmids and the generation of stable cell lines using the PiggyBac transposase system, but both yielded a very low number of METTL9-WT/CatD positive cells surviving until DIV5/6, and therefore the results concerning possible rescue effects by either construct were inconclusive. Both complementation systems in mESC (i.e. pyCAG plasmid and PiggyBac) share the fact that they re-express excessively high amounts of METTL9, due to their strong promoters (CAG and EF1a, which however are among the few promoters that are not silenced during mESCs differentiation to a neural fate). Therefore, cells might be counter-selected during the differentiation process. In support to this hypothesis, our unpublished data in *Xenopus* show that injecting a high dose of *mettl9*-mRNA into wild-type embryos (i.e. 500 pg versus 50-200 pg used for the rescue, as described below) phenocopies the effects observed upon *mettl9* knock-down (see figure below):

This indicates that it is important to maintain adequate levels of *Mettl9* for neural development. Therefore, the complementation in mouse NSCs would require more sophisticated and time-consuming genetic systems, which we believe would go beyond the scope of this revision. However, in order to tackle the Reviewer's crucial point, we conducted these experiments *in vivo* using *Xenopus laevis* embryos, a system allowing **fine tuning of *mettl9* dosage**. In fact, **re-expression in *Xenopus* can be finely controlled by microinjecting exact amounts of *in vitro* transcribed mRNA**. Embryos were co-injected with SPL-MO (knock-down) and either wild-type or CatD *mettl9* mRNA. Both conditions resulted in a comparable rescue of the neural phenotype, as indicated by *elrC* marker expression (Fig. 8 e,f). These results support two key conclusions: (1) *mettl9* is essential for neurogenesis, as co-injecting its mRNA rescued the phenotype in *mettl9*-MO embryos, and (2) the catalytic activity of *Mettl9* does not have a primary role in this process, since the catalytically inactive *mettl9* mRNA was equally effective in restoring *elrC* expression. We regret that it has not been possible to verify that even in *Xenopus* *Mettl9* misregulation affects the stability of the Golgi. However, this study would have required the development of a too large set of genetic tools that are available for mouse but not for *Xenopus laevis*, making this control unfeasible for this revision.

^[3b] Also, the authors didn't provide enzymatic function of METTL9 in this study, so the title shouldn't be "METTL9 sustains vertebrate neural development via both catalytic and non-catalytic functions". It would be clearer if they only claim to identify new binding proteins of METTL9, and how these interactions impact the function of METTL9.

We have updated the title, which better reflects the message of the manuscript, in the light of the Reviewer's comment and the novel mechanistic and functional data. Our main indications supporting a secondary role for METTL9 catalytic activity in the secretory pathway during neural differentiation are that: 1) **the molecular changes observed in the RNA sequencing and 2) the Golgi fragmentation effects** are consistent between CatD and KO cells, albeit being **much milder in the CatD**. Moreover, we have incorporated now **the rescue experiments in *Xenopus*, with CatD *mettl9* and the *in vitro* methylation assays showing that one neural METTL9-interactor (STMN1) interferes with METTL9 catalytic activity *in vitro***. We agree with the Reviewer that the previous version of the manuscript was often unclear and presented this subject in a confusing way. We have now extensively revised the text and improved its logical flow, besides **providing more experimental evidence on the secondary role of Mettl9 Catalytic activity for neural development**.

4. Fig. 3G (Mettl9Deg-DIV5) showed different genes compared with Fig. 2I ...

We would like to clarify to the Reviewer that the genes represented in the **Fig. 2i** and **Fig. 3g** exemplify the Neural-related Gene Ontologies terms affected in both KO and Deg cell lines. This point is better shown by the global analysis presented in **Supplementary Fig. 10a**, which more convincingly and quantitatively shows a consistent gene mis-regulation (albeit much milder in the Degron line) across the two experimental models. **Transcriptomic analysis in this study is performed with the aim of characterising the cell fate identity (and Gene Ontology is a proxy for this); so, we did not envisage to identify specific mRNAs consistently altered in different Mettl9 genetic systems**, but only Gene Ontology processes. We think this view is sustainable **since METTL9 is not a transcription factor and therefore it does not directly control the expression of selected genes**. Moreover, considering the differences between a full KO and a Degron (also further explained in Main Text and reply to Reviewer #2 minor 5 and Reviewer #5 point 6, 8 and 9), we did not expect nor pretended an identical regulation of genes at the mRNA level. The message that we want to convey without any overstatement is that **KO and Degron do perturb the same processes (such as those related to neurogenesis)**, and genes there represented are chosen as examples.

^[4b]... and the related RNA-seq showed poor correlation between replicates (very large padj), and thus the comparison of DEGs among different cell lines was meaningless (Fig. 4C). The same problem exists in Fig. S3E (Mettl9Deg-DIV10). The reproducibility of Mass-Spec of DIV5 Mettl9Deg appears to be poor (Fig. S5A).

We respectfully disagree with the Reviewer's conclusion that the lack of a strong separation of the samples in the plots above mentioned implies a poor reproducibility. The first few principal components obtained by this technique capture the directions of maximum variance, regardless of whether that variability is due to treatment, noise, or other factors. Therefore, a **lack of separation in PCA** might just mean that **the treatment effect is subtle in the Degron line (as it is indeed the case, as confirmed by multiple other lines of evidence, e.g. immunocytochemistry with Nestin and golgi fragmentation)**, not that the data is noisy or unreproducible. As a proof of this, in the case of the KO (where the effect is much stronger), the genotype prevails (as shown by the proportion of variance explained in **Supplementary Fig.**

4a,f). Furthermore, **the new clustering data at a single sample level (Supplementary Fig. 4b,g) show** that indeed the **technical and experimental variability due to cell differentiation is very low across multiple replicates and clones/lines sharing the same genotype**. This in our opinion contributes to support the very high level of reproducibility of our experiments.

5. *I'm confused that on Page 7 Line 24 they state "Overall, these data suggest, that residual METTL9 catalytic activity in the Mettl9Deg line might be sufficient to sustain neural commitment of mESCs.", while the subtitle is "Acute depletion of METTL9 from mESCs via DEGRON partially mimics METTL9 constitutive depletion". The two statements are contradictory and can't be both right.*

We agree with the Reviewer that the title and the conclusive sentence of the paragraph could sound as contradictory. During the extensive revision of the manuscript, **we have now edited both the title and last sentence of this paragraph**, and we hope that the text is now clearer. We edited the title of this paragraph writing "**only** partially mimics" to **highlight the difference with the KO in terms of phenotype**. Throughout the entire paragraph and particularly in its **conclusion**, we have now explained better that the residual METTL9 protein in the dTAG could be responsible for the milder phenotype: "*Overall, these data suggest that the low levels of residual METTL9 protein in the dTAG^V-1-treated Mettl9^{Deg} line might be sufficient to sustain neural commitment of mESCs **either via METTL9-dependent catalytic functions and/or through other non-catalytic activities***". We hope that this has enhanced the overall logical flow of the manuscript.

Minor Concerns:

1. *Page 9 Line 24, "and found 10 proteins significantly down-regulated and 24 up-regulated over the Ctrl (DMSO-treated samples; adjusted p value < 0.05) (Supplemental Fig. S5A, B; Supplemental Table S3)." The adjusted p value is different from the figure legend and Sup Table.*

We sincerely thank the Reviewer for pointing out a serious oversight. In the previous manuscript version, in the volcano plot we correctly showed the up- and down-regulated proteins with q value < 0.1 in red (24 proteins) and blue (10 proteins), respectively (Fig. 5b). In addition, we highlighted those with q values < 0.05 by labelling them with their Gene symbols (11 up- and 2 down-regulated, respectively). In the text, however, we indicated the wrong q value, as the numbers were relative to q value < 0.1, instead of q value < 0.05. This does not affect the relative discussion in this paragraph of the manuscript. However, since these numbers mainly serve to provide the reader with an order of magnitude for the proteome modulation, to avoid confusion **in the current version, we decided to write in the text the number of proteins that result mis-regulated with q value < 0.05 and to show them in Fig. 5b accordingly**.

^[1b] *Where the statement is "GO analysis on the misregulated proteome (154 proteins with adjusted p-value < 0.1)". The authors should also describe the criteria for picking up the "154" proteins.*

While addressing the Reviewer remark we have found and amended a typo concerning the q value threshold, and we listed the protein numbers for the GO analysis (q value < 0.2 instead of

0.1, yielding 115 up- and 63 down-regulated proteins used as an input, which form a total of 154 proteins after assigning them to the GO term database). We also agree with the Reviewer that, without an explicit explanation, the use of a different q value threshold for the GO analysis might sound somewhat cryptic to the reader. For this reason, **we have modified the new version of the text in “GO analysis on the misregulated proteome (encompassing a larger set of 115 up-regulated proteins and 63 down-regulated proteins with a less stringent q value threshold of 0.2)”**. The less stringent threshold of q value < 0.2 has been used to broaden the set of proteins employed for the **Gene Ontology analysis, which requires a larger subset of genes/proteins to be biologically meaningful (as opposed to the very few with q value < 0.05)**. Relaxing the threshold increases the chance of detecting true positives (higher sensitivity), at the cost of possibly including more false positives (lower specificity) in the input of the GO analysis: however, since the statistical test challenges the null assumption that they are randomly sampled from the full set of quantified genes/proteins, **a more permissive threshold does not compromise our confidence on the ontology terms that result to be significantly enriched.**

[1c] Circle size indicating protein numbers in Fig. 5C is unproportioned.

We thank the reviewer for pointing out this plotting artifact, **which we amended in the current version, in Fig. 5c and all the others with the same issue.**

2. The quality of WB in Fig. 2C is very poor.

We acknowledge that the unprocessed image representing the Western blot in **Fig. 2c** is a bit noisy. Unfortunately, this **depends on the low cellular levels of the METTL9 protein and on the quality of the custom anti-METTL9 antibody used**. These factors are also among **the reasons why we decided to engineer a tagged METTL9-DEGRON-FLAG mESC line**. In our hands, **all commercially available antibodies that we tried performed even worse compared to our custom-made antibody, on mESCs and NSCs** (when they work at all). However, we believe that in our manuscript we have dedicated ample space to demonstrate that 1) METTL9 presents multiple bands corresponding to post-translational modifications (see also the replies below to other points from the Reviewer) and that 2) METTL9 levels are upregulated upon neural differentiation. For instance, this is shown by qPCR performed (**Fig. 2b**). Furthermore, main **Figure 1** and **Supplementary Fig. 2** show Mettl9 expression in the *Xenopus* developing nervous system; in **Supplementary Fig. 1a** we have highlighted *METTL9* expression during foetal human brain development, indicating the overall conservation of Mettl9 expression in the developing nervous system of vertebrates. Altogether we hope that this data will be sufficient to convince the Reviewer **that the quality of the METTL9 signal detected by this polyclonal antibody in WB does not undermine the message of the paper.**

3. Page 10 Line 10: “To this end, we inhibited the first biosynthetic step of N-linked glycosylation in the ER by supplying tunicamycin49–51 in live mESCs: protein extracts were analyzed by WB, which revealed the complete depletion of the two highest METTL9 bands (Fig. 5E).” It looks like the opposite. Remove the glycosylation part if it has no impact on METTL9 function.

We acknowledge the Reviewer for pointing out this mistake. **We have corrected the swapped labels in the new Supplementary Fig. 8h** and (DMSO + / -) and Tunicamycin (+/ -) are now

placed **in the correct order**. Moreover, to further validate these results, **we have added new data (Fig. 5f and Supplementary 8g) directly demonstrating that 2 METTL9 Asparagine residues (N35;N86) are N-Glycosylated and responsible for the higher METTL9 bands. We also introduced mutations** in the sequence corresponding to the N-terminal **METTL9 Signal Peptide** and observed the same result as for the double (N35Q;N86Q) mutant, suggesting that the **SP is necessary for directing METTL9 to the ER to start its N-glycosylation.**

4. Page 11 Line7: should be "Fig. 6A"

We acknowledge the Reviewer for noticing this mistake. **We have amended the text with "Fig. 6a"** instead of "b".

5. Some recent researches on METTL9 should be referenced and discussed, particularly different substrates may or may not be involve in (PMID: 34562450, 37015930, 37522633, 37398635, 38017014). The authors should have a comprehensive understanding of prior works before concluding their study.

We apologise for falling short of including these previous works. **We have incorporated these five research articles in the introduction of our manuscript**, highlighting them as a foundation of our current understanding of METTL9 function in other systems.

Reviewer #2 (Remarks to the Author)

The manuscript by Codino et al, reveals that the METTL9 methyltransferase known to be linked to cancer biology plays an important role during embryonic development in neurogenesis.

Here the authors show that Mettl9 is highly expressed during neural differentiation in the Xenopus embryo and in mESCs. They then show that Mettl9 knock-down in the frog alters neurogenesis and that its constitutive depletion impairs mESCs to acquire a neural fate and generate differentiated neural cells. To dissect Mettl9 role during neural development, they generated and assessed by RNA-seq the ability to undergo neural differentiation of two additional mESC lines (an inducible METTL9-DEGRON, and a line endogenously expressing a catalytically inactive protein). They also profiled gene expression in mettl9 Xenopus tailbud morphants. The results obtained confirm the implication of Mettl9 in vertebrate neurogenesis and suggest catalytic-dependent but also catalytic-independent functions. By characterizing the proteome of METTL9-DEG NSCs, they obtained evidence that METTL9 carries out its function through modulation of the secretory pathway. They further show that METTL9 co-localizes with the Golgi, that its depletion leads to Golgi fragmentation in NSCs and the downregulation of Golgi-related mRNAs and proteins in neuralized Xenopus animal caps.

It is an important study in developmental neurobiology, as it establishes METTL9, currently only known to be linked to cancer biology, as a conserved key regulator of vertebrate neurogenesis. It is also of significance as it highlights that METTL9 likely exerts significant roles independently of its catalytic activity and that it contributes to the Golgi function and the secretory system. This is an interesting observation given that Golgi fragmentation has been observed in neurodegenerative diseases.

The studies are comprehensive, using multi-species paradigms: the data are of high quality and the conclusions convincing.

We thank the Reviewer for the positive recognition of our work and the insightful comments and suggestions provided.

Some major points that could be addressed to further strengthen the study:

1) Page3, line 39, Figure 1C: the authors state that Mettl9 in Xenopus is expressed in neural crest cells, based on the in situ hybridization analysis of NF31 tadpole embryos. The provided picture shows staining in the branchial arches. As branchial arches consist of ectoderm, mesoderm, and endoderm, it is more appropriate to label the image "branchial arches" rather than "neural crest". The authors could also provide horizontal sections of the head of a tadpole to show more convincingly that Mettl9 is expressed in neural crest derivatives.

We agree with the reviewer that the previous description of *mettl9* expression pattern was not sufficient to conclusively support its expression in neural crest cells. Given the resolution of the image, **we relabelled the "neural crests" into "branchial arches"** as suggested. In addition, **we have provided additional WISH experiments showing the detailed expression pattern of *mettl9* in horizontal sections of NF30 Xenopus embryos (Supplementary Fig. 2e).** These data reveal *mettl9* expression in the core of the pharyngeal arches (I-IV), but not in endodermal or ectodermal tissues, consistent with the localization of the **neural crest cell marker *twist1***. The relative **Supplementary Fig. 2e**, Fig. legends and Methods have been updated accordingly.

2) To assess whether Mettl9KO and Mettl9CatD mESC lines can differentiate into NSCs, the authors performed immunostainings with a nestin antibody. In Mettl9 KO mESCs, the staining is quantified (figure 2F), which is not the case for the Mettl9CatD (figure S4B). No staining is shown for the Mettl9Deg mESC line. Consistently showing such stainings and quantifications

would provide the reader with a mean to reliably compare the phenotypes of the three mESC lines together with the wild-type condition but also between themselves. It may serve as a readout to demonstrate and quantify the in(ability) of these lines to generate neural cells.

We acknowledge the Reviewer for noticing this important oversight, as in the previous version of the manuscript we relied mainly on the transcriptomic alterations to describe the neural phenotype and we omitted the Nestin quantifications for the other 2 lines. In the **Supplementary Fig. 6b** and related Fig. legend, **we have added the quantification of NESTIN+ cells in CatD; in Supplementary Fig. 5f we provided a new IF staining for DMSO and dTAG^V-1 treated *Mettl9^{Deg}* NSCs together with their corresponding quantification.** Overall, since the counting showed only a slight trend, but no significant decrease, in the number of NESTIN+ cells in both the CatD and the Deg(+dTAG) lines, these data are consistent with our transcriptomic findings, documenting a very mild impact on neurogenesis and the lack of macroscopic neural phenotypes.

*3) To further investigate the requirement of *mettl9* for neurogenesis in vivo, the authors may try to perform a rescue of the *mettl9* MO experiment in *Xenopus* (using i.e. *ElrC* as a readout) by coinjection of mRNA encoding *Mettl9* WT (and perhaps also *Mettl9CatD* to further provide evidence that its action is independent of its catalytic activity).*

In line with this important comment from the Reviewer, **we have now strengthened our statements by providing more experimental evidence through rescue experiments *in vivo* (see Reviewer #1 major point 3), incorporated in main Fig. 8 e,f and relative Main Text.** These experiments confirmed 1) **the requirement of *Mettl9* for neurogenesis, as *mettl9*-wt mRNA co-injection in *mettl9*-MO embryos could partially rescue the phenotype and 2) the independency of its catalytic activity for neurogenesis,** since *mettl9*-catD mRNA is also able to rescue the neural phenotype (as revealed by *elrC* mRNA expression pattern), to the same extent of *mettl9*-wt.

Minor points:

4) Sup Fig 1F: "ld" abbreviation is missing in the legend

We thank the Reviewer for pointing out the previous inconsistency in the acronyms of the figure, namely the use of "ld" and "DII" to indicate the "dorsal blastopore lip" (dbl). The **Supplementary Fig. 2d and its legend have been amended.**

*5) Page 7, line 11-12. The authors state based on the results obtained using the *Mettl9*DEG NSCs that "the downregulation of neural-related genes as well as the up-regulation of metabolic very likely represented bona fide direct effects of METTL9 depletion". Can the authors really conclude that it is a direct effect?*

We agree with the Reviewer that, at the point where it was used, **the term "direct"** was inappropriate and **replaced it by "bona fide specific effects"**. We tried to plainly address this important point in the current revised text: indeed **the term "direct" is more appropriate for the perturbations obtained by the 48 hr dTAG treatment employed in the *Mettl9^{Deg}* proteomic experiment** (see also Reviewer #5 Reply 6, 8 and 9). On the contrary, in this case, the dTAG treatment was administered for 6 days, which is still useful to provide details of the more specific effects of *Mettl9* on neural differentiation (compared to the KO) but it does not represent an *acute* depletion (as for the 48 hrs dTAG). We have now clarified the conceptual distinction between the Degron and the KO lines, and their intrinsic advantages and drawbacks. Despite not achieving complete depletion, the *Mettl9^{Deg}* line is leveraged for acute depletion of the protein, which is one way to identify the *early* molecular consequences of protein loss and is therefore useful to give hints on the mechanistic effects of METTL9

phenotype. This is in contrast with the long-term consequences of constitutive METTL9 depletion observed in the KO (which are however important for other reasons): the effects in these cells better represents the phenotype that might occur *in vivo* (e.g. in human mutations), but they also highlight indirect and compensatory effects. In fact, the massive and macroscopical neuralisation defects of the *Mettl9*^{KO} cells imply that many molecular and cellular changes had occurred in those cells and (given the cascade of events during development) some of them might be secondary, only indirectly linked to other earlier perturbations due to METTL9 loss.

6) *The cumulative plots in Figure S6F is difficult to interpret for non-expert readers. The authors could provide an expanded explanation in the associated legend.*

We agree that the previous description accompanying these results was lacking both detail and clarity. Therefore, **we have explained in detail the cumulative plots in the legend to Supplementary Fig. 10a.** We state: *“Graphs tracking how the genes mis-regulated upon *Mettl9*^{KO} (represented in the four different colors) behave in the other mESC lines under study. The distribution of the (log2) fold changes for each gene group is shown by means of both cumulative distribution (top) and box plots (bottom). This analysis reveals that up- and down-regulated genes in *Mettl9*^{KO} are generally similarly affected (i.e. they display a concordant, statistically significant up- or down-regulation trend compared to control) also in *Mettl9*^{Deg} and *Mettl9*^{CatD}, albeit to a much smaller extent. For each of the four gene groups considered, the panel also shows its number (on the right of the corresponding boxplot line)”.*

7) *In Fig S6, panel E, the colors used in the graph do not correspond to the indicated color code (grey?). Annotation of the Y axis is also missing.*

We thank the Reviewer for pointing out this inconsistency: **both the color code and the Y axis of the Golgi quantifications have been amended in Supplementary Fig.11d** (counting graphs for CatD and degron) and they are now the same as in main **Figure 7d** (KO).

8) *Page 10. In the sentence “Consistently, METTL9 expression pattern drastically changed in DIV6 NSCs after a short treatment with golgicide”, “expression pattern” could be replaced by “subcellular distribution”.*

We thank the Reviewer for this appropriate suggestion. **We have amended this in the Main Text** within the paragraph *“METTL9 localises to the Golgi in mouse NSCs”.*

Reviewer #3 (Remarks to the Author)

We thank the Reviewer for their precious contribution to the evaluation of our work.

Reviewer #4 (Remarks to the Author):

In this article, the authors investigated METTL9, an enzyme responsible for catalyzing histidine N1-methylation. They provide compelling evidence of METTL9's role in the development of the nervous system in both Xenopus and mouse models. However, a key limitation of the study lies in the incomplete understanding of METTL9's cellular function. While proteomic and transcriptomic analyses, along with the observed Golgi dispersion in knockout (KO) models, were utilized, they fall short of definitively demonstrating METTL9's involvement in the secretory pathway.

[1] A significant unresolved issue (in previous articles as here) pertains to the precise localization of METTL9 within neurons. Specifically, it remains unclear whether METTL9, which carried a signal peptide, resides in the lumen of the endoplasmic reticulum (ER) and Golgi apparatus, in the cytosol, or both. This distinction is crucial, as it would suggest that the cytosolic interaction partners identified in the study may not be true physiological partners. Furthermore, the substrates and biological functions of METTL9 would differ considerably depending on whether it operates in the Golgi lumen or the cytosol.

We acknowledge the Reviewer for the constructive feedback on our manuscript.

We agree with the Reviewer that the subcellular localisation is crucial to consolidate our data, hypotheses and working model on METTL9 function. Therefore, following the Reviewer's indication **we investigated this point further: by performing Immunofluorescence (IF) under different cell permeabilization conditions (namely Triton- versus Digitonin-treatment)** we showed that **METTL9 signal is still visible when Golgi membranes are not permeabilised (as confirmed by lack of LCS signal in Digitonin-treated samples)**. This implies that **METTL9 localises on the peripheral Golgi side**, because the lumen is not accessible. This evidence is **in agreement with the protein interactome results**, which highlight many cytosolic and cytoskeletal protein amongst the top hits of the IP-mass spectrometry experiment. **We have incorporated this important piece of data into Fig. 5j** of the revised manuscript. The **model in Fig. 8g has been updated** accordingly.

[2] The authors should also directly assess the secretory function of METTL9 in their KO cells. This could be effectively accomplished using the RUSH assay, which would provide clearer insights into whether METTL9 is involved in the secretory pathway and further clarify its cellular role.

We acknowledge the Reviewer for the suggestion. Indeed, assessing the secretory function of Mettl9-KO NSCs can definitely broaden our knowledge of METTL9 function. We took advantage of the RUSH system (Boncompain et al. 2012), which represents the state-of-the-art methodology to follow trafficking dynamics, to **assess the impact of METTL9 loss on the secretory pathway. We generated wild-type and Mettl9^{KO} mESC cells stably expressing the streptavidin protein fused to an ER-resident hook together with a streptavidin binding protein-MannosidaseII-EGFP reporter construct**, and we monitored the cargo progression by **live confocal imaging in DIV6 Neural Stem Cells**, upon biotin administration. These experiments, **now included in the manuscript (Fig.7 a,b and Supplementary Fig. 11 a,b,c and Methods)** revealed **a mild but very consistent transient delay in the ER-to-Golgi progression of Mettl9^{KO} compared to WT cells**. This confirms an alteration in the trafficking kinetics due to the loss of METTL9 in mouse developing neural cells. We also tested additional constructs probing other cargos of the secretory pathway but unfortunately, due to the challenging task of integrating the RUSH components in our mESC lines, we did not manage to get any other type of cargo expressed in a healthy and stable fashion. Indeed, **other cargo like TNF or E-Cadherin resulted to be either toxic or silenced/counter-selected during the neural differentiation of mESCs**, despite an initial expression at the mESC stage, as we did for ManII (some of the reasons might be the same as explained to the Reviewer #1 major

point 3). In the future, it would be interesting to further characterise the impact of METTL9 loss on the secretory pathway, by using CRISPR-Cas9 to insert relevant RUSH cargos in the Rosa26 locus. This would ensure that they are expressed at more physiological level and do not interfere with neuronal function. However, given the vast effort required to optimise the ManII cargo constructs during this Revision, further work in this direction was largely out of the scope and time frame of this revision.

Reviewer #5 (Remarks to the Author):

The manuscript by Pandolfini and collaborators studies the role of METTL9 in neural development. They found that the loss of Mettl9 essentially impairs differentiation at least of the ventral neuronal lineage which is both delayed and misrouted in other lineages. Then the authors proposed that the main effect that METTL9 is exerting is through regulation of its interactors within the secretory pathway to the Golgi leading to the maintenance of this latter organelle. I found the topic and part of the results really intriguing. However, I think that the manuscript in this present form is both misleading and difficult to follow. Moreover, despite some exciting data, I believe that it is (i) very preliminary (many hypotheses are of interested but not demonstrated) and (ii) based on not-solid-enough approaches (e.i. degron line). Thus, I suggest to reorganize the manuscript to underline properly the proteomic data and follow this flow for the story and provide data from newly designed experiments to functionally challenge the hypothesis (e.g. independency of catalytic activity).

We thank the Reviewer for the appreciation of our mechanistic insight. Following the suggestion of this and other Reviewers, we committed ourselves to improve the logical flow and clarity of the manuscript by rewording and rearranging several sections, and by providing further experimental evidence to support our previous hypothesis and data. We hope that the present form meets their expectations.

Below I listed some points that I noted in the manuscript in this present form:

1 - Fig S2B: please show the sequences that you get in the mutants with the “new” reading frames.

We agree with the reviewer that in the previous version of the manuscript this information was not clearly provided. In the light of the **large deletions involved (now shown in Supplementary Fig. 3c), which remove all the possible starting codons and significantly alter the splicing of the first exon**, we thought appropriate to show **the detailed genomic variants and their consequences on Mettl9 function (Supplementary Fig. 3c, lower part, table)**. Given also our qPCR (**Supplementary Fig. 3d**) and WB data (main **Fig. 2e**), all the corresponding aberrant mRNAs lack a reading frame, undergo Non-sense mediated decay (NMD) and thus they are never translated.

2 - A better WT control, than the parent line, would be a clone which underwent the same passages of the mutant ones which resulted mutation free

We acknowledge the importance of these controls, and thanks to the Reviewer's comment we realised that in the previous version of the manuscript it was sometimes unclear which WT clone/cell line was used as a control (except within the Materials and Methods section). This probably obscured the fact that **we indeed employed both a parental and clonal (non-edited, i.e. mutation free) cell line as controls. This is now more carefully specified in the manuscript and the relevant figures**. For instance, **we updated the new Supplementary Fig. 4a (PCA plot DIV5, old Fig. S2D), Supplementary Fig. 4f (PCA plot DIV10, old Fig. S2H) and also in the new panels: Supplementary Fig. 4 b,g**. We believe that this, in addition to a further, deeper characterization of the 2 control and the 2 KO cell lines (see also the previous and the next two replies), should significantly clarify and strengthen our results.

3 - An off-target analysis of the ESCs should be provided

Prompted by the Reviewer's pertinent comment, **we conducted a bioinformatics analysis to predict potential off-target sites of the CRISPR/Cas9 sgRNAs** employed to generate *Mettl9*^{KO} cell lines, then we leveraged **whole-genome sequencing to identify SNP and**

structural variants in both KO clones and their non-manipulated parental line. Cross-comparison of these data convincingly confirmed that **no off-target mutations were generated in the KO cell lines due to their genetic manipulation**: thus, these results consolidate the interpretation of our data. These analyses complement the cell line characterization and are now included in the new “Mettl9^{KO} cell line validation and off-target analysis by long-read sequencing” paragraph in the Materials & Methods section and **Supplementary Fig. 3e**.

4 - Transcriptomic data from the two KO clones should be independently indicated with the indication of the unique and common DEGs.

Thanks to the Reviewer comment, **in the present revision we provide a better characterisation of the technical and biological variance within our experiments.** Given the very high number total number of independent biological samples for the RNA-seq (more than 60, spanning three mouse cell lines and *Xenopus* tissues at different time points), when planning the experiments we decided to use two biological replicates for each *Mettl9*^{KO} cell line (KO88 and KO90). We then employed them (n=4) in the DESeq2 analysis, which performs in an optimal way with at least 3 samples (see Materials and Methods for further details). In the current experimental setting, the analysis of the DEGs for each individual KO clone would be very weak due to their numerosity (n=2). However, we felt that for the scope of this work the KO88 and KO90 could be considered largely overlapping in terms of gene expression as well as functionally, this assumption being supported by several pieces of evidence:

- **PCA of the samples used for RNA-seq analysis on *Mettl9*^{KO} cells confirms that the two clones are indeed very similar.** To emphasize this concept, we have plotted with distinct markers the individual cell lines in **Supplementary Fig. 4a** (PCA plot DIV5, old Fig. S2D), **Supplementary Fig. 4f** (PCA plot DIV10, old Fig. S2H), revealing that **most of the variance is driven by the presence or absence of METTL9 rather than differences between cell lines.**
- To further strengthen this concept, we included in the revised manuscript the **hierarchical clustering of the individual RNA-seq datasets (Supplementary Fig. 4b,g)**
- Here we provide for the Reviewer, as an additional control, the **correlations between gene-wide expression levels for each cell line at both DIV5 and DIV10.**

In our opinion, these data, taken together, might be sufficient to show that **gene regulation due to *Mettl9* knock-out dominates both inter-replicate and inter-cell line variability.**

*5 - Uninduced *Mettl9*-Deg line (i.e. with *Mettl9* expressed) should be tested for the METTL9 protein functionality: for instance, methylation of targets containing the H[ANGST]H motif, in order to confirm that the chimeric METTL9 protein is equal to WT.*

This important control, as suggested by the Reviewer, has been indeed implicitly performed **upon measuring the bulk histidine methylation levels by mass spectrometry, from mNSCs.** 1MH levels are a readout of METTL9 methyltransferase activity which, until now, is the only widely established function of METTL9. We have thus tested whether the Deg line was functional **by re-analysing the bulk 1MH levels in the uninduced *Mettl9*^{Deg} line (+DMSO) compared to WT lines, employing our existing datasets;** these were already present in main **Fig. 3i**, KO and Deg (same Y axis so comparable) and main **Fig. 4b** (same Y axis scale) but we re-plotted them here for clarity. We tested whether there was any difference among the **WT (E14 ctrl for KO, and clonal ctrl for CatD) and Degron line (DMSO-control, unperturbed condition).** This analysis shows that **1MH levels are unaffected by the presence of the Degron.** This suggests that **the *Mettl9*^{Deg} line is as functional as other parental (E14) or wt lines, in terms of METTL9-dependent methyltransferase activity in cells.** This is not surprising, since the tagged METTL9 was engineered with **a long and flexible Glycine Linker** (Reddy Chichili et al. 2013; van Rosmalen et al. 2017), to minimise the physical interference of the Degron-Flag tag with METTL9 and to preserve both MTase activity and possible protein-protein interactions. In the Materials and methods section, we now report that “The Degron construct did not interfere with METTL9 cellular methyltransferase activity, as confirmed by comparable levels of 1MH levels between untagged (WT) and (uninduced) *Mettl9*Deg lines (Fig. 3i and Fig. 4b)”.

*6 - The experiment using degron version of METTL9 are theoretically elegant and orthogonal to the complete KO. However, data pointed out that the model is basically not working properly. In fact, the transcriptomic data of *Mettl9*-Deg NSCs continuously supplied the dTAGV-1 drug is puzzling: only few DEGs retrieved in comparison with the full KO line. The authors then explained this with residual protein activity that indeed leads to the presence of 1MH level in the degron degraded cells compared to the DMSO treated (via bulk analysis). Then the question is: how these results are relevant?*

We agree with the Reviewer that in the previous version of the manuscript some data were not always clearly presented and were sometimes confusing, if not disruptive for the logical flow of the story. In contrast, in the revised manuscript, we have explained that, although imperfect, **the merit of the Degron system relies mostly on its ability to highlight and discriminate the specific effects of METTL9 loss** (which is among the reasons why the Degron approach was developed (Nabet et al. 2018)) and also explained in Reply to Reviewer #2, Minor point 5 and other Replies below). Nevertheless, we believe that the experimental data from the 5 days of differentiation in the presence of dTAG^V-1 add value to this work also because **they prove**

that low levels of METTL9 are sufficient to retain most of its neural-related function. Indeed, Mettl9 dosage seem very important: further support to this is given by our unpublished data on both mNSCs and *Xenopus*, where overexpression in mESCs (using strong promoters) is toxic and high dose of *mettl9* mRNA injected into wild-type embryos (500 pg) phenocopy the *mettl9*-K.D. morphants (see also reply to Reviewer #1 major 3). In addition, the molecular perturbations observed in this Degron system (e.g. gene misregulation, Golgi fragmentation), albeit much milder, **are concordant with those found in the other lines**; however, the results from the Degron line, **enable us to disentangle the specific** (and in the case of the 48hr dTAG experiment: **also the direct) effects of Mettl9-loss**, both because they are independently validated in more genetic and cellular systems and also because of the *inducible* METTL9 depletion via Degron (see also the paragraph “*The catalytic-dependent function of METTL9 has a secondary but convergent role in neural development and secretory system function*”). We believe that now the contribution of *Mettl9*^{Deg} has been properly put in context and compared better with the other lines, also in the light of the new wealth of experimental results, strengthening the previous assumptions.

7 - Complementation assay in MO-treated embryos with either WT or CatD Mettl9 should indicate the requirement of the enzymatic activity of Mettl9 for neural development in vivo.

Following the important suggestion prompted by the Reviewer(s), **we have performed these experiments in vivo by using *X. laevis* embryos** (see also Reply to Reviewer #1 major point 3 and Reviewer #2, point 3). **We injected either WT or CatD *mettl9* mRNA together with SPL-MO (knock-down)** and observed a comparable level of rescue of the neural phenotype (as revealed by the *elcR* mRNA marker); suggesting that ***mettl9* role in early neural development in vivo is largely independent from its enzymatic activity.** These new results **are now incorporated** into main Fig. 8e,f and relative Main Text, and significantly strengthen the message of this work.

8 - If the Mettl9 degradation is leaving residual protein able to essentially blunt the transcriptomic effect after prolonged treatment, why we should rely on data deriving from a 2 day treatment (Fig.5)?

In agreement with the Reviewer's point, in the revised manuscript we tried to better emphasise this point, that might result confusing if not properly addressed. The total **proteomic** experiment performed after **only 2 days of dTAG treatment (Fig. 5a,b) provides insights into the direct mechanism through which METTL9 protein sustains neural stem cell development in the cell.** Similarly to many other Protein-Degradation Technologies such as: SMASH (*Nat Chem Biol*, (Chung et al. 2015)), Auxin (*Nat Methods*, (Nishimura et al. 2009)), HALO-Protac (*ACS Chem Biol*, (Buckley et al. 2015)), the dTAG/Degron system (*Nat Chem Biol*, (Nabet et al. 2018)) **enables the timely protein degradation of most of METTL9 in the cell**; this allows us **to capture at the mechanistic level which proteins are affected right after the induced METTL9 depletion** (proteomic analysis), **before many indirect molecular cascades and other unrelated-cellular events might be put in place by the cell, as a consequence of prolonged METTL9 reduction.** While these effects are **expected to be mild**, they are nonetheless **highly informative.** Moreover, their occurrence following even an incomplete reduction in METTL9 levels underscores their relevance and highlights a tight functional connection. At any rate in the manuscript (as well as in the real course of the investigation) some of the experiments with the Degron line have been performed as *pathfinders*, subsequently supported by orthogonal pieces of evidence (as for the total Proteomics). Differently, the RNA-seq performed after a longer dTAG treatment (6 days, until DIV5) cannot inform us much on the direct mechanism through which METTL9 acts in the cell, because it also embeds *some of* the consequences of indirect effects upon METTL9 depletion (also because it is a transcriptomic rather than a proteomic experiment). This long-term experiment was useful to highlight the potential phenotypic consequences of METTL9 loss in

neural fate commitment, which we assessed through transcriptomic analysis (and immunofluorescence staining with Nestin). Overall, **the Gene Ontologies (related to the secretory system) concerning the proteomic experiment after acute depletion (2 days of dTAG) are confirmed (and cross-validated) by another proteomic dataset (see reply to the point below, i.e. the proteomics on KO) and by many other functional experiments in multiple genetic systems (besides by the IP-MS, which found key protein interactors involved in the secretory pathway).** However, **the acute depletion (2 days) experiment is key to determine that the Secretory pathway is modulated by METTL9 as an early molecular event.** Thus, the dTAG system, although imperfect, is still useful to understand METTL9 function. In the future, it could be interesting to engineer alternative systems allowing to precisely modulate, also in a timely, inducible manner, the cellular levels of METTL9 to confirm our mechanistic data about the immediate molecular effects of METTL9 reduction, generated here with the *Mettl9^{Deg}* line. However, for the current study and revision, we focussed on trying to better integrate the results already obtained with the 3 different mESC systems and the *Xenopus* model *in vivo*.

9 - Are the differentially enriched proteins found in the degron line, also affected in the full KO?

A brief answer to this important question is that **the differentially enriched proteins are largely shared, but their regulation goes in the opposite way** in the two loss-of-function models. Initially during the early stages of our work, we reasoned that the proteomic analysis of *Mettl9^{KO}* cells, contrary to the acute depletion by degron, could not easily uncouple the direct molecular effects of METTL9 absence from the indirect, long-term consequences on cell fate and homeostasis, due to the lack of the METTL9 protein for many days (see previous point). For this reason, we deemed more informative for the initial exploration of METTL9 function to focus on the acute degradation. However, prompted by the reviewer's comment, **we have conducted and included in the manuscript a proteomic analysis of WT and *Mettl9^{KO}* cells at DIV5 (Supplementary Fig. 8 c,d; Supplementary Data 3).** In line with our predictions, *Mettl9^{KO}* cells at DIV5 display **a significant high number of up- and down-regulated proteins (329 and 395, respectively).** By **directly comparing this experiment with the dTAG (acute depletion),** we found that, although **the biological processes misregulated were consistent (e.g. secretory-pathway related), the regulation of proteins is largely opposite,** with a robust statistical significance, both in terms of the distribution of log2 Fold Changes, as shown in **Supplementary Fig. 8e,** and in the number of differential proteins (Chi-squared p-value = 0.000393). A possible explanation could be: **upon acute METTL9 depletion (dTAG) the secretory system is perturbed and accumulates secretory-related proteins (up-regulation of these GO terms in the dTAG); then, in the long term, after persistent METTL9 loss (KO) the secretory system is greatly perturbed, and in fact the Golgi is fragmented, and the ER-to-Golgi trafficking is impaired (new RUSH experiment).** This in turn, hinders the function of the secretory system (down-regulation in the KO). This mechanism of opposite protein regulation might be implemented by the *Mettl9-KO* cells to cope with the lack of METTL9 and ensure their survival. This inverse relation between the results from the inducible and constitutive experiments in our opinion reinforces the concept that an important set of ER/Golgi/Secretory factors is under the tight, homeostatic control of METTL9, and it demonstrates that the mild effects detected upon 2 days of dTAG administration to the *Mettl9^{Deg}* line are indeed **specific and informative.** In addition, the fact that the regulation goes in opposite directions (GO secretory pathway up-regulated in the Degron and down-regulated in the KO) further **reinforces the value of the dTAG line in disentangling the direct and indirect effects of METTL9 loss, providing mechanistic insight into the cellular role of METTL9.** Furthermore, these proteomics results in the *Mettl9 KO* are also in agreement with the transcriptomics data, confirming a general impairment of the neural differentiation trajectory, with many neural stem cell markers down-regulated (e.g. NESTIN, MUSASHI 1 and 2,

TUBULIN-beta-4A) and markers of undifferentiated, highly proliferating cells up-regulated (e.g. POU5F1, UTF1, CYCLIN A2, CATENIN-beta 1). Such a trend is also reflected in the GO enrichment analysis (e.g. Nervous system development – Down-regulated).

References

- Basu A, Mestres I, Sahu SK, Tiwari N, Khongwir B, Baumgart J, Singh A, Calegari F, Tiwari VK. 2020. Phf21b imprints the spatiotemporal epigenetic switch essential for neural stem cell differentiation. *Genes Dev* **34**: 1190–1209.
- Boncompain G, Divoux S, Gareil N, de Forges H, Lescure A, Latreche L, Mercanti V, Jollivet F, Raposo G, Perez F. 2012. Synchronization of secretory protein traffic in populations of cells. *Nat Methods* **9**: 493–498.
- Buckley DL, Raina K, Darricarrere N, Hines J, Gustafson JL, Smith IE, Miah AH, Harling JD, Crews CM. 2015. HaloPROTACS: Use of Small Molecule PROTACs to Induce Degradation of HaloTag Fusion Proteins. *ACS Chem Biol* **10**: 1831–1837.
- Chung HK, Jacobs CL, Huo Y, Yang J, Krumm SA, Plemper RK, Tsien RY, Lin MZ. 2015. Tunable and reversible drug control of protein production via a self-excising degron. *Nat Chem Biol* **11**: 713–720.
- Ignatova VV, Jansen PWTC, Baltissen MP, Vermeulen M, Schneider R. 2019. The interactome of a family of potential methyltransferases in HeLa cells. *Sci Rep* **9**: 6584.
- Liu C, Song X, Nisbet R, Götz J. 2016. Co-immunoprecipitation with Tau Isoform-specific Antibodies Reveals Distinct Protein Interactions and Highlights a Putative Role for 2N Tau in Disease. *J Biol Chem* **291**: 8173–8188.
- Morrow CS, Porter TJ, Xu N, Arndt ZP, Ako-Asare K, Heo HJ, Thompson EAN, Moore DL. 2020. Vimentin coordinates protein turnover at the aggresome during neural stem cell quiescence exit. *Cell Stem Cell* **26**: 558-568.e9.
- Nabet B, Roberts JM, Buckley DL, Paulk J, Dastjerdi S, Yang A, Leggett AL, Erb MA, Lawlor MA, Souza A, et al. 2018. The dTAG system for immediate and target-specific protein degradation. *Nat Chem Biol* **14**: 431–441.
- Nishimura K, Fukagawa T, Takisawa H, Kakimoto T, Kanemaki M. 2009. An auxin-based degron system for the rapid depletion of proteins in nonplant cells. *Nat Methods* **6**: 917–922.
- Pintacuda G, Hsu Y-HH, Tsafou K, Li KW, Martín JM, Riseman J, Biagini JC, Ching JKT, Mena D, Gonzalez-Lozano MA, et al. 2023. Protein interaction studies in human induced neurons indicate convergent biology underlying autism spectrum disorders. *Cell Genomics* **3**: 100250.
- Reddy Chichili VP, Kumar V, Sivaraman J. 2013. Linkers in the structural biology of protein–protein interactions. *Protein Sci Publ Protein Soc* **22**: 153–167.
- Urbán N, van den Berg DLC, Forget A, Andersen J, Demmers JA, Hunt C, Ayrault O, Guillemot F. 2016. Return to quiescence of murine neural stem cells by degradation of a pro-activation protein. *Science* **353**: 292–295.
- Uzbas F, O'Neill AC. 2023. Spatial Centrosome Proteomic Profiling of Human iPSC-derived Neural Cells. *Bio-Protoc* **13**: e4812.
- van Rosmalen M, Krom M, Merckx M. 2017. Tuning the Flexibility of Glycine-Serine Linkers To Allow Rational Design of Multidomain Proteins. *Biochemistry* **56**: 6565–6574.

Reviewer #1 (Remarks to the Author):

The authors have addressed all of my concerns. The revised version is much improved. However, it remains too comprehensive and detail-heavy, which limits readability. I recommend simplifying the structure and clearly highlighting the key point -non-catalytic function - prior to publication.

We are pleased that our revision has addressed all the Reviewer's concerns, and we sincerely thank them for the important suggestions that have substantially improved our manuscript. While we recognize that further streamlining and simplifying the text might help presenting the core ideas more succinctly, we are concerned that doing so could compromise the completeness and integrity of the scientific content, which has been positively evaluated by the other four reviewers. Indeed, an important part of our revision was devoted to clarifying and highlighting the key findings of this manuscript, sometimes by providing more details as requested by other reviewers in specific cases. Therefore, we would respectfully refrain from making substantial revisions to the text unless there is a clear recommendation from the Editor to do so, based on a comprehensive consideration of all the reviewers' comments.

Reviewer #2 (Remarks to the Author):

The concerns we raised have been satisfactorily addressed, and we now endorse the publication of this manuscript.

We are glad that our revised manuscript addressed all the Reviewer's concerns, and we thank them for the insightful suggestions that substantially improved our manuscript.

Reviewer #3 (Remarks to the Author):

We would like to once again express our appreciation for the time dedicated to evaluating our work.

Reviewer #4 (Remarks to the Author):

The authors have satisfactorily answered the reviewers' comments.

We are pleased that our revised manuscript has addressed all the Reviewer's concerns, and we sincerely appreciate the constructive feedback provided, which contributed to substantially strengthen our work.

Reviewer #5 (Remarks to the Author):

I found the revised version of the manuscript to be significantly improved in terms of clarity, coherence, and overall structure. The authors have made a commendable effort in addressing the main concerns raised during the previous review round. The manuscript now presents its objectives and findings in a more precise and accessible manner, which greatly enhances its readability and scientific impact.

Both the strengths and the limitations of the work are now more explicitly acknowledged, which demonstrates a mature and balanced approach to the research. While some aspects still remain open or insufficiently explored, the integration of new methodologies and analyses has provided a more robust foundation for the results. These additions have not only reinforced the main conclusions but also contributed to a clearer and more convincing set of take-home messages.

Overall, the manuscript has advanced considerably, and although certain points may benefit from further clarification in future work, the current version marks a substantial step forward in terms of scientific quality and communicative effectiveness.

We sincerely thank the Reviewer for their thorough evaluation and valuable insights, and we are pleased that the revised manuscript has successfully addressed their concerns.